# A ligand-specific blockade of the integrin Mac-1 selectively targets pathologic inflammation while maintaining protective host-defense

Dennis Wolf [1,2], Nathaly Anto-Michel[1], Hermann Blankenbach[1], Ansgar Wiedemann[1], Konrad Buscher[2],
Jan David Hohmann[3], Bock Lim[3], Marina Bäuml[1], Alex Marki[2], Maximilian Mauler[1], Daniel Duerschmied[1],
Zhichao Fan [2], Holger Winkels[2], Daniel Sidler[4], Philipp Diehl[1], Dirk M Zajonc[5], Ingo Hilgendorf[1],
Peter Stachon[1], Timoteo Marchini[1], Florian Willecke[1], Maximilian Schell[1,2], Björn Sommer[6],
Constantin von zur Muhlen[1], Jochen Reinöhl[1], Teresa Gerhardt[2], Edward F. Plow[7], Valentin Yakubenko[7],
Peter Libby[8], Christoph Bode[1], Klaus Ley[2], Karlheinz Peter[3] & Andreas Zirlik[1]

Integrin-based therapeutics have garnered considerable interest in the medical treatment of inflammation. Integrins mediate the fast recruitment of monocytes and neutrophils to the site of inflammation, but are also required for host defense, limiting their therapeutic use. Here, we report a novel monoclonal antibody, anti-M7, that specifically blocks the interaction of the integrin Mac-1 with its pro-inflammatory ligand CD40L, while not interfering with alternative ligands. Anti-M7 selectively reduces leukocyte recruitment in vitro and in vivo. In contrast, conventional anti-Mac-1 therapy is not specific and blocks a broad repertoire of integrin functionality, inhibits phagocytosis, promotes apoptosis, and fuels a cytokine storm in vivo. Whereas conventional anti-integrin therapy potentiates bacterial sepsis, bacteremia, and mortality, a ligand-specific intervention with anti-M7 is protective. These findings deepen our understanding of ligand-specific integrin functions and open a path for a new field of ligand-targeted anti-integrin therapy to prevent inflammatory conditions.

[1] Cardiology and Angiology I, University Heart Center, and Medical Faculty, University of Freiburg, Freiburg, 79106, Germany. [2] Inflammation Biology, La Jolla Institute for Allergy and Immunology, La Jolla, CA 92037, USA. [3] Atherothrombosis and Vascular Biology, Baker Heart and Diabetes Institute, Melbourne, 8008 VIC, Australia. [4] Division of Nephrology, Inselspital, Bern University Hospital, Bern, 3010, Switzerland. [5] Division of Cell Biology, La Jolla Institute for Allergy and Immunology, La Jolla, CA 92037, USA. [6] Neurosurgery, Medical Faculty of the University of Erlangen, Erlangen, 91054, Germany. [7] Department of Molecular Cardiology, Lerner Research Institute, Cleveland Clinic, Cleveland, OH 44195, USA. [8] Brigham and Women's Hospital, Cardiovascular Medicine, Harvard Medical School, Boston, MA 02115, USA. Dennis Wolf and Nathaly Anto Michel contributed equally to this work. Karlheinz Peter and Andreas Zirlik jointly supervised this work. Correspondence and requests for materials should be addressed to K.P. (email: karlheinz.peter@baker.edu.au)

Inflammation drives many diseases, including atherosclerosis[1,2], type 2 diabetes[3], neurodegeneration[4], and sepsis[5]. Targeting the inflammatory response might ameliorate these conditions[6]. Yet, the critical role of the inflammatory response in many biological processes such as regeneration, thrombosis, and host defense presents a major limitation to such strategies[7]. For example, glucocorticoids potently inhibit inflammation, but have multiple undesired actions[8]. COX-2 inhibitors can suppress inflammation, but nonetheless worsen cardiovascular outcomes[9].

Inflammation involves the recruitment of leukocytes to the site of injury, typically facilitated by integrins such as Mac-1 ($\alpha_M\beta_2$, CD11b/CD18)[10]. The adhesion molecule Mac-1 can undergo rapid activation yielding a conformational change that increases affinity for its ligands that enable it to mediate rolling, firm adhesion, and transmigration of leukocytes into inflamed tissue[11–13]. Therapeutic or genetic inhibition of Mac-1 highly effectively limits experimental atherosclerosis[14], neo-intima formation[15,16], adipose tissue inflammation[17], ischemic kidney injury[18], and glomerulonephritis[19,20]. Beyond its role in inflammation, Mac-1 was initially named CR3 (complement receptor 3) due to its ability to bind complement factors, such as iC3b[21], reflecting its broad role in host defense[22–24], wound healing[25], thrombosis[26,27], and various other myeloid cell effector functions[28–30]. Myeloid cells, including monocytes, macrophages, and neutrophils express Mac-1, as do NK cells, and to a smaller extent activated lymphocytes[31]. Mac-1's functional diversity is reflected by ligand binding to a large repertoire of proteins and proteoglycans, including ICAM-1[32], fibrinogen[33], fibronectin[34], vitronectin[34], heparin[35], GPIbα[26], RAGE[36], endothelial protein C-receptor (EPCR)[37], CD40L[14], and others[38]. Inhibition of Mac-1 could thus serve as a promising therapeutic strategy in inflammatory disease[39,40]. Its major role in host defense, regeneration, and thrombosis, however, could limit its therapeutic applicability.

To overcome these limitations, we hypothesized that the inactivation of distinct integrin functions involved in inflammatory, but not in regenerative or immune pathways, could result from selective blockade of Mac-1's interaction to specific ligands, while not affecting others. For proof-of-concept studies we designed a monoclonal antibody, that targets specifically the EQLKKSKTL motif in Mac-1, required to bind to its multipotent ligand CD40L[14,41,42]. We successfully generated this antibody and compared its effect to conventional anti-Mac-1 blockade experimentally in in vivo leukocyte recruitment, peritoneal inflammation, sterile and polymicrobial sepsis. In conclusion, we report that a ligand-specific anti-Mac-1 therapy is superior to unspecific, conventional blocking strategies —in particular in conditions that are driven by inflammation and impaired host defense simultaneously, such as polymicrobial sepsis.

## Results

### The antibody anti-M7 targets the Mac-1/CD40L-binding site.
We previously demonstrated that CD40L selectively binds to the EQLKKSKTL motif (M7) within the Mac-1 ligand-binding I-domain[41]. To generate a specific inhibitor of the human binding site that can bind and block the M7 motif within the Mac-1 I-domain, we immunized mice with the human peptide V160-S172 coupled to diphtheria toxoid. The M7 sequence is highly conserved between the human and murine protein sequence (Fig. 1a). Among several hybridoma clones that demonstrated high-affinity binding to the immobilized peptide M7 in a solid-phase binding assay, one clone, termed anti-M7 (mouse IgG2bκ), showed a specific inhibition of Mac-1-CD40L binding, but not of the binding to other ligands. Anti-M7 bound to a CHO cell line that overexpresses non-activated human Mac-1 (Mac-1 WT) and

permanently activated human Mac-1 (Mac-1 del)[43], but did not bind to control CHO cells (CHO) in western blot (Fig. 1b, Supplementary Figure 1), demonstrating that the antibody binds to its intended target protein. Anti-M7 bound in a concentration-dependent manner to the immobilized peptides M7 (EQLKKSKTL), but not to the control peptides scrambled sM7 (KLSLEKQTK) or the peptide M8 (EEFRIHFT), which locates near the peptide sequence M7 (Fig. 1c). To quantify the binding of anti-M7 to Mac-1 expressing human cells, we coupled the antibody to the fluorochrome Alexa-647 and validated that anti-M7 bound concentration-dependently to human leukocytes that express Mac-1, such as monocytes and neutrophils (Fig. 1d). These findings demonstrate that anti-M7 specifically binds to the peptide sequence M7 within the intact human Mac-1 I-domain. To test whether anti-M7 blocks the functional interaction of human Mac-1 with human CD40L, we tested the adhesion of CHO-Mac-1 del cells to immobilized CD40L in a static adhesion assay. Anti-M7 blocked cell adhesion by $65.6 \pm 7.2\%$ (mean $\pm$ SEM), an effect nearly as strong as the anti-human Mac-1 reference clone 2LPM19c, a pan I-domain blocking antibody (inhibition by $92.7 \pm 2.0\%$, Fig. 1e, f). Notably, the adhesion of CHO cells expressing the non-permanently activated human Mac-1 (WT Mac-1) was neutralized by two conventional anti-human Mac-1 clones (2LPM19c and IRF44), but not by anti-M7, indicating that anti-M7 preferentially blocked the integrin's high-affinity interaction with CD40L. CHO cells not expressing Mac-1 did not adhere to CD40L (Supplementary Figure 2). Different ligands can bind to separate or overlapping binding regions within the Mac-1 I-domain. To test whether anti-M7 is functionally specific for CD40L binding, a panel of established Mac-1 ligands, including fibrinogen, vitronectin, JAM-C, ICAM-1, NIF, heparin, and RAGE, were separately immobilized and binding of Mac-1-del cells was tested in the presence of anti-M7 and anti-Mac-1 (2LPM19c), respectively. Anti-Mac-1 blocked each of the interactions highly efficient, while anti-M7 blocked cell adhesion to CD40L only (Fig. 1f). Another control experiment verified that $F_{ab}$ fragments of anti-M7 sufficed for its specific blocking activity without depending on unspecific Fc-receptor interactions (Fig. 1g). These data demonstrate that anti-M7 specifically inhibits the CD40L/Mac-1 interaction.

### Anti-M7 prevents inflammatory leukocyte recruitment.
Mac-1 is a prerequisite for leukocyte recruitment in inflammation[11–13]. It serves as receptor for different ligands expressed on the inflamed endothelium, including ICAM-1, RAGE, and CD40L and mediates slow rolling, firm adhesion, and subsequent transmigration of myeloid cells[11,36,41]. We have recently shown that a peptide-based inhibition of the CD40L binding epitope that interacts with Mac-1 protects from leukocyte accumulation by efficiently blocking cellular adhesion[41]. Here, we interrogated whether blocking the M7-binding sequence directly would be equally efficient in vitro and in vivo. First, we verified that the mouse anti-human M7 antibody blocks the static adhesion of Mac-1 expressing mouse peritoneal macrophages and mouse CD40L (Fig. 2a). An anti-mouse Mac-1 antibody (clone M1/70) served as control. Next, we tested whether anti-M7 blocks the cellular adhesion under physiological flow conditions. Therefore, murine monocytic RAW-cells were allowed to adhere on TNFα-primed primary, murine endothelial cells in vitro in a flow chamber assay. The number of adhering cells decreased after incubation with anti-M7, indicating a requirement for CD40L/Mac-1 interaction for leukocyte arrest on inflamed endothelial cells (Fig. 2b). To test the antibody's in vivo applicability, we prepared $F_{ab}$ fragments of anti-M7, anti-Mac-1 (M1/70), a control-IgG, and antibodies directed against ICAM-1, LFA-1

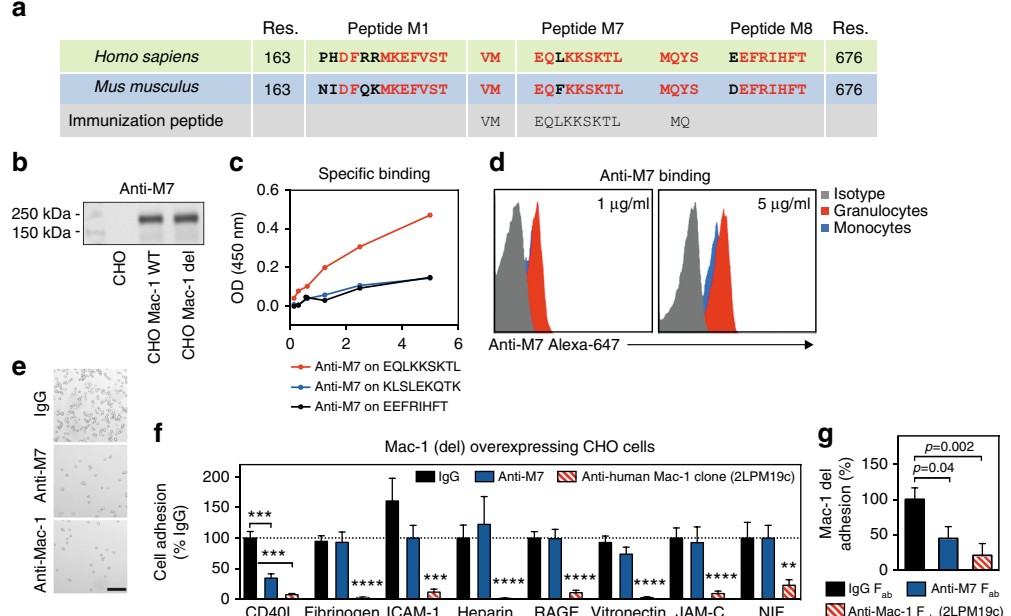

**Fig. 1** The novel monoclonal antibody anti-M7 selectively blocks the CD40L-binding site within the Mac-1 (CD11b/CD18) I-domain. **a** The peptide sequence M7 within the Mac-1 I-domain, required for binding of CD40L, is a highly conserved binding motif between the human and mouse integrin. The antibody anti-M7 was generated by immunization of mice with the binding peptide VMEQLKKAKTLMQ coupled to diphtheria toxoid. **b** Anti-M7 bound to a CHO cell line over-expressing native human (Mac-1 WT) and permanently activated human Mac-1 (Mac-1 del), but not to control CHO cells that did not express Mac-1 (CHO) in western blot. **c** Specific binding of the antibody anti-M7 to the immobilized peptides M7 (EQLKKSKTL), the scrambled control sM7 (KLSLEKQTK), and the closely located peptide M8 (EEFRIHFT) in a solid-phase binding assay with immobilized peptides. **d** Anti-M7 was coupled with the fluorochrome Alexa-647 and binding to human leukocyte subsets was quantified in flow cytometry. An Alexa-647 labeled IgG isotype antibody served as control. **e**, **f** Adhesion of CHO cells over-expressing the permanently activated Mac-1 mutant (del) on dishes coated with immobilized human CD40L or alternative human Mac-1 ligands in a static adhesion assay. Cells were incubated with anti-M7 or the human pan I-Domain blocking anti-human reference anti-Mac-1 (clone 2LPM19c) 15 min prior to adhesion. **g** CHO-Mac-1 cell adhesion on immobilized CD40L was blocked with $F_{ab}$ fragment preparations of IgG, anti-M7 or anti-human Mac-1 (2LPM19c). Error bars indicate mean ± SEM. Statistical significance was assessed by an unpaired, two-sided Student's T-test between the indicated groups, **P < 0.01, ***P < 0.001, ****P < 0.0001 (**f**, **g**) or against IgG if not otherwise indicated (**f**). Data are the result of $N \geq 3$ independent experiments. Representative pictures are shown in **b**, **e**. res. indicates the location/residue within the Mac-1 protein sequence. Scale bar (**e**) represents 100 μm

(CD11/CD18), and RAGE to avoid unspecific $F_c$-receptor interactions. Anti-M7 did not change peripheral leukocyte counts after an i.p. injection (Supplementary Figure 3). In intravital microscopy (Fig. 2c) of inflamed mesenteric venules, the number of adhering (Fig. 2d) but not of rolling leukocytes (Fig. 2e) fell after anti-M7 injection. In accordance, leukocyte rolling velocity, displayed as cumulative frequency, did not change (Fig. 2f). Notably, this effect was comparable to the effect after blockade of the receptor/ligand pairs Mac-1/ICAM-1, LFA-1/ICAM-1, and Mac-1/RAGE. Anti-M7 failed to reduce leukocyte recruitment in Mac-1$^{-/-}$ mice (Fig. 2g), suggesting specificity of the antibody. In addition, we excluded that anti-M7 induces leukocyte depletion after injection (data not shown). To test whether impaired monocyte arrest would affect down-stream effects of the integrin, such as leukocyte transmigration, mice expressing GFP in monocytes (CX3CR1-GFP) were subjected to intravital micro-scopy in the presence of IgG or anti-M7 $F_{ab}$ preparations after a TNFα challenge for 4 h (Fig. 2h). Fewer monocytes migrated to the para-vascular space after anti-M7 treatment (Fig. 2i). Leu-kocyte recruitment during sepsis and abdominal inflammation caused by an i.p. TNFα injection occurs through high endothelial venules (HEVs) in the greater omentum[44]. Therefore, we applied confocal intravital microscopy of this region after local TNFα treatment of LysM-GFP reporter mice that express GFP in myeloid cells. Anti-M7 treatment blocked the emigration of LysM$^+$ myeloid cells to the para-vascular space (Fig. 2j, left and middle panel) and induced a predominant crawling behavior of

LysM-GFP$^+$ neutrophils in the vascular lumen (right panel), indicating a defect in Mac-1-mediated firm adhesion. Time lapse tracking of single cells demonstrated that myeloid cells in mice injected with anti-M7 were less likely to transmigrate from inflamed venules. In addition to lowered cell adhesion, we observed that plasma levels of the pro-inflammatory cytokines TNFα, IL-6, and MCP-1 fell significantly in mice after an anti-M7 treatment (Fig. 2k). Finally, anti-M7 efficiently reduced the number of macrophages in the peritoneal cavity 72 h after induction of a sterile, thioglycollate-induced peritonitis (Fig. 2l). In addition, anti-M7 improved, while an anti-Mac-1 treatment enhanced, the levels of pro-inflammatory cytokines in the peri-toneal cavity (Supplementary Figure 4). These results indicate that leukocyte adhesion, transmigration, and accumulation pro-ceeds in vitro and in vivo via binding of CD40L with Mac-1 —an interaction that anti-M7 effectively blocked.

**Anti-M7 does not induce integrin outside-in signaling.** Besides mediating a leukocyte-endothelial cell or leukocyte-leukocyte cross-talk, Mac-1 engagement by soluble or membrane-bound ligands can initiate down-stream pro-inflammatory signaling events[13,17,42]. Such outside-in-signaling[45], which comprises activation of the MAP-kinases ERK and p38 is also observed after anti-Mac-1 antibody binding[7,46]. We previously showed that CD40L is a biased agonist that does not induce outside-in signaling events[38,41]. Our observation that a treatment with anti-

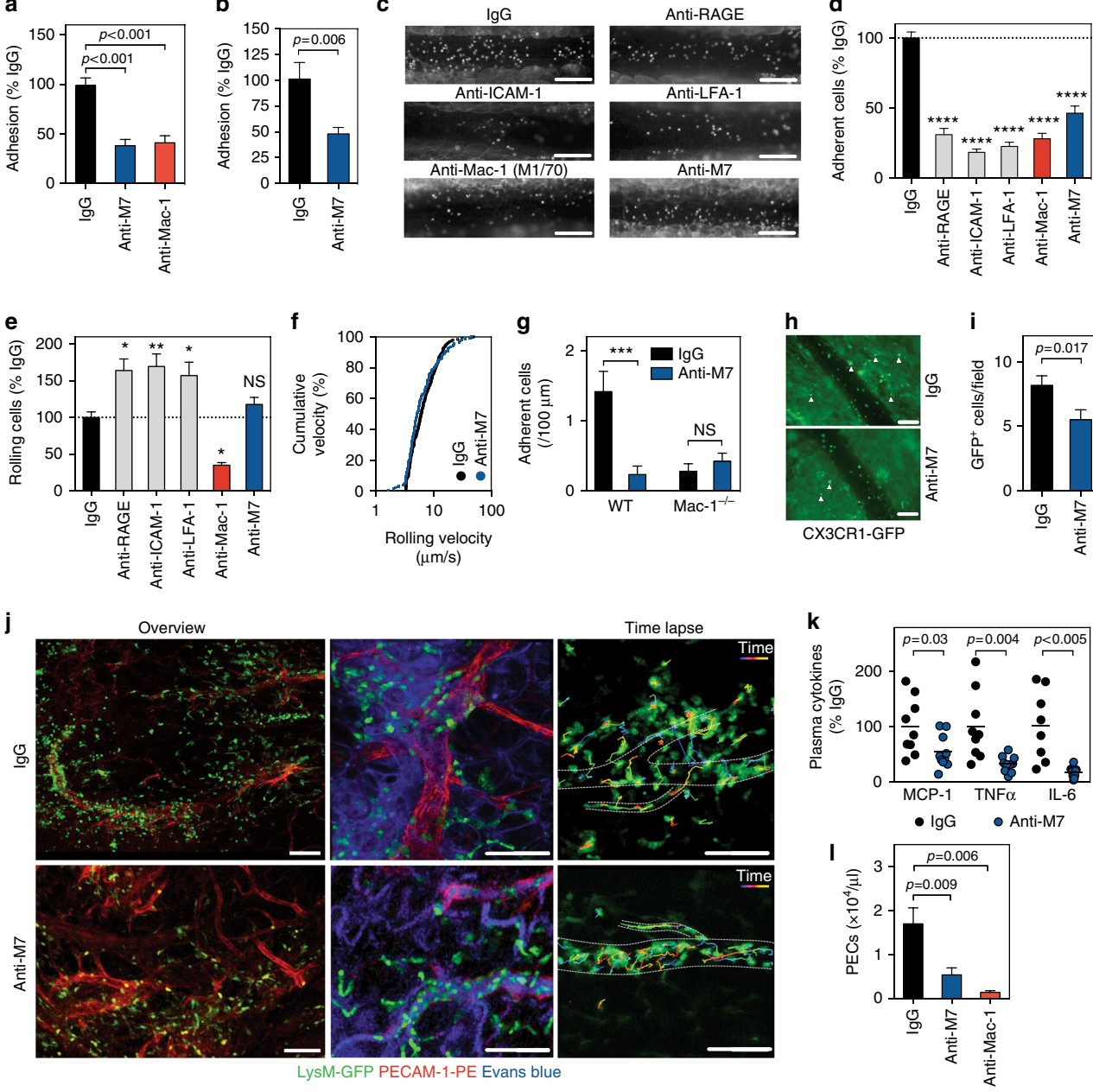

**Fig. 2** Treatment with anti-M7 prevents inflammatory leukocyte recruitment in vitro and in vivo. **a** Murine peritoneal macrophages were incubated on dishes coated with mouse CD40L in the presence of 10µg/ml of IgG, anti-M7, or anti-Mac-1 (clone M1/70). Adhering cells were normalized to % of the IgG-treatment. **b** Murine RAW-cell adhesion on TNFα-primed murine endothelial cells under physiological flow in the presence of IgG or anti-M7 antibodies (10 µg/ml). **c–f** C57Bl/6 mice were injected i.p. with 200 ng TNFα and 15 min before microscopy with $F_{ab}$ fragments of an IgG control antibody, or antibodies against RAGE, ICAM-1, LFA-1, Mac-1 (clone M1/70), or anti-M7 (100 µg i.p.). Leukocyte recruitment was monitored by intravital microscopy 4 h later: Adhering (**d**) and rolling leukocytes (**e**) were quantified as % of IgG. (**f**) Cumulative frequency of leukocyte rolling velocity. **g** Leukocyte adhesion in intravital microscopy in IgG or anti-M7 treated WT and Mac-1$^{-/-}$ mice. **h, i** Number of CX3CR1-GFP$^+$ monocytes (white arrows) in the para-vascular space 4 h after treatment with IgG or anti-M7 and 200ng TNFα. **j** Confocal in vivo imaging of the greater omentum after local stimulation with TNFα and i.v. injection of the indicated antibodies (100 µg) in LysM-GFP reporter mice. LysM-GFP$^+$ myeloid cells were tracked over time. Time lapse and cell tracking is shown over 15 min (right panel). **k** Plasma cytokine levels of mice subjected to intravital microscopy after IgG or anti-M7 $F_{ab}$ treatment. **l** Peritoneal exudate cells (PECs) 72 h after induction of a sterile peritonitis and the indicated antibody treatment. Scale bars represent 100µm (**c, h, j**). Error bars indicate mean ± SEM. Statistical significance was assessed by a two-sided, unpaired Student's *T*-test between the indicated groups (**a, b, g, i, k, l**) or in comparison to IgG-treatment (**d, e**). *$P < 0.05$, **$P < 0.01$, ***$P < 0.001$, ****$P < 0.0001$. $N \geq 3$ independent experiments (**a, b**). $N \geq 10$ mice per group (**d, e, f, g**), $N \geq 6$ mice per group (**l, h**).

Mac-1 protected from recruitment to the peritoneal cavity (Fig. 2l) while enhancing pro-inflammatory cytokine expression (Supplementary Figure 4), tempted us to verify potential outside-in signaling events promoted by anti-M7 and anti-Mac-1. Therefore, we collected thioglycollate-elicited peritoneal macrophages from male, 8-week-old C57Bl/6 mice and incubated these with 10 µg/ml of either mouse IgG, anti-human Mac-1 (clone 2LPM19c) as species-mismatch control, anti-mouse Mac-1 (clone M1/70), or anti-M7 for 30 min. Anti-Mac-1 treatment (M1/70) induced phosphorylation of ERK and p38 as quantified by an elevated ratio of the phosphorylated proteins in western blotting (Fig. 3a, b, Supplementary Figure 5, 6). Species-mismatched anti-Mac-1 (2LPM19c) or anti-M7 showed no activation of these MAP-kinases, indicating that the binding epitope M7 targeted by anti-M7 is not involved in outside-in signaling. To assess the in vivo relevance of such anti-Mac-1 agonism, mice received anti-Mac-1 antibodies from various clones i.p. and serum concentration of IL-6, TNFα, and MCP-1 were quantified 4 h after injection. The anti-Mac-1 reference clone in the mouse, M1/70, strongly elevated cytokine levels, while anti-M7 did not (Fig. 3c). Pro-inflammatory cytokine concentrations also increased in cultured mouse macrophages after antibody co-incubation (data not shown). Vice versa, human monocyte-derived macrophages showed the same pattern in anti-M7 and anti-human Mac-1 (reference clone 2LPM19c) stimulated macrophages (Fig. 3d). To further verify that anti-M7 does not act as an agonist for Mac-1, we quantified binding of the two anti-human β2-subunit antibodies KIM127 and Mab24, which are indicating cellular activation and subsequent changes in integrin conformation by detecting β2-extension (KIM127) and β2 high-affinity conformation (Mab24)[47], in human primary neutrophils in a real-time flow cytometry binding assay (Fig. 3e–i). Only binding of Mab24 was slightly reduced after co-incubation with the human chemokine IL-8, which is required for full integrin

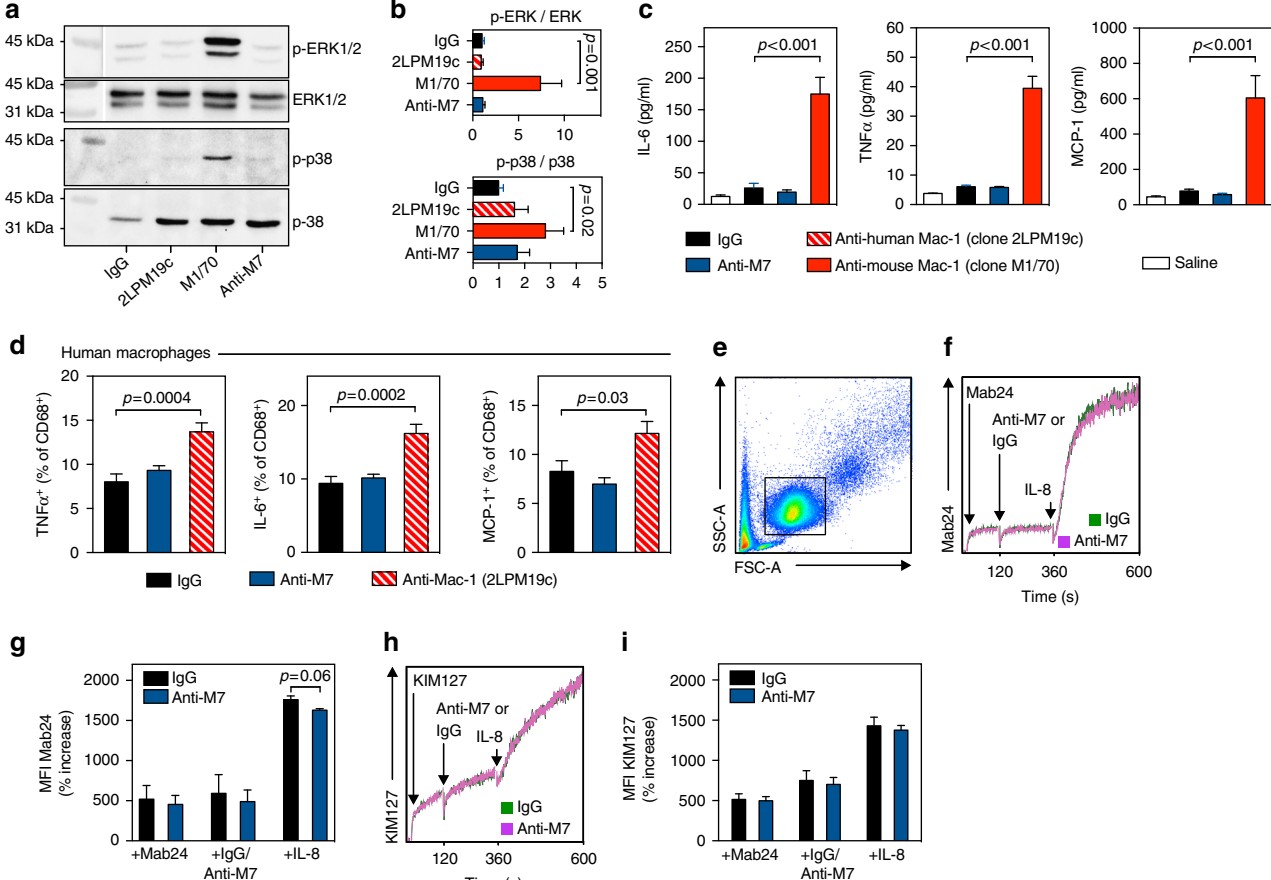

**Fig. 3** Conventional anti-Mac-1 therapy, but not a ligand-specific inhibition by anti-M7, is a potent inducer of integrin outside-in signaling and aberrant cytokine secretion. **a, b** Mouse macrophages were isolated from the peritoneal cavity of C57Bl/6 mice 72 h after injection of 4% thioglycollate. Peritoneal cells were collected by lavage and the purity was confirmed by flow cytometry (>90% F4/80+ macrophages). Cells were cultured in 5%FCS/RPMI overnight and co-incubated with 10 µg/ml of mouse IgG, anti-human Mac-1 (clone 2LPM19c), anti-mouse Mac-1 (clone M1/70), or anti-M7 for 30 min in the presence of an anti-mouse CD16/CD32 Fc-block. Cells were lysed and total and phosphorylated ERK1/2 and p38 were visualized by western blot and the ratio of phosphorylated fractions was calculated. Values were calculated as relative arbitrary units (AU) normalized to signal of cells stimulated with saline alone. **c** Saline, or IgG, anti-M7, anti-Mac-1 (clone M1/70) antibodies were injected i.p. in mice and plasma concentrations of IL-6, TNFα, and MCP-1 were measured by cytometric bead array 4 h after injection. **d** Human macrophages were differentiated from peripheral monocytes and co-incubated with the indicated antibody clones including the anti-human Mac-1 clone 2LPM19c in the presence of an anti-human CD16/CD32 Fc-block for 5 h. Cytokines were quantified by intracellular flow cytometry in CD68+ macrophages. **e** Human neutrophils were enriched by density centrifugation of peripheral blood. **f–i** The anti-β2 antibodies Mab24 (detecting the β2 high-affinity conformation) and KIM127 (for β2-extension) and anti-M7 or IgG (10 µg/ml) were added during acquisition in flow cytometry. The mean fluorescence intensity (MFI) for Mab24 (**f, g**) and KIM127 (**h, i**) was continuously recorded over 10 min within the neutrophil gate (**e**) and quantified as % of the starting MFI (**g, i**). Error bars indicate mean ± SEM. Statistical significance was assessed by a two-sided, unpaired Student's T-test between the indicated groups. Data are the result of N ≥ 3 independent experiments (**a, b**). N ≥ 9 mice per group (**c**) N ≥ 6 human donors (**d**). N = 3 human donors (**f–i**). A representative FACS plot is shown in **e**

activation, and anti-M7, while we observed no changes in the baseline activation. These findings suggest that anti-M7 targets an epitope on Mac-1 that does not cause outside-in signaling and pro-inflammatory cytokine secretion in Mac-1 expressing cells during anti-integrin therapy.

**A ligand-specific blockade of Mac-1 improves sepsis.** Aberrant activation of leukocytes and excessive cytokine secretion increase disease severity and mortality in sepsis[48–50]. On the other hand, integrins are required for host defense, demonstrated by the requirement for Mac-1 to clear bacteria[22–24,51]. Clinically, blood concentrations of soluble CD40L[52] and sCD18[53] are associated with sepsis, suggesting that both, Mac-1 and CD40L,

participate in the acute inflammatory response in sepsis. To outweigh its conflicting effects on host defense and pathologic inflammation, we hypothesized that a ligand-specific blockade would be superior to an unspecific inhibition of Mac-1 in polymicrobial sepsis. We first evaluated the effects of anti-M7 in a sterile systemic inflammatory response syndrome (SIRS) induced by an i.p. injection of LPS, a known TLR agonist. We applied a novel FACS gating strategy that made it possible to identify several Mac-1+ cell subsets, including granulocytes, F4/80low (monocyte-derived) and F4/80high (resident) macrophages, dendritic cells, myeloid-derived suppressor cells (MDSCs), and others (Fig. 4a, b, Supplementary Figure 7). 20 h after injection of a medium dose of LPS (20 µg) we observed an increase in total cell numbers in the peritoneal cavity (not shown) with a relative

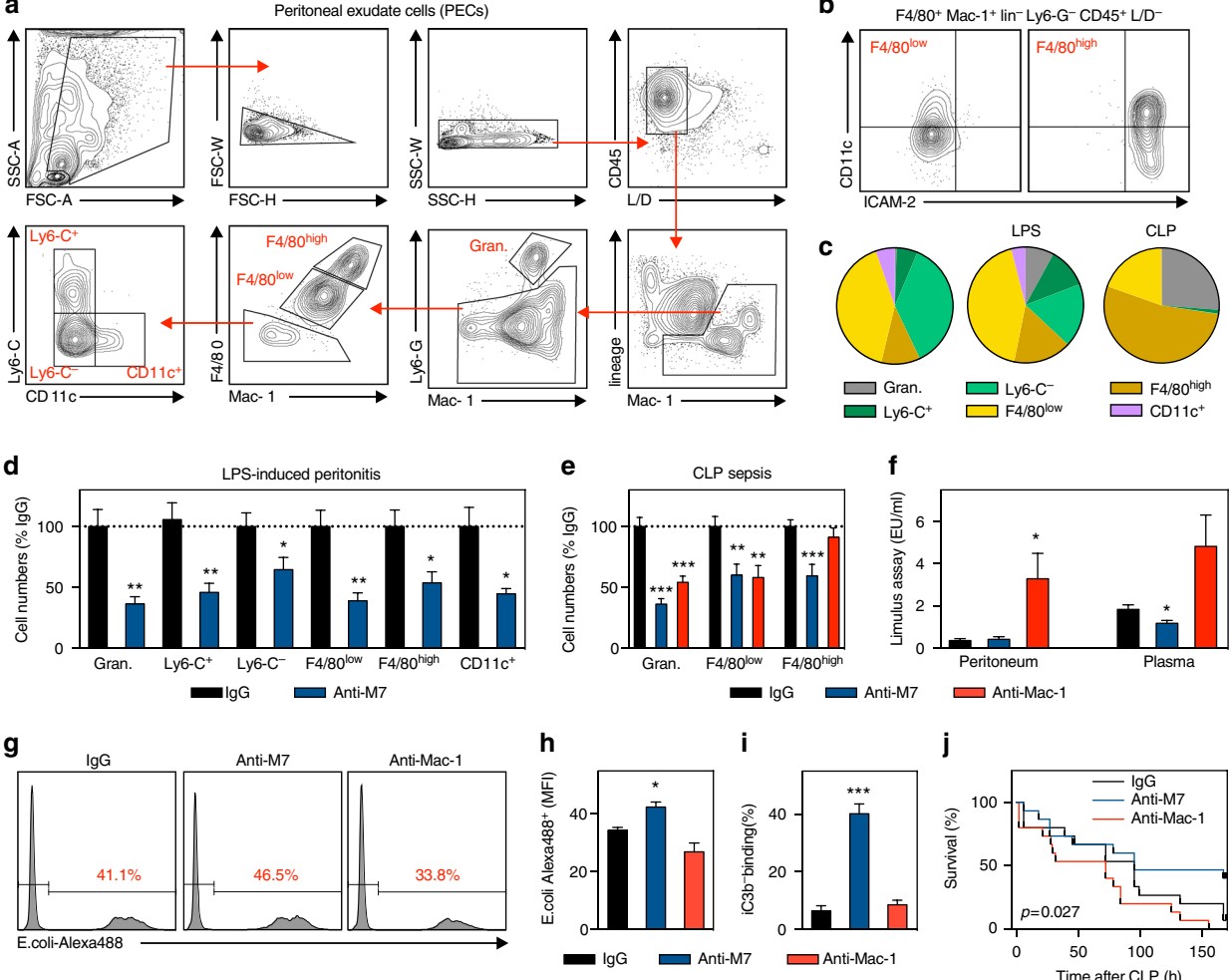

**Fig. 4** Ligand-specific Mac-1 inhibition improves sterile and bacterial sepsis by preventing excessive inflammation and enhanced bacterial clearance. A systemic inflammatory response syndrome (SIRS) induced in C57Bl/6 mice by an i.p. injection of 20 µg LPS (0111:B4). **a** Leukocyte accumulation in the peritoneal cavity was characterized with a 15-parameter panel in flow cytometry. **b** Macrophages were sub-divided into F4/80low and F4/80high. **c** Relative myeloid cell composition in the peritoneal cavity was compared in untreated mice (left cake diagram), after LPS injection (LPS), and after surgical induction of a cecal ligation and puncture (CLP) polymicrobial sepsis. **d** Male, 8-week-old C57Bl/6 mice were injected with 20 µg LPS (0111:B4) and 50 µg of the indicated antibody clones. After 20 h, myeloid cell populations were quantified in the peritoneal cavity (expressed as % of IgG-treated animals). **e** Myeloid cell populations in the peritoneal cavity 20 h after surgical induction of a CLP sepsis in male, 8-week-old C57Bl/6 mice. 1 h before surgery 50 µg of the indicated antibodies, including anti-mouse Mac-1 (M1/70), were injected. **f** Limulus assay in the peritoneal cavity and plasma to detect bacterial LPS titers after CLP. **g**, **h** Uptake of Alexa-488 labeled *E. coli* into macrophages and **i** binding of the complement factor iC3b in vitro after pre-incubation with the indicated antibody clones. **j** CLP was induced in male, 8-week-old C57Bl/6 mice. To assess whether treatment with Mac-1 antibody clones (50 µg) affects survival, mice were treated by intraperitoneal injection with either anti-Mac-1 (clone M1/70) or anti-M7 Fab preparations at −1, 48, and 96 h after induction of CLP sepsis. Relative survival was calculated and displayed as Kaplan–Maier survival curve. Error bars indicate mean ± SEM. Statistical significance was assessed by two-sided, unpaired T-test (**d**, **e**, **f**, **h**, **i**) against IgG or a Log-rank (Mantel-Cox) test (**j**). *P < 0.05, **P < 0.01, ***P < 0.001. Data are the result of N ≥ 9 mice per group (**c**–**f**), N = 5 mice per group (**h**, **i**), or N = 15 mice per group (**j**). Representative FACS plots are shown in (**a**, **b**, **g**)

increase of granulocytes (Fig. 4c). Anti-M7 efficiently blocked the infiltration of Mac-1[+] myeloid cells, in particular of granulocytes and F4/80[low] macrophages (Fig. 4d), validating its neutralizing effect on inflammatory cell recruitment. This effect was not caused by changes in the frequency of apoptosis in particular myeloid cell subsets, while an unspecific anti-Mac-1 inhibition increased the percentage of apoptotic macrophages (Supplementary Figure 8). Also, we did not detect changes in the generation of reactive oxygen species as a surrogate for myeloid cell effector function (Supplementary Figure 9) in anti-M7 treated mice.

We next performed a model of polymicrobial sepsis by a surgical cecal ligation and puncture (CLP) that causes an increase in granulocytes and macrophages in the peritoneal cavity (Fig. 4c). A preventative injection of anti-M7 dampened this increase in myeloid cell recruitment to the peritoneal cavity in 8-week-old, male C57Bl/6 mice, while an unspecific anti-Mac-1 treatment was only partially effective (Fig. 4e). The effect of anti-M7 was consistent in female mice and aged mice (40 weeks, Supplementary Figure 10). Anti-M7 failed when injected 2 h after the onset of sepsis (Supplementary Figure 11), confirming the necessity for an early intervention to reduce immediate myeloid cell infiltration[44]. Systemically, anti-M7 treatment attenuated the CLP-induced increase in inflammatory monocytosis and of the acute-phase protein SAA (Supplementary Figure 12), indicating that anti-M7 specifically reduced the excessive inflammatory response during sepsis. Accordingly, we did not find an inhibitory effect of anti-M7 on peritoneal leukocyte recruitment in a less severe CLP sepsis model, where peritoneal cell numbers did not relevantly increase over baseline (Supplementary Figure 13). Surprisingly, anti-M7 improved bacterial clearance in the plasma, while anti-Mac-1 worsened bacterial load in both, plasma and in the peritoneal cavity (Fig. 4f). This tempted us to directly test bacterial uptake, which was increased in anti-M7 treated mice, likely by favoring complement iC3b binding (Fig. 4g, Supplementary Figure 14).

Bacterial uptake and antigen-presentation is crucial to induce an antigen-specific immune response. In accordance, we detected higher numbers of activated T cells with an effector and memory phenotype, decreased systemic levels of the immune-suppressive cytokine IL-10 (Supplementary Figure 12, 15), a reduced number of immune-dampening MDSCs in the peritoneal cavity (within the Ly6-C[−] population, Fig. 4d), and increased IgM antibody levels (Supplementary Figure 12), indicating an improvement in host defense. In contrast, histology of organs vulnerable for septic injury, such as the kidney, was not affected by an anti-M7 treatment 20 h after CLP, although we detected an expected decrease in granulocyte accumulation (Supplementary Figure 16, 17). Finally, we assessed whether our ligand-specific approach ultimately affects survival during sepsis. After CLP, mice received F_ab preparations of IgG, anti-Mac-1, or anti-M7 at 1 h before, 48, and 96 h after surgery. Survival rates were calculated by Kaplan−Maier analysis and compared by log-rank testing. Anti-M7 treated showed a lethality of 60% at the end of the study compared with 93% in IgG-treated mice (log-rank test, $P = 0.027$, Fig. 4j). All animals treated with anti-Mac-1 died. These findings indicate that —in contrast to an unspecific blockade— a ligand-specific anti-integrin therapy can reduce excessive inflammation without negatively affecting host defense.

## Discussion

The adhesion receptor Mac-1 mediates a variety of inflammatory mechanisms[10]. Although Mac-1 has attraction as a target for limiting leukocyte recruitment in atherosclerosis and other acute and chronic inflammatory conditions[14,41], various undesired actions of blocking this integrin temper enthusiasm for this approach[25,54]. For example, patients with leukocyte adhesion deficiency (LAD), caused by defects of the integrins Mac-1, LFA-1, and CD11c in their common $\beta_2$-subunit, causes immune deficiency[55]. Therefore, global therapeutic inhibition of Mac-1 may have unfavorable consequences. To circumvent this limitation, we designed a novel monoclonal antibody that targets specifically the binding of CD40L to Mac-1's major ligand-binding I-domain within the $\alpha_M$-subunit of the integrin. We have previously shown that CD40L is a biased Mac-1 ligand, stimulating leukocyte recruitment by mediating binding to endothelial cell-expressed CD40L, without activating outside-in signaling[38,41]. In this previous study, peptide-based inhibition of the CD40L/Mac-1 interaction did not interfere with CD40L/CD40, Mac-1/GP1bα, or Mac-1/ICAM-1 binding, suggesting unique binding epitopes on each of the molecules[41]. Therefore, specifically targeting the CD40L-Mac-1 interaction could obviate some of the unwanted effects of global Mac-1 blockade.

The β2-integrins Mac-1 (CD11b/CD18), LFA-1 (CD11a/CD18), along with members of the β1- and β3-integrin families support leukocyte recruitment to inflamed tissue[11–13,56]. Although several studies have demonstrated that reducing myeloid cell accumulation by inhibition of the integrins α3β1 and αVβ3 decreases aberrant leukocyte infiltration in sepsis and protects from sepsis-related mortality[48–50], neutralization of Mac-1 results in a strong inhibition of leukocyte mobilization, but also causes an accelerated bacterial sepsis with a higher mortality[22–24,52]. This phenomenon is best explained by the observation that unlike α3β1 and αVβ3, Mac-1 is necessary to bind the complement factor iC3b, to opsonize bacteria, and to ultimately clear bacterial particles[22–24,52,54]. Unexpectedly, we found that binding of anti-M7 to its epitope M7 within the Mac-1 I-domain, increased iC3b binding, promoted phagocytosis, and improved adaptive T cell immunity. Currently, it remains unclear how an engagement of the epitope M7 enhances iC3b binding: The binding epitopes for iC3b within the I-domain have been located to the residues $P^{147}$-$R^{152}$, $P^{201}$-$K^{217}$, and $K^{245}$-$R^{261}$ within the $\alpha_M$ I-domain[57–59] and are therefore distinct from the epitope targeted by anti-M7 ($E^{162}$-$L^{170}$). $P^{147}$-$R^{152}$ (according peptide M1), however, is a flanking sequence of M7 (Fig. 1a). In a previous study, the contribution of M1 to CD40L binding could not be clearly excluded[41]. It therefore remains speculative whether CD40L has a partial, low-affinity interaction with M1 that is lost after inhibition of M7, and thus would make this epitope re-accessible to bind iC3b. As proposed previously, the M7 epitope may also (sterically) regulate iC3b binding due to a regulatory activity of this domain[60].

That anti-M7 specifically blocked the high-affinity interaction of Mac-1 with CD40L and therefore was selective to high-grade (pathologic), but not low-grade (beneficial), inflammation, might represent an additional mechanism by which a CD40L-specific blockade of Mac-1 could limit sepsis-related mortality. Here, we show by several in vitro and in vivo approaches that in addition to the established ligands ICAM-1[61] and RAGE[36], CD40L represents a powerful adhesive ligand that is expressed on inflamed endothelial cells[41,42]. Anti-M7 specifically blocked the interaction of Mac-1 with CD40L, but not with ICAM-1 or RAGE. Overall, anti-M7 was less effective compared to anti-Mac-1 or anti-ICAM-1 treatment. In contrast to an unspecific anti-Mac-1 treatment, which blocked adhesion and rolling of leukocytes, anti-M7 only blocked cellular adhesion. These findings are in accord with previous observations[11,32,41] and point out that the interaction of Mac-1 with one or more of its alternative ligands, but not with CD40L, may mediate leukocyte rolling in TNF-α stimulated mesenteric venules. Notably, neither anti-Mac-1, nor

anti-M7 caused a depletion of leukocytes, ruling out a reduction of the circulating leukocyte pool as potential explanation for this effect. In addition, anti-M7 lost its ability to block leukocyte recruitment in the setting of a milder CLP procedure. These results indicate that the Mac-1/CD40L interaction selectively contributes to aberrant inflammation by blocking firm adhesion and transmigration of leukocytes in the setting of acute inflammation.

The role of pro-inflammatory cytokine secretion in sepsis remains controversial. Depending on the specific context, dose, and location, these can either improve or aggravate host defense[62]. However, some therapeutic strategies to block aberrant cytokines have been demonstrated efficacy in experimental sepsis: blocking IL-10, an immuno-suppressive cytokine, improves survival during CLP[63]. Genetic inhibition of IL-3, a driver of mye-lopoiesis, reduced mortality[64]. Likewise, mice deficient for the receptor of TNF survived significantly longer than WT mice[65]. In this study, we observed that the treatment with anti-M7 decreased, while treatment with a conventional anti-Mac-1 anti-body clone (M1/70 in the mouse and 2LPM19c in humans) strongly increased systemic and local pro-inflammatory cytokine secretion. These findings confirm previous reports on isolated macrophages and likely result from outside-in signaling by non-selective antibody binding[66], an action avoided by anti-M7. The decrease in cytokines observed for anti-M7, in contrast, is likely caused by a reduction of cell infiltration and subsequent cytokine secretion by these cells[36] without an induction of outside-in-signaling. During sepsis, some cytokines released by myeloid cells or lymphocytes can further fuel myeloid cell recruitment by driving monocyte egress from the bone marrow or the spleen[64,67] or by secreting chemokines, such as MCP-1. Selective Mac-1 blockade by anti-M7 may avoid this positive feedback loop by inhibiting TNFα, MCP-1, or IL-10. In line, anti-M7 treatment attenuated inflammatory monocytosis post CLP. The decrease of IL-10 may also stem from a decrease of immune-suppressive MDSCs.

The highly promiscuous integrin Mac-1 interacts with more than 40 different ligands, for most of which we lack knowledge of the detailed molecular mechanisms of binding[13,38]. Therefore, other ligands may share the CD40L-binding site. However, the data presented here suggest that CD40L does not share many features with other Mac-1 ligands: (I) While binding epitopes identified for fibrinogen and other ligands show overlapping regions[30,33], the EQLKKSKTL motif within the Mac-1 I-domain has not been reported to be involved in binding of alternative ligands[41], (II) neither CD40L itself, nor anti-M7 induced integrin outside-in signaling, while many other ligands do[38,46], (III) CD40L's interaction with Mac-1 does not dampen immune function, while most of Mac-1 ligands, such as fibrinogen, par-ticipate in multiple pathological processes[30]. Our data shown here and previously[41] indicate that immune function, hemostasis, and regenerative responses do not involve binding of CD40L to Mac-1. The concept of such ligand/function-specificity was recently supported by the finding that inhibition of the M2 sequence required for binding of GP1bα selectively reduces the cross-talk of leukocyte and platelets in thrombosis[27].

Taken together, here, we present a novel and powerful strategy to selectively target the CD40L-binding motif on Mac-1 with the monoclonal antibody anti-M7. This antibody is selective for the targeted binding site, does not interfere with alternative binding partners, and —in contrast to conventional anti-Mac-1 anti-bodies— does not affect host defense. We propose that a ligand-targeted anti-integrin therapy may represent a promising therapy against inflammation in cardiovascular disease. Future research will have to clarify whether this concept is translatable to human disease.

## Methods

**Animal protocols**. Male mice on a C57BL/6 background received a standard chow diet. All mice were maintained under standardized conditions (12 h light, 12 h dark cycle) and had access to food and water ad libitum. At the age of 8 or 40 weeks, mice were subjected to intravital microscopy, sterile peritonitis, LPS injection, or CLP sepsis. As indicated, we additionally tested female mice when indicated in the results section. Treatment with antibodies (F$_{ab}$ fragment preparations, 50 μg/mouse) was performed by intraperitoneal or intravenous injection at a volume of 100 μL per injection. In some intravital experiments, GFP-transgenic animals under the control of CXCR3-promoter (CXCR3-GFP) or LysM (LysM-GFP), or Mac-1-deficient mice (Mac-1$^{-/-}$, B6.129S4-Itgam$^{tm1Myd}$/J, Jackson Laboratories, stock number: 003991) and aged-matched WT littermate controls were subjected to intravital microscopy.

**Study approval**. All experimental animal protocols were approved by the animal ethics committee of the Alfred Medical Research and Education Precinct (AMREP), Melbourne, Australia, the local animal ethics committee at the University of Freiburg, and at the La Jolla Institute for Allergy and Immunology, CA, USA. All procedures were carried out in accordance with institutional guidelines.

**Antibody generation**. An antibody specific for a peptide corresponding to Mac-1 I-domain sequence V160-S172 was obtained by immunizing mice with the peptide C-VMEQLKKSKTLFS-NH2 coupled to diphtheria toxoid (Monash Antibody Technologies Facility, Monash University, Melbourne, Australia). For further selection, hybridoma supernatants were screened for binding to the peptide M7 in solid-phase binding assays[41]. Among different clones with high affinity to M7, one clone (RC3), subsequently termed anti-M7, was further characterized.

**Antibody-binding studies**. Specific binding of the antibody anti-M7 to the immobilized peptides M7 (EQLKKSKTL), or the control peptides scrambled sM7 (KLSLEKQTK), or the peptide M8 (EEFRIHFT)[41] was tested in a solid-phase binding with immobilized peptides in 96-well ELISA plates (Nunc). Binding of anti-M7 was detected by addition of biotinylated anti-mouse IgG and subsequent color reaction after the incubation with HRP-coupled streptavidin and TMB-substrate. Specific binding was calculated by subtracting the binding of mouse IgG to the peptides. Alternatively, binding of anti-M7 was tested in western blot on cell lysates from native CHO cells or CHO cells stably transfected with human Mac-1. Full western blots are shown in Supplementary Figure 1. To test binding of the antibody anti-M7 to human leukocytes, anti-M7 was labeled with Alexa Fluor-647 according to the manufacturer's protocols (Monoclonal Antibody Labeling Kit, Life Technologies). Human leukocytes were isolated from healthy donors by centrifugation and Red Blood Cell lysis, and stimulated with PMA (200ng/ml), incubated with anti-M7-Alexa 647 (1 μg and 5 μg) and antibody binding was quantified by flow cytometry.

**Antibody inhibition studies**. For in vitro or in vivo use, the following antibodies have been used: Anti-mouse Mac-1 (clone M1/70, ThermoFisher, Cat.Nr. 14-0112-82), IgG2b-kappa (ThermoFisher Cat.Nr. 14-4732-82), anti-mouse ICAM-1 (clone YN1/1.7.4, Biolegend, Cat.Nr. 116101), anti-LFA-1 (clone M17/4, BioLegend, Cat.Nr. 101118), anti-mouse RAGE (clone 697023, R&D, Cat.Nr. MAB11795), anti-human Mac-1 (clone ICRF44, BioLegend, Cat.Nr. 301312), and anti-human Mac-1 (clone 2LPM19c, GeneTex, Cat.Nr. GTX72023). Antibodies were supplied as stock concentrations of 1 mg/ml are were diluted 1:100 for in vitro experiment (final concentration of 10 μg/ml). For in vivo experiments, antibodies were diluted 1:10 in sterile saline and 100μg were injected in a final volume of 100 μl i.p.

**Flow cytometry**. Peritoneal exudate cells (PECs), mouse peripheral blood samples, mouse splenocytes, or human blood samples from healthy volunteers were obtained as described. Red blood cells were lysed by a RBC-lysis buffer (eBioscience). Cells were washed multiple times in PBS and F$_c$-Receptors were blocked by an anti-mouse CD16/CD32 antibody cocktail (Tonbo) for 10 min on ice. Cells were then labeled with the indicated extracellular antibodies, and as indicated fixed in 4%PFA, permeabilized, and stained with intracellular antibodies before analysis on a flow cytometer (FACS Calibur or a FACS LSRII, BD Biosciences). Distinct leukocyte populations were identified by the gating strategies shown in Fig. 4 and Supplementary Figure 7, 15. As indicated, binding of the complement factor iC3b was quantified by an anti-iC3B antibody after 30 min of incubation with rat serum (source of iC3b). Apoptosis was quantified by binding of Annexin-V-Alexa488 and incubation in a Ca$^{2+}$ rich Annexin-V-binding buffer (BioLegend, Cat. No. 640945). Generation of reactive oxygen species (ROS) was quantified by incubation of cells with 1,2,3 Dihydrorhodamine (123 DHR, Invitrogen) for 30 min in the cell culture incubator at 37 °C. Uptake of Alexa Fluor 488-labeld E. coli (Molecular Probes, Cat. E-13231) was evaluated by co-incubation of peritoneal macrophages with bacteria in the presence of freshly prepared opsonizing reagent (Molecular Probes, Cat E-2870) for 30 min and subsequent quantification of Alexa Fluor 488$^+$ macrophages. Cytokine production in human

monocyte-derived macrophages (CD68[+]) was quantified in intracellular flow cytometry by the following antibodies: anti-human TNFα, clone Mab11, Biolegend, Cat.Nr. 502915), anti-human IL-6 (clone MQ2-13A5, Biolegend, Cat.Nr. 501114), and anti-human MCP-1 (clone 5D3-F7, Biolegend, Cat. No. 502604).

**Static and dynamic adhesion assay.** 96-well plates (Nunc) were coated with human sCD40L (Biozol, Germany) or human fibrinogen, ICAM-1, heparin, RAGE, vitronectin, JAM-C, or NIF (10 µg/ml, all from R&D Systems, USA) and incubated with CHO cells expressing constitutively activated, human Mac-1 (Mac-1 del) or CHO cells expressing the naïve, non-activated Mac-1 (Mac-1 WT) [43]. As controls, CHO cells expressing no integrin (CHO) were used. Cells were pre-incubated with blocking antibodies (10 µg/ml) for 15 min and allowed to adhere for 50 min. Adhering cells were counted after repeated washing with PBS. Alternatively, we tested the adhesion of primary mouse peritoneal macrophages to mouse CD40L (R&D Systems, USA). For dynamic adhesion assays, mouse endothelial cell cultures (HUVECs) were grown to confluency in 35 mm cell culture dishes as previously described[41], stimulated with TNFα overnight, and placed in a parallel flow chamber system (Glycotech). The number of adhering cells was quantified at the indicated shear rate in the presence of the indicated antibodies (10 µg/ml).

**F$_{ab}$ fragment preparation.** Before enzymatic digestion, antibodies were dialyzed in a SnakeSkin Dialysis Tubing 10 k MWCO against PBS overnight at 4 °C. Immobilized papain was used to prepare F$_{ab}$ fragments from the indicated antibodies according to the manufacturer's instructions (Pierce F$_{ab}$ Preparation Kit, Thermo Scientific). Briefly, F$_{ab}$ fragments were generated in the presence of 25 mM cysteine for 3 h at 37 °C, followed by purification on Protein-A Spin Columns. Purity of F$_{ab}$ fragments was validated by SDS–PAGE.

**Isolation and mouse peritoneal macrophages/sterile peritonitis.** 2 ml of 4% thioglycollate broth (Sigma) were injected into 8-week-old, male C57Bl/6 mice. A peritoneal lavage was performed after 72 h. Peritoneal exudate cells (PECs) were identified as described above.

**Differentiation of human monocyte-derived macrophages.** Peripheral blood mononuclear cells (PBMCs) were isolated from fresh human buffy coats (Transfusion Medicine, University Hospital Freiburg, Freiburg, Germany) in CPT tubes (BD Bioscience) by gradient centrifugation. PBMCs were collected, washed in RPMI-1640 (10% FCS, 5% Human serum albumin, 1% Penicillin/Streptomycin), and plated out in tissue culture dishes at a density of $2 \times 10^6$ cells/ml for 24 h. After 24 h, cells were washed with RPMI1-640 to remove non-adherent cells and cultured for another 7 days in cell culture media supplemented with 10 ng/ml M-CSF. Purity of macrophages was confirmed by expression of CD68 and Mac-1.

**Human neutrophil isolation.** Heparinized whole blood was obtained from healthy human donors after informed consent, as approved by the Institutional Review Board of the La Jolla Institute, La Jolla, USA. Neutrophils were isolated by using a sodium metrizoate/Dextran 500 density gradient (Polymorphprep). After centrifugation at 500 g for 35 min at room temperature, neutrophils were collected from the resulting intermediate layer. Neutrophils were washed with PBS without $Ca^{2+}$ and $Mg^{2+}$ twice to avoid integrin activation. Neutrophils were 95% pure, kept in 2% HAS/RPMI-1640, and were used within 4 h. Neutrophils were incubated with anti-human F$_c$-blocking antibodies (anti-CD16/CD32) for 10 min at room temperature before all experiments.

**Cell signaling studies.** Mouse peritoneal macrophages were obtained as described above. Flow cytometry revealed that the majority (>90%) of PECs were positive for the macrophage marker F4/80. After overnight starvation, macrophages were co-incubated with an anti-CD16/CD32 F$_c$-blocking antibody cocktail before addition of the indicated antibody clones against Mac-1 (10 µg/ml) for 30 min. At the indicated time points, cells were lysed, proteins were separated by SDS–PAGE and blotted to polyvinylidene difluoride membranes. Total protein and the phosphorylated fraction of NFκB, ERK1/2, and p38 were detected by specific antibody binding (Cell Signaling) in western blot. The ratio of phosphorylated fractions was calculated and expressed as relative arbitrary unit (AU) normalized to the signal of cells incubated with saline alone. Human macrophages were stimulated with anti-human antibodies for 5 h as described above before an intracellular staining for the indicated cytokines. Cells were analyzed in flow cytometry. Full western blots are shown in Supplementary Figure 5, 6.

**Integrin activation assay.** Human neutrophils were enriched by density centrifugation of peripheral blood as described above. During acquisition in flow cytometry, the anti-β2 antibodies Mab24 (detecting the β2 high-affinity conformation) and KIM127 (for β2-extension) were added at the indicated time point. 120 s later, anti-M7 or IgG (10 µg/ml) antibodies, and 360 s later human IL-8 was added during continuous acquisition over time. Recording was stopped at 600 s as previously reported[47]. The mean fluorescence intensity (MFI) for Mab24 and KIM127 was continuously recorded within the neutrophil gate and quantified as % of the starting MFI (directly after addition of Mab24 and KIM127).

**Intravital microscopy.** Quantification of leukocyte recruitment was performed as previously reported[41,68]. Mice received an intraperitoneal injection of 100 µg of whole IgG antibodies or 50µg of F$_{ab}$ fragments i.p. After 15 min, mice were injected i.p. with 200 ng murine TNFα (R&D Systems, USA) and surgery started 4 h later. Briefly, mice were anesthetized by intraperitoneal injection of ketamine hydrochloride (Essex, USA) and xylazin (Bayer, Germany). The mesentery was exteriorized and placed under an upright intravital microscope (AxioVision, Carl Zeiss, Germany). Videos of rolling and adhering in mesenteric venules were taken after retro-orbital injection of rhodamine. Rolling leukocyte flux was defined as the number of leukocytes moving at a velocity less than erythrocytes. Adherent leukocytes were defined as cells that remained stationary for at least 30 s. Intravital imaging of the greater omentum in LysM-GFP mice[69] has been previously described[44]. Briefly, after anesthesia, the omentum was exteriorized via a small median incision and carefully mounted on a coverslip. Antibodies (100µg) were injected 5 min prior to TNFα (500 ng/ml), which was applied locally to induce inflammation. 90 min later, imaging was started. Throughout the experiment, the specimen was kept moist using PBS. Anti-CD31-PE (10 µg/mouse, azide-free, clone 390) and Evans Blue was injected i.v. to visualize the circulation. Due to the high permeability in omental milky spots, Evans Blue is also apparent in the parenchyma and served as surrogate marker for endothelial leakage. A SP5 microscope (Leica) in point-scanning confocal mode equipped with a 20x/0.70 HCX PL APO objective was used for imaging z-stacks at $512 \times 512$ pixels 10 µm apart every 8 s. Post-processing included linear contrast adjustment, motion correction and cell tracking using ImageJ and Imaris (Bitplane).

**LPS-induced inflammation.** A systemic inflammatory response syndrome (SIRS) was induced in 8-week-old, male C57Bl/6 mice by an i.p. injection of 20µg LPS (0111:B4, Invivogen, USA). Simultaneously, F$_{ab}$ fragment preparations of the indicated antibodies (50 µg of IgG, anti-M7, or anti-mouse Mac-1 (clone M1/70)) were injected i.p. After 20 h, mice were killed and a peritoneal lavage was performed. Cell infiltration in the peritoneal cavity was calculated as follows: total count of cells (within the size range of 5 to 15 µm, assessed by a Beckman Coulter Z2 cell counter) x fraction of cell sub-population of all leukocytes in flow cytometry and subsequent normalization as % of cell numbers in the IgG-treated control in the same experimental cohort.

**Cecal ligation and puncture.** Surgical induction of polymicrobial sepsis was performed according to available standard protocols[70]. The peritoneal cavity of 8- or 40-week old male C57Bl/6 mice was opened during ketamine/xylazin anesthesia, and the cecum was exteriorized and ligated about 2 mm distal of the ileo-cecal valve using a non-absorbable 6–0 suture. As indicated, 8-week-old female C57Bl/6 mice were tested in addition. The distal end of the cecum was then perforated using a 23 G or a 30 G needle as indicated, and a small drop of feces was extruded through the puncture. The cecum was relocated into the peritoneal cavity, and the peritoneum was closed by a suture. Buprenorphine was applied as s.c. injection. 1 h before or 2 h after surgery F$_{ab}$ fragment preparations of the indicated antibodies (50 µg of IgG, anti-M7, or anti-mouse Mac-1 (clone M1/70)) were injected i.p. After 20 h, mice were killed and a peritoneal lavage was performed. Organs were prepared for cell isolation or histology. A plasma sample was stored at −20 °C for further analysis. Alternatively, a set of mice was followed up in survival analysis and received additional antibody injections at 48 and 96 h after surgery. Cell infiltration in the peritoneal cavity was calculated as described above.

**Analysis of murine plasma and peritoneal lavage samples.** Plasma levels of Serum Amyloid-A (SAA), the mouse analog for C-reactive protein (CRP), were measured by ELISA, according to the manufacturer's protocols (USCN Life Science). Cytokines in the plasma were determined by a cytometric bead array (CBA, BD Biosciences), according to the manufacturer's protocol. LPS titers in plasma and peritoneal lavage samples were quantified by a Limulus assay (Lonza), IgM titers by a ELISA (R&D Systems, USA).

**Statistical analysis.** Data are presented as mean ± SEM. Statistical testing employed an unpaired, two-sided Student's T-test between the indicated groups. P-values are indicated in each Figure legend. Survival was calculated using Kaplan–Meier curves and tested by log-rank test. Probability values <0.05 were considered significant.

**Data availability.** All relevant data are available from the authors upon request.

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

## Acknowledgements

This study was supported by research grants from the German Research Foundation (DFG ZI743/3-1 and 3-2) and the Else-Kröner-Fresenius Foundation (EKFSP30/10 and A43/10) to Dr. Zirlik, from the National Health and Medical Research Council (NHMRC) of Australia to Dr. Peter (#586653). Dr. Peter is a Principal Research Fellow of the NHMRC. Dr. Wolf was supported by the Wilhelm-Stoffel-Stipendium from MSD Sharpe and Dome, Germany, and the German Research Foundation (WO1994/1-1), Dr. Buscher from the German Research Foundation (BU 3247/1-1), and T. Gerhardt from the Boehringer-Ingelheim Foundation.

## Author contributions

D.W., N.A.M., H.B., A.W., K.B., J.D.H., B.L., M.B., A.M., M.M., Z.F., H.W., D.S., Z.F., H. W., D.S., T.M., M.S., B.S., and T.G. performed experiments. D.W., D.D., C.B., K.P., P.D., and A.Z. designed the study with the critical input of P.D., C.v.z.M., J.R., P.S., I.H., F.W., P.L., and K.L. D.Z. E.F.P., Z.F., and V.Y. helped with binding studies, D.W., N.A.M., P.L., K.P., and A.Z. wrote the manuscript.

## Additional information

**Competing interests:** The authors declare no competing financial interests.

