## [Peer Review File · Nature Communications]

Reviewer #1 (Remarks to the Author):

Wolf and colleagues describe in their work a newly developed antibody called M7. This ab is directed against the binding site of CD40L on the $\beta 2$ integrin Mac-1. CD40L recognizes a binding motif within the Mac-1 I-domain which has not been reported for other Mac-1 ligands and helps to explain why CD40L binding to Mac-1 leads to biological effects not observed for other Mac-1 ligands. Most prominently, binding of CD40L to Mac-1 does not induce outside-in signaling (at least for the read-outs tested here) although it affects leukocyte adhesion and extravasation. The study is of great interest, experiments are carefully conducted and the text is written in a comprehensive manner. The authors have used several in vitro and in vivo assays to test M7 revealing new functions mediated by CD40L/Mac-1 interactions. Overall, the findings of the study support the idea of using M7 (inhibition of CD40L/Mac-1 interactions) as a therapeutic strategy in patients with sepsis.

In its current version, the study is not complete. Additional experiments need to be performed to make the findings more conclusive.

- 1) Control experiments for M7 have only been conducted with additional Mac-1 antibodies. Mac-1 deficient mice (also in combination with M7) need to be used as well, at least in crucial experiments (f.e. Figs 1F, 2A, 2C, 4, 5G).
- 2) Figure 2A and C: the effect of M7 on adhesion is surprisingly strong. What about the other Mac-1 ligands ICAM-1 and RAGE and the $\beta 2$ integrin LFA-1. Additional experiments need to be conducted to clarify the contribution of ICAM-1, RAGE and also LFA-1 on adhesion.
- 3) p.6: Figures are mislabeled in the text. 4F should be 2F and 4I should be 2I.
- 4) p.7: Text on M1/70 and 2LPM19c does not match the figure. Please correct. Did the authors also test cytokine levels for 2LPM19c?
- 5) p.13, adhesion assay: please also add information on additional Mac-1 ligands.
- 6) Fig2H: use same scale, add scale bar.
- 7) Fig 2 and 5: for some of the panels units are missing for the y-axis. Please correct.
- 8) Abstract, l.3: rolling should precede adhesion.

Reviewer #2 (Remarks to the Author):

The manuscript of Wolf et. al., develops a novel antibody to an epitope on Mac-1 associated with CD40L recognition and adhesion. A significant finding is that this epitope does not induce outside-in signaling and maintains PMNs and monocytes adhering via Mac-1/CD40L in a more quiescent state. In the context of bacterial sepsis, treatment with a Fab fragment of anti-M7, silenced the exuberant innate immune response to a cecal puncture injury that normally results in sepsis and death. The impactful finding is the role of Mac-1/CD40L interaction in the induction of systemic inflammation including cytokine production associated with migration and arrest of PMN and monocytes in the microcirculation during acute inflammatory events. The novel aspect of the M7 monoclonal as a therapeutic is clearly shown in the cecal bacterial sepsis model. The major shortcoming of this manuscript is the lack of specific mechanism by which antagonism of Mac-1/CD40L provides protection in this model. While the present study demonstrated that treatment with anti-M7 or anti-Mac-1 reduced leukocyte adhesion to a similar degree, there is no data showing whether it is the blocking activity against monocytes, macrophages, or PMN that confer protection in the polymicrobial sepsis. For instance, treatment with anti-M7 could potentially confer survival of cells types critical to protection in this sepsis model. In essence, as written this study lacks mechanistic identification of how the M7 domain regulates leukocyte recruitment and inhibits their activation during bacterial sepsis.

Major Points

1. The authors focus on outside-in signaling of Mac-1 when blocking with anti-M7 and quantifying total adhesion to HUVEC cells. It would be worthwhile to perform experiments with chemokine activation to determine if Mac-1-CD40L blockade via the M7 binding domain can alter leukocyte

activation in the presence of chemokine? Can it still bind complement? This is important if the anti-M7 is to be used as a therapeutic, since should confirm that it does not compromise other innate immune functions.

2. Does anti-m7 prevent activation of Mac-1 through stabilizing a specific conformation? One possible explanation for reduction of arrest is the effect on B2-integrin affinity. Utilizing mAb24 as a reported for high affinity and KIM127 for extension of B2 integrin on human leukocytes, the effect of anti-M7 peptide on Mac-1 affinity could be determined.

3. What leukocytes are inhibited by anti-M7 during bacterial sepsis? PMN recruitment to inflamed endothelium is mediated by LFA-1, so it would be important to identify what cell types anti-M7 blocks (Fig 2).

4. One potential mechanism for anti-M7's therapeutic effect during bacteremia is its binding to and altering complement component iC3b, which is important for the opsonization of bacteria.

Minor Issues

1. The authors reference a Figure 4 on page 6 when I believe they mean Figure 2.

Reviewer #3 (Remarks to the Author):

Wolf and colleagues have conducted some sophisticated experimentation and novel work regarding the use of the monoclonal antibody, anti-M7, to specifically block M7's binding to CD40L during/after sepsis. The authors demonstrated that this antibody was able to specifically reduce only the leukocyte recruitment to the infectious source as well decreasing cytokine expression in a murine model of polymicrobial sepsis while not interfering with other immune functions. This, in turn, was responsible for a reduced mortality after cecal ligation and puncture (CLP). The statistics utilized in this work are appropriate and research described appears to be reproducible. The intravital microscopy was particularly impressive work. Although the research is novel, its interest to the scientific community is limited by several factors:

1. Murine model of sepsis: Currently, due the difficulty of translating murine sepsis to human sepsis, most individuals are moving towards animal models that better imitate the human condition. In the authors' model of CLP, they have not used crystalloid resuscitation and antibiotics for their polymicrobial sepsis work. Crystalloids and antibiotics combined with CLP are considered to be a better model to recapitulate human infection in hospitals. There are also some other models that seem to better imitate human sepsis, such as surgically removing the cecal after several days. Regardless, there are several improved models of murine sepsis in the literature that could be used to improve the capacity of this work to be translatable to human patients.

2. Aging and gender: For research regarding inflammation biology or trauma, young mice are very much acceptable for basic science work. However, sepsis is considered a disease of the elderly (or neonates), and the literature has illustrated that the aged immune system is fundamentally different than that of younger mammals. The authors should repeat their work, or at least some of the latter aspects of this research, in aged mice (18-24 month old) to strengthen their conclusions as well as help influence the current scientific thinking in this field. In addition, the NIH, as well as other national government agencies, are starting to insist that mixed gender experiments been done for animal research. Their work should included mixed gender animals instead of just male mice - trying to remove the influence of sex hormones from the results of sepsis research is no longer encouraged.

3. Timing of intervention of intraperitoneal injection of antibodies: Certainly, delivery of the antibody only some time after polymicrobial sepsis (and not at time point 0) would improve the excitement regarding this work (if it lead to improved outcomes). Multiple interventions have been demonstrated in the literature to improve outcomes in murine sepsis when given before or during the conclusion of the CLP operation, but have not lead to alterations in our understanding of human sepsis. Demonstrating the efficacy of the antibody when it is delivered only sometime after the infection has been induced would increase the translational and clinical interest of the work.

4. Myeloid-derived suppressor cells: The authors work involves the analysis of granulocytes,

macrophages, inflammatory monocytes and non-inflammatory monocytes. Myeloid-derived suppressor cells have recently been demonstrated to play a very important role in sepsis. It is unclear if the authors staining and isolation techniques do not include this immature myeloid cell population, which have been shown to significantly increase after murine and human sepsis. The authors should further evaluate the leukocytes they have analyzed to see if MDSCs make up some of the population. This would include seeing if the leukocytes they analyzed after CLP suppress T-cell function. This is particularly important as it's still unclear why exactly decreasing myeloid cell accumulation to the site of infection specifically improves outcomes (although the authors speculate in the conclusions). In addition, it's still unclear if lowering cytokine attenuation is pathologic or beneficial - lack of TNF-alpha, IL-1 or IFN-gamma can be good or bad depending on the timing and location/organ of the cytokine/chemokine post infection.

5. Sacrifice times and CLP lethality: It's unclear in the manuscript and figures when the exact mouse sacrifice times were. In addition, although significant, the improvement in mortality in the mice after CLP over the long-term is unclear in what appears to be a fairly lethal CLP model (all mice in the control group were dead prior to 7 days). Since longer term outcomes are of significant interest now in sepsis, the authors should follow the mice out for 10-14 days for survival. In addition, is the antibody as effective in a less lethal model of CLP, such as an LD40-50?

6. Organ failure: Since decreased organ failure is proposed by the authors as a potential mechanism for their improved outcomes, the authors should evaluate lung, kidney and liver histology after CLP with and without the antibody and not just leukocyte infiltration.

7. Innate Immune Function: The authors should conduct further experimentation to see how the antibody affects myeloid cell phagocytosis, generation of reactive oxygen species and T-cell activation/antigen presentation.

Minor critiques:

1. The term "fuels" in the title is somewhat of a colloquialism and should be modified to another term.
2. What did the authors mean when they stated the mice were subjected to "wound healing" in the Methods section. I believe this is the equivalent of what is known as a "Sham" operation, and if this is true the wording should be changed for the audience.
3. The authors need to include as a supplemental figure how the flow cytometry analysis was conducted, including how the FSC and SSC were used to determine the leukocyte subsets as well as the subsequent white blood cell analysis.
4. Although there is a timing aspect to what cells are present at an infectious source to induce source control, I'm surprised that most of the peritoneal exudate cells (PECs) were F4/80 positive and macrophages. Usually in abdominal sepsis and pneumonia murine models a good portion of these cells are still neutrophils at the time points the authors analyzed the PECs.
5. The authors need to report what the LD50 is in their model of CLP in their methods, or what the LD is over a set period of time (as CLP models can vary depending on how and who is performing them).

Response-to-Reviewers

We thank the Reviewers for their thorough, fair, and positive evaluation of our manuscript “A ligand-specific blockade of the integrin Mac-1 selectively targets pathologic inflammation and maintains its protective role in host-defense”. We addressed each of the Reviewers’ comments that are quoted in bold (marked as Reviewer) and are directly followed by our response (marked as Authors). **New text is marked with red.**

Reviewer 1:

Wolf and colleagues describe in their work a newly developed antibody called M7. This ab is directed against the binding site of CD40L on the b2 integrin Mac-1. CD40L recognizes a binding motif within the Mac-1 I-domain which has not been reported for other Mac-1 ligands and helps to explain why CD40L binding to Mac-1 leads to biological effects not observed for other Mac-1 ligands. Most prominently, binding of CD40L to Mac-1 does not induce outside-in signaling (at least for the read-outs tested here) although it affects leukocyte adhesion and extravasation. The study is of great interest, experiments are carefully conducted and the text is written in a comprehensive manner. The authors have used several in vitro and in vivo assays to test M7 revealing new functions mediated by CD40L/Mac-1 interactions. Overall, the findings of the study support the idea of using M7 (inhibition of CD40L/Mac-1 interactions) as a therapeutic strategy in patients with sepsis. In its current version, the study is not complete. Additional experiments need to be performed to make the findings more conclusive.

Authors: We thank the Reviewer for deeming our study of great interest. We have followed the Reviewer’s suggestions to improve our manuscript with a set of new *in vitro* and *in vivo* experiments. Please find our detailed responses below.

Comment 1: Control experiments for M7 have only been conducted with additional Mac-1 antibodies. Mac-1 deficient mice (also in combination with M7) need to be used as well, at least in crucial experiments (f.e. Figs 1F, 2A, 2C, 4, 5G).

Authors: We thank the Reviewer for this highly important point. We agree that testing Mac-1 KO may be informative to further validate the role of Mac-1 in the disease models we applied. Of note, all of the experiments encouraged by the Reviewer have been previously published, including Mac-1 KO mice challenged in intravital microscopy, adhesion experiments, as well as experimental thrombosis, peritonitis, and sepsis (Ehlers, Ustinov et al. 2003, Zirlik, Maier et al. 2007, Wolf, Hohmann et al. 2011, Liu, Han et al. 2014, Wang, Gao et al. 2017, Wolf, Bukosza et al. 2017). We cited and discussed these results in the revised manuscript. In addition, to adequately accommodate the point raised by the Reviewer we performed a series of new experiments validating our antibody-based approach in cell lines not expressing Mac-1 as well as in intravital microscopy in Mac-1 KO mice. Since the Reviewer alluded to several experiments, we address each of these point-by-point:

- Static adhesion CHO cells (Fig. 1F): Here, we show static adhesion of a Chinese Hamster ovarian (CHO) cell line stably transfected with human Mac-1 that is permanently activated to facilitate robust binding to its ligands. CHO cells per se do not express Mac-1, so that non-transfected CHO cells could be considered to be a control that is close to Mac-1 KO cells. We now show an additional set of data including static adhesion of non-transfected CHO cells (control CHO cells) in Supplemental Figure 1. In this experiment, we also included CHO cells transfected with WT Mac-1. These cells express Mac-1, but in its non-activated conformation (where the ligand binding I-domain containing the CD40L/Mac-1 binding motif is not exposed). Our new data in New Supplemental Figure 1 shows:

- Control CHO cells (not expressing Mac-1) do not adhere on CD40L. Notably, this experiment is a confirmation of our previous studies (Zirlik, Maier et al. 2007).
- CHO cells expressing WT Mac-1 show less adhesion than those expressing permanently activated Mac-1. This indicates that the interaction of CD40L and Mac-1 is driven by integrin activation, but can still occur in integrin states of lower activation. This is in accordance with newer concepts of integrin activation mechanisms (Fan, McArdle et al. 2016). Interestingly, anti-M7 preferentially blocked the interaction between activated Mac-1 and CD40L, but not interactions with non-activated Mac-1. This finding supports our concept that anti-M7 selectively enfold its action at sites of acute inflammation (such as sepsis).

- Dynamic adhesion Mac-1 on EC (Fig. 2A → New Fig. 2B): Here, we show the interaction of the mouse cell line RAW expressing Mac-1 with mouse endothelial cells in a dynamic adhesion assay. Unfortunately, these cells are not available as a variant not expressing Mac-1. In an effort to accommodate the Reviewer's request we considered gene silencing as an option, but in our opinion, this would add to much bias to this experiment. This assay is part of a stepwise approach to first demonstrate anti-M7 effects *in vitro* (static), *in vitro* (dynamic), and *in vivo*. We therefore decided to test Mac-1 KO mice directly *in vivo* (see next point) to quantify leukocyte recruitment. In previous studies, we demonstrated that CD40L binding strictly depends on Mac-1 *in vitro* and *in vivo* and that inhibition strategies were highly specific (Wolf, Hohmann et al. 2011). One technical problem with Mac-1 KO mice/cells is that a lack of Mac-1 nearly completely abolishes adhesion and cell recruitment, which is at least demonstrating the dominant effect of Mac-1 in the context of these assays. Accordingly, the lack of any relevant leukocyte adhesion will not allow to reliably assess additional effect. Leukocytes from Mac-1 KO mice were tested previously in static and dynamic adhesion assays. In these reports, adhesion of Mac-1 KO cells is almost completely blocked (Ding, Babensee et al. 1999, Dunne, Ballantyne et al. 2002).

- Intravital Microscopy (Fig. 2C → New Figure 2C-E): To adequately accommodate the Reviewer's suggestion, we challenged WT and Mac-1 KO mice in intravital microscopy in the presence and absence of blocking anti-M7. These results are now presented in revised Fig. 2G and show that adhesion is almost completely blocked in Mac-1 KO mice and in contrast to WT mice cannot be further enhanced by administration of anti-M7.

- Laser-induced thrombosis (Experiment shown in former Fig. 4): Encouraged by the Reviewer we performed mesenteric, laser-induced thrombosis in WT and Mac-1 KO mice in the presence and absence of anti-M7 or IgG. In these new experiments, we observed that (1) anti-M7 had no effect on thrombus formation in both conditions and (2) the average size of thrombi did not differ between WT and Mac-1 KO mice. These data corroborate our previous findings that anti-M7 treatment does not affect thrombosis. Since we aim to focus on inflammatory disease in this manuscript, we have decided not to show thrombosis data. For the reviewer's information only, please find our data obtained below:

- **Sepsis survival (Experiment 5):** We understand the point of the Reviewer asking for survival experiments with Mac-1 deficient mice performing CLP sepsis. However, this experiment has previously been published. In addition, this experiment is problematic since Mac-1 deficient mice afford some abnormalities in leukocyte development, apoptosis, and generation of reactive oxygen species (Coxon, Rieu et al. 1996, Lu, Smith et al. 1997). Thus, Mac-1 KO mice show aggravated inflammation, enhanced bacteremia, reduced survival in CLP sepsis, and impaired wound healing (Rosenkranz, Coxon et al. 1998, Sisco, Chao et al. 2007, Liu, Han et al. 2014). This phenotype will preclude the investigation of any specific effects on leukocyte recruitment in the sepsis model. We are convinced that a proof of specificity is much more significant on a functional basis. We previously published that blocking the CD40L/Mac-1 interaction with a peptide based strategy is not efficient to reduce peritoneal accumulation of leukocytes in sterile peritonitis in Mac-1 KO mice (Wolf, Hohmann et al. 2011). In addition, we now show data on sterile peritonitis that demonstrate that anti-M7 reduces accumulation of macrophages 72 hours after induction (Revised Figure 2, New Supplemental Figure 2). In addition, we have now discussed more thoroughly the role of Mac-1 in the context of peritoneal cell accumulation and sepsis.

All new data (except for thrombosis) have been incorporated into the revised version of the manuscript:

New text: Results part, Page 4, Paragraph 1: Anti-M7 blocked cell adhesion by $65.6 \pm 7.2\%$, an effect nearly as strong as the anti-human Mac-1 reference clone 2LPM19c, a pan I-domain blocking antibody (inhibition by $92.7 \pm 2.0\%$, Fig. 1E, F). **Notably, the adhesion of CHO cells expressing the non-permanently activated human Mac-1 (WT Mac-1) was neutralized by two conventional anti-human Mac-1 clones (2LPM19c and IRF44), but not by anti-M7, indicating that anti-M7 preferentially blocked the integrin's high-affinity interaction with CD40L. CHO cells not expressing Mac-1 did not adhere to CD40L (Supplemental Fig. 1).**

New text: Results part, Page 4, Paragraph 2: **We have recently shown that a peptide-based inhibition of the CD40L binding epitope that interacts with Mac-1 protects from leukocyte accumulation by efficiently blocking cellular adhesion (Wolf, Hohmann et al. 2011). Here, we interrogated whether blocking the M7-binding sequence directly would be equally efficient *in vitro* and *in vivo*. First, we verified that the mouse anti-human M7 antibody blocks the static adhesion of Mac-1 expressing mouse peritoneal macrophages**

and mouse CD40L (Fig. 2A). An anti-mouse Mac-1 antibody (clone M1/70) served as control.

New text: Results part, Page 4, Paragraph 2: In accordance, leukocyte rolling velocity, displayed as cumulative frequency, did not change (Fig. 2F). Notably, this effect was comparable to the effect after blockade of the receptor/ligand pairs Mac-1/ICAM-1, LFA-1/ICAM-1, and Mac-1/RAGE. Anti-M7 failed to reduce leukocyte recruitment in Mac-1^{-/-} mice (Fig. 2G), suggesting specificity of the antibody.

New text: Results part, Page 5, Paragraph 1: Finally, anti-M7 efficiently reduced the numbers of macrophages in the peritoneal cavity 72 hours after induction of a sterile, thioglycollate-induced peritonitis (Fig. 2L). In addition, anti-M7 improved, while an anti-Mac-1 treatment enhanced, the levels of pro-inflammatory cytokines in the peritoneal cavity (Supplemental Fig. 2).

New text: Discussion part, Page 8, Paragraph 2: Our data shown here and previously (Wolf, Hohmann et al. 2011) indicate that immune function, haemostasis, and regenerative responses do not involve binding of CD40L to Mac-1. The concept of such ligand/function-specificity was recently supported by the finding that inhibition of the M2 sequence required for binding of GP1b α selectively reduces the cross-talk of leukocyte and platelets and thrombosis (Wang, Gao et al. 2017).

New text: Methods part, Page 8, Paragraph 4: In some intravital experiments, GFP-transgenic animals under the control of CXCR3-promoter (CXCR3-GFP) or LysM (LysM-GFP), or Mac-1-deficient mice (Mac-1^{-/-}, B6.129S4-Itgam^{tm1Myd}/J, Jackson Laboratories, stock number: 003991) or aged-matched WT littermate controls were subjected to intravital microscopy.

New text: Methods part, Page 9, Paragraph 5: **Static and dynamic adhesion assay.** 96-well plates (Nunc) were coated with human sCD40L (Biozol, Germany) or human fibrinogen, ICAM-1, heparin, RAGE, vitronectin, JAM-C, or NIF (10 μ g/ml, all from R&D Systems, USA) and incubated with CHO-cells expressing constitutively activated, human Mac-1 (Mac-1 del) or CHO-cells expressing the naïve, non-activated Mac-1 (Mac-1 WT) (Schuler, Assefa et al. 2003). As controls, CHO cells expressing no integrin (CHO) were used. Cells were pre-incubated with blocking antibodies (10 μ g/ml) for 15 minutes and allowed to adhere for 50 minutes. Adhering cells were counted after repeated washing with PBS. Alternatively, we tested the adhesion of primary mouse peritoneal macrophages to mouse CD40L (R&D Systems, USA).

New text: Figure Legends, Page 20: **Figure 2: Treatment with anti-M7 prevents inflammatory leukocyte recruitment *in vitro* and *in vivo*.** (A) Murine peritoneal macrophages were incubated on dishes coated with mouse CD40L in the presence of 10 μ g/ml of IgG, anti-M7, or anti-Mac-1 (clone M1/70). Adhering cells were normalized to % of the IgG-treatment. (B) Murine RAW-cell adhesion on TNF α -primed murine endothelial cells under physiological flow in the presence of IgG or anti-M7 antibodies (10 μ g/ml). (C-F) C57Bl/6 mice were injected i.p. with 200ng TNF α and 15min before microscopy with F_{ab}-fragments of an IgG control antibody, or antibodies against RAGE, ICAM-1, LFA-1, Mac-1 (clone M1/70), or anti-M7 (100 μ g i.p.). Leukocyte recruitment was monitored by intravital microscopy 4 hours later: Adhering (D) and rolling leukocytes (E) were quantified as % of IgG. (F) Cumulative frequency of leukocyte rolling velocity. (G) Leukocyte adhesion in intravital microscopy in IgG or anti-M7 treated WT and Mac-1^{-/-} mice as described above. (H, I) Number of CX3CR1-GFP⁺ monocytes (white arrows) in

the para-vascular space 4 hours after treatment with IgG or anti-M7 and 200ng TNF α . (J) Confocal *in vivo* imaging of the greater omentum after local stimulation with TNF α and i.v. injection of the indicated antibodies (100 μ g) in LysM-GFP reporter mice. LysM-GFP⁺ myeloid cells were tracked over time. Time lapse and cell tracking is shown over 15min (right panel). (K) Plasma cytokine levels of mice subjected to intravital microscopy after IgG or anti-M7 F_{ab} treatment. (L) Peritoneal exudate cells (PECs) 72 hours after induction of a sterile peritonitis and the indicated antibody treatment. Scale bars represent 100 μ m (C, H, J). Error bars indicate mean \pm SEM. Statistical significance was assessed by a two-sided, unpaired Student's T-test between the indicated groups (A, B, G, I, K, L) or in comparison to IgG-treatment (D, E). * P<0.05, ** P<0.01, *** P<0.001, **** P<0.0001. N \geq 3 independent experiments (A, B). N \geq 10 mice per group (D, E, F, G), N \geq 6 mice per group (L, H).

New text, Supplement, Page 2: **Supplemental Figure 1: Anti-M7 specifically blocks the interaction of the activated, open-conformation integrin Mac-1 with immobilized CD40L.** Human sCD40L (CD40L) was coated on 96-well plates (10 μ g/ml), blocked with 1%BSA/PBS, and incubated with CHO-cells expressing constitutively activated, human Mac-1 (Mac-1 del CHO) (Schuler, Assefa et al. 2003), CHO cells expressing the human wildtype integrin (Mac-1 WT CHO), or naïve CHO cells that do not express Mac-1 (control CHO). As a control for CD40L, plates were coated with 1% BSA/PBS (BSA). Cells were pre-incubated with blocking antibodies (10 μ g/ml) against the CD40L-binding site (anti-M7) or against the entire Mac-1 ligand binding I-domain (anti-human Mac-1). Two blocking anti-human Mac-1 antibody clones were tested: 2LPM19c and ICRF44. These clones do not cross-react with mouse Mac-1. Cells were allowed to adhere for 50 minutes. Adhering cells were counted after repeated washing with PBS and were normalized to % of IgG control (Mac-1 del CHO adhesion on CD40L). Error bars indicate mean \pm SEM. Statistical significance was tested for the following groups by an unpaired, two-sided Student's T-test: IgG antibody treatment against other antibody treatments of the same CHO-cell type. In addition, significance was tested between control IgG-treated naïve CHO cells and control IgG-treated Mac-1 del CHO or control IgG-treated Mac-1 WT CHO (indicated by brackets), * P < 0.05, ** P < 0.01, *** P < 0.001, **** P < 0.0001. N \geq 4 per group of three individual experiments.

New text, Supplement, Page 3: **Supplemental Figure 2: Anti-M7 prevents the accumulation of macrophages and dampens inflammatory gene expression in the peritoneal cavity during sterile peritonitis.** 8-week-old, male C57Bl/6J mice were injected with 2ml 4% thioglycollate broth i.p. to induce a sterile peritonitis with an accumulation of inflammatory cells in the peritoneal cavity. Simultaneously, 50 μ g of F_{ab}-fragment preparations of the following antibodies were injected i.p.: Unspecific IgG isotype-control, anti-M7, or conventional anti-mouse Mac-1 (anti-mouse clone M1/70). After 72 hours, peritoneal cells were collected with a peritoneal lavage and characterized by flow cytometry. Macrophages were identified as F4/80⁺Ly-6G⁻ viable CD45⁺ leukocytes. Macrophage content was expressed as percentage of all viable leukocytes (A). Concentrations of the cytokines TNF α (B) and MCP-1 (C) were quantified by a cytometric bead array (CBA). Error bars indicate mean \pm SEM. Significance was assessed by an unpaired, two-sided Student's T-test between the indicated conditions. N \geq 6 mice per group.

Comment 2: Figure 2A and C: the effect of M7 on adhesion is surprisingly strong. What about the other Mac-1 ligands ICAM-1 and RAGE and the β 2 integrin LFA-1. Additional

experiments need to be conducted to clarify the contribution of ICAM-1, RAGE and also LFA-1 on adhesion.

Authors: We agree with the Reviewer that the relative contribution of CD40L in the context of other adhesion factors was not clear in the initial version of the manuscript. To accommodate this important point of the Reviewer we now tested inhibition of the proposed ligands (ICAM-1, RAGE, LFA-1) with antibodies in intravital microscopy. These results are now included in revised Fig. 2 C-E. Herein, we show that all treatments range in between 20-50% inhibition with anti-ICAM-1 being the strongest inhibitor and anti-M7 being the weakest. Our data are in accord with previously published work and demonstrate that anti-M7 has an effect that is weaker than a complete anti-Mac-1 blockade, suggesting that the residual interaction of Mac-1 with ICAM-1 and RAGE is sufficient to maintain some leukocyte influx.

New text: Results part, Page 4, Paragraph 2: To test the antibody's *in vivo* applicability, we prepared F_{ab}-fragments of anti-M7, anti-Mac-1 (M1/70), a control-IgG, and antibodies directed against ICAM-1, LFA-1 (CD11/CD18), and RAGE to avoid unspecific F_c-receptor interactions. In intravital microscopy (Fig. 2C) of inflamed mesenteric venules, the number of adhering (Fig. 2D) but not of rolling leukocytes (Fig. 2E) fell after anti-M7 injection. In accordance, leukocyte rolling velocity, displayed as cumulative frequency, did not change (Fig. 2F). Notably, this effect was comparable to the effect after blockade of the receptor/ligand pairs Mac-1/ICAM-1, LFA-1/ICAM-1, and Mac-1/RAGE. Anti-M7 failed to reduce leukocyte recruitment in Mac-1^{-/-} mice (Fig. 2G), suggesting specificity of the antibody.

New text: Discussion part, Page 7, Paragraph 3: Here, we show by several *in vitro* and *in vivo* approaches that in addition to the established ligands ICAM-1 (Diamond, Staunton et al. 1990) and RAGE (Chavakis, Bierhaus et al. 2003), CD40L represents a powerful adhesive ligand that is expressed on inflamed endothelial cells (Wolf, Hohmann et al. 2011, Michel, Zirlik et al. 2017). Anti-M7 specifically blocked the interaction of Mac-1 with CD40L, but not with ICAM-1 or RAGE. Overall, anti-M7 was less effective compared to anti-Mac-1 or anti-ICAM-1 treatment.

New text: Figure Legends, Page 20: **Figure 2: Treatment with anti-M7 prevents inflammatory leukocyte recruitment *in vitro* and *in vivo*.** ... C57Bl/6 mice were injected i.p. with 200ng TNF α and 15min before microscopy with F_{ab}-fragments of an IgG control antibody, or antibodies against RAGE, ICAM-1, LFA-1, Mac-1 (clone M1/70), or anti-M7 (100 μ g i.p.). Leukocyte recruitment was monitored by intravital microscopy 4 hours later: Adhering (**D**) and rolling leukocytes (**E**) were quantified as % of IgG. (**F**) Cumulative frequency of leukocyte rolling velocity. (**G**) Leukocyte adhesion in intravital microscopy in IgG or anti-M7 treated WT and Mac-1^{-/-} mice. Error bars indicate mean \pm SEM. Statistical significance was assessed by a two-sided, unpaired Student's T-test between the indicated groups (**A**, **B**, **G**, **I**, **K**, **L**) or in comparison to IgG-treatment (**D**, **E**). * P<0.05, ** P<0.01, *** P<0.001, **** P<0.0001. N \geq 3 independent experiments (**A**, **B**). N \geq 10 mice per group (**D**, **E**, **F**, **G**), N \geq 6 mice per group (**L**, **H**).

Comment 3: p.6: Figures are mislabeled in the text. 4F should be 2F and 4I should be 2I.

Authors: We apologize for this. As we have completely re-organized the manuscript, all Figures are now correctly labeled.

Comment 4: p.7: Text on M1/70 and 2LPM19c does not match the figure. Please correct. Did the authors also test cytokine levels for 2LP M19c?

Authors: We apologize for this mistake. Unfortunately, we had switched 2LPM19c and M1/70 in the text. We have now corrected the text to match the figure.

As to 2LPM19c: In the previous version of the manuscript, the anti-human Mac-1 clone 2LPM19c that was used a species-mismatch (negative) control for mouse macrophages (Figure 3A, B). However, we did not use this control in the *in vivo* experiments (Fig. 2C). As suggested by the Reviewer, we now included a new *in vitro* experiment testing the effect of 2LPM19c (Fig. 3D), in which we stimulated human monocyte-derived macrophages with anti-human Mac-1 (2LPM19c) followed by intracellular flow cytometry. Intracellular cytokines were up-regulated in the same manner as mouse macrophages were. We hope these results help to clarify the effect of this antibody clone.

New text: Results part, Page 5, Paragraph 2: **Our observation that a treatment with anti-Mac-1 protected from recruitment to the peritoneal cavity (Fig. 2L) while enhancing pro-inflammatory cytokine expression (Supplemental Fig. 2), tempted us to verify potential outside-in signaling events promoted by anti-M7 and anti-Mac-1. Therefore, we collected thioglycollate-elicited peritoneal macrophages from male, 8-week-old C57Bl/6 mice and incubated these with 10 μ g/ml of either mouse IgG, anti-human Mac-1 (clone 2LPM19c) as species-mismatch control, anti-mouse Mac-1 (clone M1/70), or anti-M7 for 30min. Anti-Mac-1 treatment (M1/70) induced phosphorylation of ERK and p38 as quantified by an elevated ratio of the phosphorylated proteins in Western Blotting (Fig. 3A,B). Species-mismatched anti-Mac-1 (2LPM19c) or anti-M7 showed no activation of these MAP-kinases, indicating that the binding epitope M7 targeted by anti-M7 is not involved in outside-in signaling. To assess the *in vivo* relevance of such anti-Mac-1 agonism, mice received anti-Mac-1 antibodies from various clones i.p. and serum concentration of IL-6, TNF α , and MCP-1 were quantified 4 hours after injection. The anti-Mac-1 reference clone in the mouse, M1/70, strongly elevated cytokine levels, while anti-M7 did not (Fig. 3C). Pro-inflammatory cytokine concentrations also increased in cultured mouse macrophages after antibody co-incubation (data not shown). Vice versa, human monocyte-derived macrophages showed the same pattern in anti-M7 and anti-human Mac-1 (reference clone 2LPM19c) stimulated macrophages (Fig. 3D).**

New text: Discussion part, Page 8, Paragraph 1: **In this study, we observed that the treatment with anti-M7 decreased while treatment with a conventional anti-Mac-1 antibody clone (M1/70 in the mouse and 2LPM19c in humans) strongly increased systemic and local pro-inflammatory cytokine secretion. These findings confirm previous reports on isolated macrophages and likely result from outside-in signaling by non-selective antibody binding (Ding, Wright et al. 1987), an action avoided by anti-M7. The decrease in cytokines observed for anti-M7, in contrast, is likely caused by a reduction of cell infiltration and subsequent cytokine secretion by these cells³⁶ without an induction of outside-in-signaling.**

New text: Figure Legends, Page 21: **Figure 3: Conventional anti-Mac-1 therapy, but not a ligand-specific inhibition by anti-M7, is a potent inducer of integrin outside-in signaling and aberrant cytokine secretion. (A, B) Mouse macrophages were isolated from the peritoneal cavity of C57Bl/6 mice 72 hours after injection of 4% thioglycollate. Peritoneal cells were collected by lavage and the purity was confirmed by flow cytometry (>90% F4/80⁺ macrophages). Cells were cultured in 5%FCS/RPMI overnight and co-incubated with 10 μ g/ml of mouse IgG, anti-human Mac-1 (clone 2LPM19c), anti-mouse Mac-1 (clone M1/70), or anti-M7 for 30min in the presence of an anti-mouse CD16/CD32 F_c-block. Cells were lysed and total and phosphorylated ERK1/2 and p38 were visualized by Western blot and the ratio of phosphorylated fractions was calculated. Values were calculated as relative arbitrary units (AU) normalized to signal of cells stimulated with**

saline alone. **(C)** Saline, or IgG, anti-M7, anti-Mac-1 (clone M1/70) antibodies were injected i.p. in mice and plasma concentrations of IL-6, TNF α , and MCP-1 were measured by cytometric bead array 4 hours after injection. **(D)** Human macrophages were differentiated from peripheral monocytes and co-incubated with the indicated antibody clones including the anti-human Mac-1 clone 2LPM19c in the presence of an anti-human CD16/CD32 Fc-block for 5 hours. Cytokines were quantified by intracellular flow cytometry in CD68⁺ macrophages. **(E)** Human neutrophils were enriched by density centrifugation of peripheral blood. **(F-I)** The anti- β 2 antibodies Mab24 (detecting the β 2 high-affinity conformation) and KIM127 (for β 2-extension) and anti-M7 or IgG (10 μ g/ml) were added during acquisition in flow cytometry. The mean fluorescence intensity (MFI) for Mab24 **(F, G)** and KIM127 **(H, I)** was continuously recorded over 10min within the neutrophil gate **(E)** and quantified as % of the starting MFI **(G, I)**. Error bars indicate mean \pm SEM. Statistical significance was assessed by a two-sided, unpaired Student's T-test between the indicated groups. Data are the result of N \geq 3 independent experiments **(A, B)**. N \geq 9 mice per group **(C)** N \geq 6 human donors **(D)**. N=3 human donors **(F-I)**. A representative FACS plot is shown in **(E)**.

Comment 5: p.13, adhesion assay: please also add information on additional Mac-1 ligands.

Authors: Following the Reviewer's suggestion we added the required information on these Mac-1 ligands in the revised methods section.

New text: Methods part, Page 9, Paragraph 5: Static and dynamic adhesion assay. 96-well plates (Nunc) were coated with human sCD40L (Biozol, Germany) or human fibrinogen, ICAM-1, heparin, RAGE, vitronectin, JAM-C, or NIF (10 μ g/ml, all from R&D Systems, USA) and incubated with CHO-cells expressing constitutively activated, human Mac-1 (Mac-1 del) or CHO-cells expressing the naïve, non-activated Mac-1 (Mac-1 WT) (Schuler, Assefa et al. 2003). As controls, CHO cells expressing no integrin (CHO) were used. Cells were pre-incubated with blocking antibodies (10 μ g/ml) for 15 minutes and allowed to adhere for 50 minutes. Adhering cells were counted after repeated washing with PBS. Alternatively, we tested the adhesion of primary mouse peritoneal macrophages to mouse CD40L (R&D Systems, USA).

Comment 6: Fig2H: use same scale, add scale bar.

Authors: We apologize for the confusion and consequently now added the same scale bar and included more representative pictures from intravital microscopy experiments. The scale bars for all pictures have been adjusted to 100 μ m.

Comment 7: Fig 2 and 5: for some of the panels units are missing for the y-axis. Please correct.

Authors: We have added missing y-axis labels to all figures.

Comment 8: Abstract, I.3: rolling should precede adhesion.

Authors: We thank the Reviewer for this comment. We have reworded the abstract accordingly.

New text: Summary, Page 2: Integrin based therapeutics have garnered considerable interest in the medical treatment of inflammation. Integrins mediate the fast recruitment of monocytes and neutrophils to the site of inflammation, but are also required for host defense, limiting their therapeutic use. Here, we report a novel monoclonal antibody, anti-M7, that specifically blocks the interaction of the integrin Mac-1 with its pro-inflammatory ligand CD40L while not interfering with alternative ligands. Anti-M7 selectively reduced

leukocyte recruitment *in vitro* and *in vivo*. In contrast, conventional anti-Mac-1 therapy was not specific and blocked a broad repertoire of integrin functionality, inhibited phagocytosis, promoted apoptosis, and fueled a cytokine storm *in vivo*. As a result, conventional anti-integrin therapy potentiated bacterial sepsis, bacteremia, and mortality, while a ligand-specific intervention with anti-M7 was protective. These findings deepen our understanding of ligand-specific integrin functions and open a path for a new field of ligand-targeted anti-integrin therapy to prevent inflammatory conditions.

Reviewer 2:

The manuscript of Wolf et. al., develops a novel antibody to an epitope on Mac-1 associated with CD40L recognition and adhesion. A significant finding is that this epitope does not induce outside-in signaling and maintains PMNs and monocytes adhering via Mac-1/CD40L in a more quiescent state. In the context of bacterial sepsis, treatment with a Fab fragment of anti-M7, silenced the exuberant innate immune response to a cecal puncture injury that normally results in sepsis and death. The impactful finding is the role of Mac-1/CD40L interaction in the induction of systemic inflammation including cytokine production associated with migration and arrest of PMN and monocytes in the microcirculation during acute inflammatory events. The novel aspect of the M7 monoclonal as a therapeutic is clearly shown in the cecal bacterial sepsis model. The major shortcoming of this manuscript is the lack of specific mechanism by which antagonism of Mac-1/CD40L provides protection in this model. While the present study demonstrated that treatment with anti-M7 or anti-Mac-1 reduced leukocyte adhesion to a similar degree, there is no data showing whether it is the blocking activity against monocytes, macrophages, or PMN that confer protection in the polymicrobial sepsis. For instance, treatment with anti-M7 could potentially confer survival of cells types critical to protection in this sepsis model. In essence, as written this study lacks mechanistic identification of how the M7 domain regulates leukocyte recruitment and inhibits their activation during bacterial sepsis.

Authors: We thank the Reviewer for considering our findings “impactful, clearly demonstrated, and novel”. We agree with the Reviewer that the specific mechanism by which anti-M7 protects from sepsis was not sufficiently explained by the data provided in the previous version of the manuscript. To accommodate this point, we performed extensive new experiments as encouraged by the Reviewers. This helped uncovering several novel aspects that considerably improved our work and will hopefully help to better understand the protective effects of anti-M7. We summarize these new findings in the following paragraphs and included these data into the revised version of our manuscript:

Why did anti-M7 protect?

1. Immune defense in abdominal sepsis is delicate. A too intense engagement of the innate immune system will result in overwhelming inflammation with a self-perpetuating cytokine storm that limits survival. These cytokines partially stem from leukocytes recruited to the peritoneal cavity. On the other side, a complete inhibition of the innate immune response will greatly affect host defense and impair survival, too (Chousterman, Swirski et al. 2017). In contrast to anti-Mac-1 therapy, anti-M7 did not completely block leukocyte recruitment into the peritoneal cavity. In particular, during low-grade CLP (new data, Supplemental Fig. 9) anti-M7 had no measurable effect. Mechanistically, we could show that only the interaction of the high affinity integrin (on highly activated leukocytes) was inhibited by anti-M7 (new data, Supplemental Fig. 1). Also, anti-M7-treated leukocytes will still interact with alternative ligands, such as ICAM-1 and RAGE. As a result, a basic level of innate immune defense is maintained. In contrast, anti-Mac-1 blocks independently of integrin

- conformation/activation and neutralizes all Mac-1/ligand interactions, which will result in an almost complete inhibition and block the innate immune response to a great extent.
2. Anti-M7 unexpectedly increased iC3b binding and bacterial phagocytosis (new data, Revised Figure 4 G-I, New Supplemental Fig. 10. This may be explained by competitive binding sites for CD40L and iC3b, induction of neo-epitopes on the integrin (i.e. ligand-induced binding epitopes), or a regulatory function of the M7 sequence (Zhang and Plow 1996). As a result, phagocytosis was even improved in anti-M7 treated mice.
 3. Besides classical pro-inflammatory leukocytes, anti-M7 strongly reduced a population of cells called Myeloid-derived Suppressor Cells (MDSCs) that was previously reported to regulate sepsis by modulation of adaptive and innate immunity. Accordingly, we observed enhanced T cell activation, lowered levels of IL-10, and increased IgM levels in anti-M7 treated animals (new data, Supplemental Fig. 3, 8, 11). While such immune-activating effects could also be secondary to an increased bacterial uptake, antigen-processing, and activation of immune cells, our results clearly indicate a more 'active' immune system.

Why did anti-Mac-1 not protect?

1. Anti-Mac-1 treatment induced a storm of pro-inflammatory cytokines, such as $\text{TNF}\alpha$, IL-6, and MCP-1. These cytokines have been associated with a poor outcome in sepsis (Kellum, Kong et al. 2007, Chousterman, Swirski et al. 2017). We previously showed a similar pattern in adipose tissue macrophages after anti-Mac-1 treatment (Wolf, Bukosza et al. 2017).
2. Anti-Mac-1 treatment inhibited phagocytosis of bacteria (new data, Main Figure 4) as previously reported (Liu, Han et al. 2014). This not only affects bacterial clearance, but also dampens subsequent (beneficial) traits of adaptive immunity.
3. Activation of leukocytes by outside-in signaling (induced by antibody binding) potentiated leukocyte apoptosis without significant changes in ROS generation (a substrate required for killing bacteria) (new data, Supplemental Fig. 4, 5). As a result, leukocyte infiltration was not only limited by unspecifically blocking cell adhesion, but also enhanced leukocyte death after anti-Mac-1 treatment. Both will negatively affect host defense.

What is the mechanism underlying anti-M7-dependent regulation of leukocyte recruitment?

1. We previously showed that CD40L is an endothelial ligand for Mac-1 that is expressed on activated endothelial cells in the context of inflammation (Wolf, Hohmann et al. 2011, Michel, Zirlik et al. 2017).
2. We previously tracked down the binding site for CD40L on Mac-1 by peptide-based and genetic tools. The M7 peptide sequence represents the distinct binding site for CD40L (Wolf, Hohmann et al. 2011).
3. Here, we succeeded to specifically block this binding site by a monoclonal antibody. Blocking by anti-M7 results in a selective inability to bind to CD40L and to adhere to activated endothelium and to transmigrate as known for ICAM-1 and RAGE (Chavakis, Bierhaus et al. 2003, Ley, Laudanna et al. 2007).
4. In addition, anti-M7 may have (some) regulatory activity by slightly reducing integrin activation (new data, Revised Fig. 3); this potentially affects binding affinity to some other endothelial ligands and therefore contributes to the decrease in leukocyte recruitment.

Please find our detailed answer and the according text changes in the revised version of the manuscript below.

Did anti-M7 specifically confer survival of some myeloid cells?

We thank the Reviewer for this important question. We have conducted new experiments to clarify this question and quantified Annexin-V as read-out for cellular apoptosis. We found that anti-M7 did not promote survival of myeloid cells or its sub-populations (New data, New Supplemental

Fig. 4). However, we observed a tendency towards less apoptosis in granulocytes ($P=0.13$ IgG vs. anti-M7). Conversely, anti-Mac-1 treatment significantly increased apoptosis in Macrophages (likely caused by the intrinsic 'over-stimulation' by outside-in signaling). These data confirm previous findings (Wolf, Hohmann et al. 2011). In addition, apoptosis was lower in peritoneal B cells following anti-M7, indicating that the improved immune phenotype after anti-M7 directly affected B cell survival.

New text, Supplement, Page 5: Supplemental Figure 4: Treatment with an unspecific anti-Mac-1 antibody, but not with anti-M7, induces apoptosis and cell death in peritoneal macrophages and B cells. Peritoneal cells were collected from untreated, male 8-week-old C57Bl/6J mice by a peritoneal lavage. 400.000 leukocytes were transferred into a 96-well plate in 50 μ l complete RPMI cell culture media containing 1:125 cell stimulation cocktail with PMA and Ionomycin (eBioscience) per well. The indicated antibodies (anti-mouse clone M1/70 as anti-Mac-1) were added to the wells at a final concentration of 20 μ g/ml. After 4 hours, cells were stained for flow cytometry with an antibody cocktail containing Annexin-V-FITC and a live/dead dye (L/D). Apoptotic cells were identified as L/D⁻Annexin-V⁺ and quantified as % of all cells (**A**). Representative FACS plots of macrophage Annexin-V binding are shown in (**B**). Error bars indicate mean \pm SEM. Significance was assessed by an unpaired, two-sided Student's T-test. * $P < 0.05$, ** $P < 0.01$, *** $P < 0.001$, **** $P < 0.0001$. N=5 donor mice per group. n.s.: not significant.

We now show and discuss these novel findings in the revised version of the manuscript. We hope the revised manuscript sufficiently addresses the Reviewer's concerns.

Comment 1: The authors focus on outside-in signaling of Mac-1 when blocking with anti-M7 and quantifying total adhesion to HUVEC cells. It would be worthwhile to perform experiments with chemokine activation to determine if Mac-1-CD40L blockade via the M7 binding domain can alter leukocyte activation in the presence of chemokine?

Authors: This is a very interesting point brought up by the Reviewer. As stated correctly, full activation of leukocytes occurs in the presence of chemokines thereby promoting firm adhesion and transmigration. We therefore tested a leukocyte-activation assay based on real-time flow cytometry (Fan and Ley 2015, Fan, McArdle et al. 2016). A central readout in this assay is the measurement of the binding of the two antibodies Mab24 and KIM127 that indicate conformational changes (high affinity and extension of the beta-subunit of the integrin) and thus indicate leukocyte activation. Using this assay with human density gradient-isolated neutrophils and the chemokine IL-8 (functional human homologue for KC/CXCL1) we could show that (1) anti-M7 without IL-8 at baseline does not alter integrin activation/conformation, (2) anti-M7 only slightly reduced leukocyte activation after IL-8 stimulation. These new data are now shown in revised Figure 3. In conclusion, we assume that the epitope employed by anti-M7 does not relevantly modulate leukocyte activation. In accord with these observations, we previously showed that CD40L represents a biased agonist for Mac-1 not provoking outside-in signaling (Simon 2011, Wolf, Hohmann et al. 2011).

New text: Results part, Page 5, Paragraph 2: To further verify that anti-M7 does not act as an agonist for Mac-1, we quantified binding of the two anti-human β 2-subunit antibodies KIM127 and Mab24 indicating cellular activation and subsequent changes in integrin conformation by detecting β 2-extension (KIM127) and β 2 high-affinity conformation (Mab24) (Fan, McArdle et al. 2016), in human primary neutrophils in a real-time flow cytometry binding assay (Fig. 3E-I). Only binding of Mab24 was slightly reduced after co-incubation with the human chemokine IL-8, which is required for full integrin activation, and anti-M7, while we observed no changes in the baseline activation. These findings

suggest that anti-M7 targets an epitope on Mac-1 that does not cause outside-in signaling and pro-inflammatory cytokine secretion in Mac-1 expressing cells during anti-integrin therapy.

New text: Discussion part, Page 8, Paragraph 1: In this study, we observed that the treatment with anti-M7 decreased, while treatment with a conventional anti-Mac-1 antibody clone (M1/70 in the mouse and 2LPM19c in humans) strongly increased systemic and local pro-inflammatory cytokine secretion. These findings confirm previous reports on isolated macrophages and likely result from outside-in signaling by non-selective antibody binding (Ding, Wright et al. 1987), an action avoided by anti-M7.

New text: Methods part, Page 10, Paragraph 5: **Human neutrophil isolation.** Heparinized whole blood was obtained from healthy human donors after informed consent, as approved by the Institutional Review Board of the La Jolla Institute, La Jolla, USA. Neutrophils were isolated by using a sodium metrizoate/Dextran 500 density gradient (Polymorphprep). After centrifugation at 500g for 35 minutes at room temperature, neutrophils were collected from the resulting intermediate layer. Neutrophils were washed with PBS without Ca^{2+} and Mg^{2+} twice to avoid integrin activation. Neutrophils were 95% pure, kept in 2% HAS/RPMI-1640, and were used within 4 hours. Neutrophils were incubated with anti-human F_c -blocking antibodies (anti-CD16/CD32) for 10 minutes at room temperature before all experiments.

New text: Methods part, Page 10, Paragraph 7: **Integrin activation assay.** Human neutrophils were enriched by density centrifugation of peripheral blood as described above. During acquisition in flow cytometry, the anti- $\beta 2$ antibodies Mab24 (detecting the $\beta 2$ high-affinity conformation) and KIM127 (for $\beta 2$ -extension) were added at the indicated time point. 120 seconds later, anti-M7 or IgG (10 $\mu\text{g}/\text{ml}$) antibodies, and 360 seconds later human IL-8 was added during continuous acquisition over time. Recording was stopped at 600 seconds as previously reported (Fan, McArdle et al. 2016). The mean fluorescence intensity (MFI) for Mab24 and KIM127 was continuously recorded within the neutrophil gate and quantified as % of the starting MFI (directly after addition of Mab24 and KIM127).

New text: Figure Legends, Page 21: **Figure 3: Conventional anti-Mac-1 therapy, but not a ligand-specific inhibition by anti-M7, is a potent inducer of integrin outside-in signaling and aberrant cytokine secretion.** (A, B) Mouse macrophages were isolated from the peritoneal cavity of C57Bl/6 mice 72 hours after injection of 4% thioglycollate. Peritoneal cells were collected by lavage and the purity was confirmed by flow cytometry (>90% F4/80⁺ macrophages). Cells were cultured in 5%FCS/RPMI overnight and co-incubated with 10 $\mu\text{g}/\text{ml}$ of mouse IgG, anti-human Mac-1 (clone 2LPM19c), anti-mouse Mac-1 (clone M1/70), or anti-M7 for 30min in the presence of an anti-mouse CD16/CD32 F_c -block. Cells were lysed and total and phosphorylated ERK1/2 and p38 were visualized by Western blot and the ratio of phosphorylated fractions was calculated. Values were calculated as relative arbitrary units (AU) normalized to signal of cells stimulated with saline alone. (C) Saline, or IgG, anti-M7, anti-Mac-1 (clone M1/70) antibodies were injected i.p. in mice and plasma concentrations of IL-6, $\text{TNF}\alpha$, and MCP-1 were measured by cytometric bead array 4 hours after injection. (D) Human macrophages were differentiated from peripheral monocytes and co-incubated with the indicated antibody clones including the anti-human Mac-1 clone 2LPM19c in the presence of an anti-human CD16/CD32 F_c -block for 5 hours. Cytokines were quantified by intracellular flow cytometry in CD68⁺ macrophages. (E) Human neutrophils were enriched by density centrifugation of peripheral blood. (F-I) The anti- $\beta 2$ antibodies Mab24 (detecting the $\beta 2$ high-affinity

conformation) and KIM127 (for β 2-extension) and anti-M7 or IgG (10 μ g/ml) were added during acquisition in flow cytometry. The mean fluorescence intensity (MFI) for Mab24 (F, G) and KIM127 (H, I) was continuously recorded over 10min within the neutrophil gate (E) and quantified as % of the starting MFI (G, I). Error bars indicate mean \pm SEM. Statistical significance was assessed by a two-sided, unpaired Student's T-test between the indicated groups. Data are the result of N \geq 3 independent experiments (A, B). N \geq 9 mice per group (C) N \geq 6 human donors (D). N=3 human donors (F-I). A representative FACS plot is shown in (E).

Comment 2: Can it still bind compliment? This is important if the anti-M7 is to be used as a therapeutic, since should confirm that it does not compromise other innate immune functions.

Authors: This is a very important consideration in the context of sepsis raised by the Reviewer. We now quantified binding of iC3b to peritoneal macrophages after *in vitro* blockade of the CD40L binding site with anti-M7. Unexpectedly, we found that anti-M7 treatment enhanced binding of iC3b (Revised Fig. 4 I). This suggests that blocking the CD40L interaction favors binding of iC3b. In the previous version of the manuscript we showed that bacteremia was improved after anti-M7 treatment, but impaired after unspecific anti-Mac-1 blockade as shown before (Fossati-Jimack, Ling et al. 2013, Liu, Han et al. 2014). Consistently, we now observed that phagocytosis of Alexa-488 labeled bacteria increased after anti-M7, but decreased after anti-Mac-1 treatment (Revised Fig. 4 G, H). We now discuss several potential causes for this phenomenon in the revised version of the manuscript, including:

1. The M7 sequence in Mac-1 mediates most of the binding to CD40L (as shown by the loss of binding after anti-M7), but additional binding epitopes (on Mac-1) cannot be excluded entirely. For instance, we previously showed that the M1 peptide binding epitope also contributes to some extent to CD40L binding, although effects of M1 peptide inhibition were not consistent in each assay and smaller than these of M7 (Wolf, Hohmann et al. 2011). Interestingly, M1 is also a reported binding epitope for iC3b (Ueda, Rieu et al. 1994, Zhang and Plow 1999, Li and Zhang 2003). Given the excess of circulating sCD40L in the context of inflammation and sepsis it is to speculate whether CD40L would cover the M1 binding epitope. Blocking with anti-M7 would thus inactivate the major binding epitope on Mac-1 resulting in a total loss of CD40L binding and unveil additional binding sites via M1 that would now be accessible for instance to iC3b.
2. The M7 sequence (EQLKSKTL) has previously been suggested to enfold a regulatory role that may inhibit binding of other ligands (Zhang and Plow 1996). This suggestion was based on mutant Mac-1 swapping mutants, where the M7 sequence was replaced by the corresponding part of LFA-1 (CD11a). Such mutant Mac-1 showed increased interaction with iC3b. Although such chimeric proteins are highly artificial models, whose relevance for WT proteins is questionable, it may be that inhibition with anti-M7 may at least partially exhibit an agonistic/steric effect that specifically enhances iC3b binding. It is noteworthy that such effect would have not affected overall integrin activity/conformation (please see response to the reviewer's next point).

We now show and discuss these new data in the revised version of the manuscript.

New text: Results part, Page 6, Paragraph 2: Surprisingly, anti-M7 improved bacterial clearance in the plasma, while anti-Mac-1 worsened bacterial load in both, plasma and in the peritoneal cavity (Fig. 4F). This tempted us to directly test bacterial uptake, which was increased in anti-M7 treated mice, likely by favoring complement iC3b binding (Fig. 4G, Supplemental Fig. 10).

New text: Discussion part, Page 7, Paragraph 3: The β 2-integrins Mac-1 (CD11b/CD18), LFA-1 (CD11a/CD18), along with members of the β 1- and β 3-integrin family support leukocyte recruitment to inflamed tissue^{11, 12, 13, 56}. Although several studies have demonstrated that reducing myeloid cell accumulation by inhibition of the integrins α 3 β 1 and α V β 3 decreases aberrant leukocyte infiltration in sepsis and protects from sepsis-related mortality (Sarangi, Hyun et al. 2012, Lerman, Lim et al. 2014, Chen, Ding et al. 2016), neutralization of Mac-1 results in a strong inhibition of leukocyte mobilization, but also causes an accelerated bacterial sepsis with a higher mortality^{22, 23, 24, 52}. This phenomenon is best explained by the observation that unlike α 3 β 1 and α V β 3, Mac-1 is necessary to bind the complement factor iC3b, to opsonize bacteria, and to ultimately clear bacterial particles (Liu, Han et al. 2014)^{22, 23, 24, 52}. Unexpectedly, we found that binding of anti-M7 to its epitope M7 within the Mac-1 I-domain, increased iC3b binding, promoted phagocytosis, and improved adaptive T cell immunity. Currently, it remains unclear an engagement of the epitope M7 enhances iC3b binding. The binding epitopes for iC3b within the I-domain have been located to the residues P¹⁴⁷-R¹⁵², P²⁰¹-K²¹⁷, and K²⁴⁵-R²⁶¹ within the α _M I-domain (Ueda, Rieu et al. 1994, Zhang and Plow 1999, Li and Zhang 2003) and are therefore distinct from the epitope targeted by anti-M7 (E¹⁶²-L¹⁷⁰). P¹⁴⁷-R¹⁵² (according peptide M1), however, is a flanking sequence of M7 (Fig. 1A). In a previous study, the contribution of M1 to CD40L binding could not be clearly excluded (Wolf, Hohmann et al. 2011). It therefore remains speculative whether CD40L has a partial, low-affinity interaction with M1 that is lost after inhibition of M7, and thus re-accessible to iC3b binding. As proposed before, the M7 epitope may also be required to (sterically) regulate iC3b binding due to a regulatory activity of this domain (Zhang and Plow 1996).

New text: Methods part, Page 9, Paragraph 4: As indicated, binding of the complement factor iC3b was quantified by an anti-iC3B antibody after 30min incubation with rat serum (source of iC3b). Apoptosis was quantified by binding of Annexin-V-Alexa488 and incubation in a Ca²⁺ rich Annexin-V-binding buffer (eBioscience). Generation of reactive-oxygen species (ROS) was quantified by incubation of cells with 1,2,3 Dihydrorhodamine (123 DHR, Invitrogen) for 30 minutes in the cell culture incubator at 37°C. Uptake of Alexa Fluor 488-labeled E. coli (Molecular Probes, Cat. E-13231) was evaluated by co-incubation of peritoneal macrophages with bacteria in the presence of freshly prepared opsonizing reagent (Molecular Probes, Cat E-2870) for 30 minutes and subsequent quantification of Alexa Fluor 488⁺ macrophages.

New text: Figure Legends part, Page 22: Ligand-specific Mac-1 inhibition improves sterile and bacterial sepsis by preventing excessive inflammation and enhanced bacterial clearance. A systemic inflammatory response syndrome (SIRS) induced in C57Bl/6 mice by an i.p. injection of 20 μ g LPS (0111:B4). **(A)** Leukocyte accumulation in the peritoneal cavity was characterized with a 15-parameter panel in flow cytometry. **(B)** Macrophages were sub-divided into F4/80^{low} and F4/80^{high}. **(C)** Relative myeloid cell composition in the peritoneal cavity was compared in untreated mice (left cake diagram), after LPS-injection (LPS), and after surgical induction of a cecal-ligation and puncture (CLP) polymicrobial sepsis. **(D)** Male, 8-week-old C57Bl/6 mice were injected with 20 μ g LPS (0111:B4) and 50 μ g of the indicated antibody clones. After 20 hours, myeloid cell populations were quantified in the peritoneal cavity (expressed as % of IgG treated animals). **(E)** Myeloid cell populations in the peritoneal cavity 20 hours after surgical induction of a CLP sepsis in male, 8-week-old C57Bl/6 mice. 1 hour before surgery 50 μ g of the indicated antibodies, including anti-mouse Mac-1 (M1/70), were injected. **(F)** Limulus assay in the peritoneal cavity and plasma to detect bacterial LPS titers after CLP. **(G, H)** Uptake of Alexa-488 labeled E. coli into macrophages and **(I)** binding of the

complement factor iC3b *in vitro* after pre-incubation with the indicated antibody clones. (J) Cecal-ligation and puncture sepsis (CLP) was induced in male, 8-week-old C57Bl/6 mice. To assess whether treatment with Mac-1 antibody clones (50µg) affects survival, mice were treated by intraperitoneal injection with either anti-Mac-1 (clone M1/70) or anti-M7 F_{ab} preparations at -1, 48, and 96 hours after induction of CLP sepsis. Relative survival was calculated and displayed as Kaplan-Maier survival curve. Error bars indicate mean ± SEM. Statistical significance was assessed by two-sided, unpaired T-test (D, E, F, H, I) against IgG or a Log-rank (Mantel-Cox) test (J). * P<0.05, ** P<0.01, *** P<0.001. Data are the result of N≥9 mice per group (C-F), N=5 mice per group (H, I), or N=15 mice per group (J). Representative FACS plots are shown in (A, B, G).

Comment 3: Does anti-m7 prevent activation of Mac-1 through stabilizing a specific conformation? One possible explanation for reduction of arrest is the effect on B2-integrin affinity. Utilizing mAb24 as a reported for high affinity and KIM127 for extension of B2 integrin on human leukocytes, the effect of anti-M7 peptide on Mac-1 affinity could be determined.

Authors: This is a highly important point. We have performed the assay suggested by the Reviewer (please also see response to comment 1) and detected no difference in the conformation of Mac-1 at baseline (without addition of IL-8) between anti-M7 and IgG treated human neutrophils. We therefore conclude that anti-M7 does not change integrin conformation.

New text: Results part, Page 5, Paragraph 2: To further verify that anti-M7 does not act as an agonist for Mac-1, we quantified binding of the two anti-human β2-subunit antibodies KIM127 and Mab24, which are indicating cellular activation and subsequent changes in integrin conformation by detecting β2-extension (KIM127) and β2 high-affinity conformation (Mab24)(Fan, McArdle et al. 2016), in human primary neutrophils in a real-time flow cytometry binding assay (Fig. 3E-I). Only binding of Mab24 was slightly reduced after co-incubation with the human chemokine IL-8, which is required for full integrin activation, and anti-M7, while we observed no changes in the baseline activation. These findings indicate that anti-M7 targets an epitope on Mac-1 that does not cause outside-in signaling and pro-inflammatory cytokine secretion in Mac-1 expressing cells during anti-integrin therapy.

New text: Discussion part, Page 8, Paragraph 1: In this study, we observed that a treatment with anti-M7 decreased, while treatment with a conventional anti-Mac-1 antibody clone (M1/70 in the mouse and 2LPM19c in humans) strongly increased systemic and local pro-inflammatory cytokine secretion. These findings confirm previous reports on isolated macrophages and likely result from outside-in signaling by non-selective antibody binding (Ding, Wright et al. 1987), an action avoided by anti-M7.

New text: Methods part, Page 10, Paragraph 5: Human neutrophil isolation. Heparinized whole blood was obtained from healthy human donors after informed consent, as approved by the Institutional Review Board of the La Jolla Institute, La Jolla, USA. Neutrophils were isolated by using a sodium metrizoate/Dextran 500 density gradient (Polymorphprep). After centrifugation at 500g for 35 minutes at room temperature, neutrophils were collected from the resulting intermediate layer. Neutrophils were washed with PBS without Ca²⁺ and Mg²⁺ twice to avoid integrin activation. Neutrophils were 95% pure, kept in 2% HAS/RPMI-1640, and were used within 4 hours. Neutrophils were incubated with anti-human F_c-blocking antibodies (anti-CD16/CD32) for 10 minutes at room temperature before all experiments.

New text: Methods part, Page 10, Paragraph 7: **Integrin activation assay.** Human neutrophils were enriched by density centrifugation of peripheral blood as described above. During acquisition in flow cytometry, the anti- $\beta 2$ antibodies Mab24 (detecting the $\beta 2$ high-affinity conformation) and KIM127 (for $\beta 2$ -extension) were added at the indicated time point. 120 seconds later, anti-M7 or IgG (10 μ g/ml) antibodies, and 360 seconds later human IL-8 was added during continuous acquisition over time. Recording was stopped at 600 seconds as previously reported (Fan, McArdle et al. 2016). The mean fluorescence intensity (MFI) for Mab24 and KIM127 was continuously recorded within the neutrophil gate and quantified as % of the starting MFI (directly after addition of Mab24 and KIM127).

New text: Figure Legends, Page 21: **Figure 3: Conventional anti-Mac-1 therapy, but not a ligand-specific inhibition by anti-M7, is a potent inducer of integrin outside-in signaling and aberrant cytokine secretion.** (A, B) Mouse macrophages were isolated from the peritoneal cavity of C57Bl/6 mice 72 hours after injection of 4% thioglycollate. Peritoneal cells were collected by lavage and the purity was confirmed by flow cytometry (>90% F4/80⁺ macrophages). Cells were cultured in 5%FCS/RPMI overnight and co-incubated with 10 μ g/ml of mouse IgG, anti-human Mac-1 (clone 2LPM19c), anti-mouse Mac-1 (clone M1/70), or anti-M7 for 30min in the presence of an anti-mouse CD16/CD32 Fc-block. Cells were lysed and total and phosphorylated ERK1/2 and p38 were visualized by Western blot and the ratio of phosphorylated fractions was calculated. Values were calculated as relative arbitrary units (AU) normalized to signal of cells stimulated with saline alone. (C) Saline, or IgG, anti-M7, anti-Mac-1 (clone M1/70) antibodies were injected i.p. in mice and plasma concentrations of IL-6, TNF α , and MCP-1 were measured by cytometric bead array 4 hours after injection. (D) Human macrophages were differentiated from peripheral monocytes and co-incubated with the indicated antibody clones including the anti-human Mac-1 clone 2LPM19c in the presence of an anti-human CD16/CD32 Fc-block for 5 hours. Cytokines were quantified by intracellular flow cytometry in CD68⁺ macrophages. (E) Human neutrophils were enriched by density centrifugation of peripheral blood. (F-I) The anti- $\beta 2$ antibodies Mab24 (detecting the $\beta 2$ high-affinity conformation) and KIM127 (for $\beta 2$ -extension) and anti-M7 or IgG (10 μ g/ml) were added during acquisition in flow cytometry. The mean fluorescence intensity (MFI) for Mab24 (F, G) and KIM127 (H, I) was continuously recorded over 10min within the neutrophil gate (E) and quantified as % of the starting MFI (G, I). Error bars indicate mean \pm SEM. Statistical significance was assessed by a two-sided, unpaired Student's T-test between the indicated groups. Data are the result of N \geq 3 independent experiments (A, B). N \geq 9 mice per group (C) N \geq 6 human donors (D). N=3 human donors (F-I). A representative FACS plot is shown in (E).

Comment 4: What leukocytes are inhibited by anti-M7 during bacterial sepsis? PMN recruitment to inflamed endothelium is mediated by LFA-1, so it would be important to identify what cell types anti-M7 blocks (Fig 2).

Authors: This is indeed a crucial point raised by the Reviewer to understand how exactly anti-M7 works. In the previous version of the manuscript we have only shown quantification of granulocyte, although a broad range of myeloid cells (monocytes, macrophages, myeloid-derived DCs) express Mac-1 and could be affected by Mac-1 blockade. Following the Reviewer's suggestion we performed a series of new experiments: We now quantified different leukocyte subsets in the peritoneal cavity after peritonitis (Supplemental Figure 2), LPS-induced peritonitis (new data, revised Fig. 4 A-D), and bacterial CLP sepsis (Fig. 4 E). We observed that numbers of all myeloid cell subsets were reduced in comparison to an IgG control in a range of 36 to 64 % inhibition (Revised Fig. 4 D, E). Interestingly, granulocytes (64% inhibition), inflammatory monocytes (54% inhibition), and F4/80⁺ macrophages (51% inhibition) were reduced to a higher extend than

patrolling monocytes (36% inhibition). Mac-1 has been previously reported to facilitate the recruitment of myeloid cells in acute and chronic inflammation (Rosetti and Mayadas 2016, Michel, Zirlik et al. 2017). Of note, the recruitment of Mac-1⁻ Lin⁺ lymphocytes was not affected by anti-M7 treatment (New Supplemental Figure 7).

The Reviewer is certainly right about the role of LFA-1, which we not aim to discredit with our study. In contrast, we add an important role for Mac-1 in the context of peritoneal/sepsis-related leukocyte recruitment. This idea, however, is not entirely new: For instance, we previously reported on the requirement of Mac-1, but not of LFA-1, in rapid leukocyte recruitment through the greater omentum (the main entry site for peritoneal leukocyte accumulation) by transmigrating through high endothelial venules (Buscher, Wang et al. 2016), suggesting that Mac-1 is particularly important for peritoneal cell accumulation. Also, it is known that Mac-1 and LFA-1 have differential roles in directing cells to different tissue, which is likely to be regulated by differential expression of ligands on distinct endothelial cells or tissues (Kolaczkowska and Kubes 2013). In addition, Mac-1 expression intensity and activation may differ between myeloid sub-population. For instance, patrolling monocytes are mostly LFA-1-dependent, while inflammatory Ly6-C⁺ monocytes are more depending on Mac-1 (Auffray, Fogg et al. 2007, Quintar, McArdle et al. 2017). In accordance, we see higher inhibition of inflammatory, than of patrolling monocytes. Yet, it remains unknown whether the modulation of myeloid cells that are either Mac-1^{low} expressers or known to be independent of Mac-1 (patrolling monocytes) is a primary effect of anti-M7 blockade or secondary to lowered inflammation (as result of anti-M7).

We hope our new results help to clarify the Reviewer's question. We are now showing and discussing these new results in the revised version of our manuscript.

New text: Results part, Page 5, Paragraph 1: Finally, anti-M7 efficiently reduced the numbers of macrophages in the peritoneal cavity 72 hours after induction of a sterile, thioglycollate-induced peritonitis (Fig. 2L). In addition, anti-M7 improved, while an anti-Mac-1 treatment enhanced, the levels of pro-inflammatory cytokines in the peritoneal cavity (Supplemental Fig. 2). These results indicate that leukocyte adhesion, transmigration, and accumulation proceeds *in vitro* and *in vivo* via binding of CD40L with Mac-1 – an interaction that anti-M7 effectively blocked.

New text: Results part, Page 6, Paragraph 1: Aberrant activation of leukocytes and excessive cytokine secretion increases disease severity and mortality in sepsis (Sarangi, Hyun et al. 2012, Lerman, Lim et al. 2014, Chen, Ding et al. 2016). On the other hand, integrins are required for host defense, demonstrated by the requirement for Mac-1 to clear bacteria (Natanson, Hoffman et al. 1994, Rosenkranz, Coxon et al. 1998, Liu, Han et al. 2014, Jawhara, Pluskota et al. 2017). Clinically, blood concentrations of soluble CD40L (Lorente, Martin et al. 2011) and sCD18 (Kragstrup, Juul-Madsen et al. 2017) are associated with sepsis, suggesting that both, Mac-1 and CD40L, participate in the acute inflammatory response in sepsis. To outweigh its conflicting effects on host defense and pathologic inflammation, we hypothesized that a ligand-specific blockade would be superior to an unspecific inhibition of Mac-1 in polymicrobial sepsis. We first evaluated the effects of anti-M7 in a sterile systemic inflammatory response syndrome (SIRS) induced by an i.p. injection of LPS, a known TLR agonist. We applied a novel FACS gating strategy that made it possible to identify several Mac-1⁺ cell subsets, including granulocytes, F4/80^{low} (monocyte-derived) and F4/80^{high} (resident) macrophages, dendritic cells, myeloid-derived suppressor cells (MDSCs), and others (Fig. 4A, B, Supplemental Fig. 3). 20 hours after injection of a medium dose of LPS (20µg) we observed an increase in total cell numbers in the peritoneal cavity (not shown) with a relative increase of granulocytes (Fig. 4C). Anti-M7 efficiently blocked the infiltration of Mac-1⁺ myeloid cells, in particular of granulocytes and F4/80^{low} macrophages (Fig. 4D), validating its neutralizing effect on inflammatory cell recruitment. This effect was not caused by changes in the frequency of

apoptosis in particular myeloid cell subsets, while an unspecific anti-Mac-1 inhibition increased the percentage of apoptotic macrophages (Supplemental Fig. 4). Also, we did not detect changes in the generation of reactive-oxygen-species as a surrogate for myeloid cell effector function (Supplemental Fig. 5) in anti-M7 treated mice.

We next performed a model of polymicrobial sepsis by a surgical cecal-ligation and puncture (CLP), that causes an increase in granulocytes and macrophages in the peritoneal cavity (Fig. 4C).

New text: Discussion part, Page 7, Paragraph 3: The β 2-integrins Mac-1 (CD11b/CD18), LFA-1 (CD11a/CD18), along with members of the β 1- and β 3-integrin family support leukocyte recruitment to inflamed tissue^{11, 12, 13, 56}. Although several studies have demonstrated that reducing myeloid cell accumulation by inhibition of the integrins α 3 β 1 and α V β 3 decreases aberrant leukocyte infiltration in sepsis and protects from sepsis-related mortality (Sarangi, Hyun et al. 2012, Lerman, Lim et al. 2014, Chen, Ding et al. 2016), neutralization of Mac-1 results in a strong inhibition of leukocyte mobilization, but also causes an accelerated bacterial sepsis with a higher mortality^{22, 23, 24, 52}.

New text: Methods part, Page 9, Paragraph 4: **Flow cytometry.** Peritoneal exudate cells (PECs), mouse peripheral blood samples, mouse splenocytes, or human blood samples from healthy volunteers were obtained as described. Red blood cells were lysed by a RBC-lysis buffer (eBioscience). Cells were washed multiple times in PBS and Fc γ -Receptors were blocked by an anti-mouse CD16/CD32 antibody cocktail (Tonbo) for 10 minutes on ice. Cells were then labeled with the indicated extracellular antibodies, and as indicated fixed in 4%PFA, permeabilized, and stained with intracellular antibodies before analysis on a flow cytometer (FACS Calibur or a FACS LSRII, BD Biosciences). Distinct leukocyte populations were identified by the gating strategies shown in Fig. 4 and Supplemental Fig. 3, 11.

Comment 5: One potential mechanism for anti-M7's therapeutic effect during bacteremia is its binding to and altering complement component iC3b, which is important for the opsonization of bacteria.

Authors: Here, we like to refer to our answer to comment 2, which should address the current reviewer question as well. Taken together, by testing direct binding to Mac-1 (aka CR3: 'complement receptor 3'), we found that anti-M7 enhances binding to iC3b. Currently, we can only hypothesize about the reason. However, we hope this finding can assure the Reviewer that anti-M7 does not (negatively) interfere with complement binding and host defense. In contrast, recent work has shown that a lack of Mac-1 (on a genetic level) increases bacteremia (Liu, Han et al. 2014), a result that is in accordance with increased bacteremia after anti-Mac-1 treatment in our study (Revised Fig.4 F). Please also see our response to Reviewer 1, comment 2.

New text: Results part, Page 6, Paragraph 2: Surprisingly, anti-M7 improved bacterial clearance in the plasma, while anti-Mac-1 worsened bacterial load in both, plasma and in the peritoneal cavity (Fig. 4F). This tempted us to directly test bacterial uptake, which was increased in anti-M7 treated mice, likely by favoring complement iC3b binding (Fig. 4G, Supplemental Fig. 10).

New text: Discussion part, Page 7, Paragraph 3: The β 2-integrins Mac-1 (CD11b/CD18), LFA-1 (CD11a/CD18), along with members of the β 1- and β 3-integrin family support leukocyte recruitment to inflamed tissue^{11, 12, 13, 56}. Although several studies have demonstrated that reducing myeloid cell accumulation by inhibition of the integrins α 3 β 1 and α V β 3 decreases aberrant leukocyte infiltration in sepsis and protects from sepsis-

related mortality (Sarangi, Hyun et al. 2012, Lerman, Lim et al. 2014, Chen, Ding et al. 2016), neutralization of Mac-1 results in a strong inhibition of leukocyte mobilization, but also causes an accelerated bacterial sepsis with a higher mortality^{22, 23, 24, 52}. This phenomenon is best explained by the observation that unlike $\alpha 3\beta 1$ and $\alpha V\beta 3$, Mac-1 is necessary to bind the complement factor iC3b, to opsonize bacteria, and to ultimately clear bacterial particles (Liu, Han et al. 2014)^{22, 23, 24, 52}. Unexpectedly, we found that binding of anti-M7 to its epitope M7 within the Mac-1 I-domain, increased iC3b binding, promoted phagocytosis, and improved adaptive T cell immunity. Currently, it remains unclear an engagement of the epitope M7 enhances iC3b binding. The binding epitopes for iC3b within the I-domain have been located to the residues P¹⁴⁷-R¹⁵², P²⁰¹-K²¹⁷, and K²⁴⁵-R²⁶¹ within the α_M I-domain (Ueda, Rieu et al. 1994, Zhang and Plow 1999, Li and Zhang 2003) and are therefore distinct from the epitope targeted by anti-M7 (E¹⁶²-L¹⁷⁰). P¹⁴⁷-R¹⁵² (according peptide M1), however, is a flanking sequence of M7 (Fig. 1A). In a previous study, the contribution of M1 to CD40L binding could not be clearly excluded (Wolf, Hohmann et al. 2011). It therefore remains speculative whether CD40L has a partial, low-affinity interaction with M1 that is lost after inhibition of M7, and thus re-accessible to iC3b binding. That levels of sCD40L are particularly increased in inflammation, including septic patients, has been shown before (Lorente, Martin et al. 2011). As proposed previously, the M7 epitope may also be required to (sterically) regulate iC3b binding due to a regulatory activity of this domain (Zhang and Plow 1996).

New text: Methods part, Page 9, Paragraph 4: As indicated, binding of the complement factor iC3b was quantified by an anti-iC3B antibody after 30min incubation with rat serum (source of iC3b). Apoptosis was quantified by binding of Annexin-V-Alexa488 and incubation in a Ca²⁺ rich Annexin-V-binding buffer (eBioscience). Generation of reactive-oxygen species (ROS) was quantified by incubation of cells with 1,2,3 Dihydrorhodamine (123 DHR, Invitrogen) for 30 minutes in the cell culture incubator at 37°C. Uptake of Alexa Fluor 488-labeled E. coli (Molecular Probes, Cat. E-13231) was evaluated by co-incubation of peritoneal macrophages with bacteria in the presence of freshly prepared opsonizing reagent (Molecular Probes, Cat E-2870) for 30 minutes and subsequent quantification of Alexa Fluor 488⁺ macrophages.

New text: Figure Legends, Page 22: **Figure 4: Ligand-specific Mac-1 inhibition improves sterile and bacterial sepsis by preventing excessive inflammation and enhanced bacterial clearance.** A systemic inflammatory response syndrome (SIRS) induced in C57Bl/6 mice by an i.p. injection of 20 μ g LPS (0111:B4). **(A)** Leukocyte accumulation in the peritoneal cavity was characterized with a 15-parameter panel in flow cytometry. ... **(F)** Limulus assay in the peritoneal cavity and plasma to detect bacterial LPS titers after CLP. **(G, H)** Uptake of Alexa-488 labeled E. coli into macrophages and **(I)** binding of the complement factor iC3b *in vitro* after pre-incubation with the indicated antibody clones. ... Error bars indicate mean \pm SEM. Statistical significance was assessed by two-sided, unpaired T-test **(D, E, F, H, I)** against IgG or a Log-rank (Mantel-Cox) test **(J)**. * P<0.05, ** P<0.01, *** P<0.001. Data are the result of N \geq 9 mice per group **(C-F)**, N=5 mice per group **(H, I)**, or N=15 mice per group **(J)**. Representative FACS plots are shown in **(A, B, G)**.

Comment 6: The authors reference a Figure 4 on page 6 when I believe they mean Figure 2.

Authors: We apologize for this mishap, which we have corrected in the revised version of the manuscript.

Reviewer 3:

Wolf and colleagues have conducted some sophisticated experimentation and novel work regarding the use of the monoclonal antibody, anti-M7, to specifically block M7's binding to CD40L during/after sepsis. The authors demonstrated that this antibody was able to specifically reduce only the leukocyte recruitment to the infectious source as well decreasing cytokine expression in a murine model of polymicrobial sepsis while not interfering with other immune functions. This, in turn, was responsible for a reduced mortality after cecal ligation and puncture (CLP). The statistics utilized in this work are appropriate and research described appears to be reproducible. The intravital microscopy was particularly impressive work. Although the research is novel, it's interest to the scientific community is limited by several factors.

Authors: We thank the Reviewer for deeming our work “sophisticated, novel, and impressive”. We understand that the previous version of our manuscript was limited by some important points highlighted by the Reviewer’s extremely helpful comments. As encouraged by Reviewer 3, we have now included in the revised version of the manuscript:

- several new animal studies, including testing of aged mice, both genders
- a milder sepsis model (CLP with a smaller needle size and LPS-induced sepsis)
- a detailed characterization of leukocyte subsets (including MDSCs and T cell activation markers)
- organ histology
- evaluation of phagocytosis
- complement binding assays
- validation of ROS generation

We believe that these extensive new data adequately address the Reviewer’s important and insightful concerns. Please see below for our detailed answer to the Reviewer’s helpful critique.

Comment 1: Murine model of sepsis: Currently, due the difficulty of translating murine sepsis to human sepsis, most individuals are moving towards animal models that better imitate the human condition. In the authors' model of CLP, they have not used crystalloid resuscitation and antibiotics for their polymicrobial sepsis work. Crystalloids and antibiotics combined with CLP are considered to be a better model to recapitulate human infection in hospitals. There are also some other models that seem to better imitate human sepsis, such as surgically removing the cecal after several days. Regardless, there are several improved models of murine sepsis in the literature that could be used to improve the capacity of this work to be translatable to human patients.

Authors: As clinician scientists, we appreciate the Reviewer’s important comment. We agree that human bacterial ‘sepsis’ cannot be classified easily due to the many different causes, manifestations, and co-morbidities. We also believe – again in agreement with the Reviewer – that mouse models often do not reflect human sepsis. In particular in the context of adaptive immunity (as brought up later by the Reviewer in his later comments), mouse models come with limitations in the quest to understand human disease. The intention of this work is, however, not to established a new therapy in sepsis but primarily to describe an experimental proof-of-concept work with a novel ligand-specific integrin inhibitor that may be superior to unspecific integrin blockade. The centerpiece of our study is the characterization of the antibody anti-M7 that specifically blocks cellular adhesion without interfering with beneficial functions of the integrin (Fig 1-3, 4 A-D). Only the last part of this work (Fig. 4), where the antibody is tested in a CLP sepsis model (Fig. 4 E-J), is designed to test both, detrimental and protective integrin functions simultaneously using CLP-sepsis as a model of microbial-driven inflammation. As a result, we suggest that unspecific integrin function may be inferior to a specific blockade. We have extensively re-organized our manuscript and changed the structure and title of the paper to

emphasize this dichotomy, rather than focusing solely on a novel sepsis treatment (New title: ‘*A ligand-specific blockade of the integrin Mac-1 selectively targets pathologic inflammation, while maintaining protective host-defense*’ instead: ‘*Leukocyte mobilization through CD40L/Mac-1 fuels polymicrobial sepsis in mice*’)

Our statement that anti-M7 treatment could potentially work in humans is based on (1) the high sequence homology of the mouse and human Mac-1 I-domain and (2) the fact that anti-M7 is an antibody that was raised on the human binding peptide sequence (for differences in sequences, please see Main Figure 1). (3) A major part of the validation work, including binding, specificity, intrinsic activity of anti-Mac-1 antibodies, has been tested with the help of anti-human antibodies, human proteins, human cells lines, and primary human cells (granulocytes, monocyte-derived macrophages). We agree with the thoughtful point of the Reviewer that moving towards a translation of these results is necessary in future. We are therefore planning to humanize this antibody and to initiate a human phase-I and –II clinical trial. This will ultimately test the applicability of this antibody in a translational context (much better than in adapted or ‘humanized’ mouse models).

We also like to comment on the two suggestions that the Reviewer made: crystalloids and antibiotics:

- 1) While fluid resuscitation is important, crystalloids, such as HES, have been shown to be associated with more complications, such as renal disease (Brunkhorst, Engel et al. 2008, Myburgh, Finfer et al. 2012).
- 2) Antibiotics are given as first line treatment and timely administration is likely to be the best predictor of survival (depending on the cause of sepsis: abscess, neutropenia, immune-suppression, endocarditis, ...). However, the CLP model is closest to abdominal sepsis, that in most cases is treated primarily with surgical removal of the infected part (appendix, abscess, tumor, ...). Since we intended to test the differential contribution of Mac-1 to aberrant inflammation versus host defense, we critically require bacteria to be present in this model. Antibiotic treatment would therefore not allow us to test the wanted effect of anti-M7/unwanted effect of anti-Mac-1 on phagocytosis and bacteremia. Also, antibiotics may have several side effects on leukocyte recruitment, ROS production, and the microcirculation (Al-Banna, Pavlovic et al. 2013) and would introduce a strong bias.

Having said that, to adequately accommodate the Reviewer’s point that one sepsis model is not enough, we performed a series of new *in vivo* experiment to substantiate the evidence of anti-M7 in sepsis:

- (1) less severe CLP model with a needle size of 30, instead of 30G;
- (2) a sterile LPS-sepsis model (no bacteria)
- (3) the previously used CLP model (23G) with a delayed injection of the antibody;
- (4) 30G-CLP model with female mice;
- (5) 30G-CLP model with aged mice.

These new results are now included and discussed in the revised version of our manuscript. Please see our detailed response in the comments below. We hope these experiments along with the modified scope of this manuscript, will satisfy the Reviewer.

Comment 2: Aging and gender: For research regarding inflammation biology or trauma, young mice are very much acceptable for basic science work. However, sepsis is considered a disease of the elderly (or neonates), and the literature has illustrated that the aged immune system is fundamentally different than that of younger mammals. The authors should repeat their work, or at least some of the latter aspects of this research, in

aged mice (18-24 month old) to strengthen their conclusions as well as help influence the current scientific thinking in this field. In addition, the NIH, as well as other national government agencies, are starting to insist that mixed gender experiments been done for animal research. Their work should included mixed gender animals instead of just male mice - trying to remove the influence of sex hormones from the results of sepsis research is no longer encouraged.

Authors: To address the Reviewer's important concern, we have now tested different ages (young vs. 40 weeks of age) and gender (male vs. female) and their impact on the number of granulocytes accumulating in the peritoneal cavity, which we consider representing the main readout. These new data are now presented in New Supplementary Fig. 6 and show that anti-M7 decreased the number of peritoneal granulocytes in each of the tested conditions (young, old, male, female mice). As expected, older animals had higher numbers of granulocytes in each of both conditions compared to 8-week-old mice. We hope to have clarified the Reviewer's concern.

New text: Results part, Page 6, Paragraph 2: We next performed a model of polymicrobial sepsis by a surgical cecal-ligation and puncture (CLP), that causes an increase in granulocytes and macrophages in the peritoneal cavity (Fig. 4C). A preventative injection of anti-M7 dampened this increase in myeloid cell recruitment to the peritoneal cavity in 8-week-old, male C57Bl/6 mice, while an unspecific anti-Mac-1 treatment was only partially effective (Fig. 4E). The effect of anti-M7 was consistent in female mice and aged mice (40 weeks, Supplemental Fig. 6). Anti-M7 failed when injected 2 hours after the onset of sepsis (Supplemental Fig. 7), confirming the necessity for an early intervention to reduce immediate myeloid cell infiltration (Buscher, Wang et al. 2016).

New text: Methods part, Page 11, Paragraph 4: Cecal ligation and puncture (CLP). Surgical induction of polymicrobial sepsis was performed according to available standard protocols (Rittirsch, Huber-Lang et al. 2009). The peritoneal cavity of 8- or 40-week old male C57Bl/6 mice was opened during ketamine/xylazine anesthesia, and the cecum was exteriorized and ligated about 2mm distal of the ileo-cecal valve using a non-absorbable 6-0 suture. As indicated, 8-week-old female C57Bl/6 mice were tested in addition. The distal end of the cecum was then perforated using a 23G or a 30G needle as indicated, and a small drop of feces was extruded through the puncture. The cecum was relocated into the peritoneal cavity, and the peritoneum was closed by a suture.

New text, Supplement, Page 7: Supplemental Figure 6: Anti-M7 prevents the accumulation of granulocytes in the peritoneal cavity independent of gender and age. Cecal-ligation and puncture (CLP) sepsis was induced in 8-week-old male or female or 40-week-old male C57Bl/6J mice. 50µg of the indicated F_{ab}-antibody preparations (IgG or anti-M7) were injected i.p. 1 hour before CLP surgery. 20 hours after CLP surgery, the number of granulocytes (Ly6-G⁺Mac-1⁺F4/80⁺) in the peritoneal cavity was quantified and expressed as % of IgG control in the same treatment group. Error bars indicate mean ± SEM. Significance was assessed by an unpaired, two-sided Student's T-test. * P < 0.05, ** P < 0.01, *** P < 0.001. N≥5 mice per group.

Comment 3: Timing of intervention of intraperitoneal injection of antibodies: Certainly, delivery of the antibody only some time after polymicrobial sepsis (and not at time point 0) would improve the excitement regarding this work (if it lead to improved outcomes). Multiple interventions have been demonstrated in the literature to improve outcomes in murine sepsis when given before or during the conclusion of the CLP operation, but have not lead to alterations in our understanding of human sepsis. Demonstrating the efficacy of the antibody when it is delivered only sometime after the infection has been induced would increase the translational and clinical interest of the work.

Authors: We thank the Reviewer for raising this interesting point. To address the Reviewer's suggestion we now tested the administration of our antibody at the previously reported time point and 2 hours after. These results are now shown in New Supplemental Figure 7. As expected by the Reviewer, the delayed treatment did not inhibit the accumulation of inflammatory leukocytes in the peritoneal cavity, which may be explained by the predominant rapid mobilization of most inflammatory cells in the first hours of abdominal sepsis (Hotchkiss and Karl 2003, Buscher, Wang et al. 2016). We observed that macrophage accumulation showed a tendency towards reduced numbers even after 2 hours, indicating that macrophage accumulation may also migrate into the peritoneal cavity at later stages. In this regard, our study supports the current thinking of disease development. The new data are now included and discussed in the revised manuscript.

New text: Results part, Page 6, Paragraph 2: We next performed a model of polymicrobial sepsis by a surgical cecal-ligation and puncture (CLP), that causes an increase in granulocytes and macrophages in the peritoneal cavity (Fig. 4C). A preventative injection of anti-M7 dampened this increase in myeloid cell recruitment to the peritoneal cavity in 8-week-old, male C57Bl/6 mice, while an unspecific anti-Mac-1 treatment was only partially effective (Fig. 4E). The effect of anti-M7 was consistent in female mice and aged mice (40 weeks, Supplemental Fig. 6). Anti-M7 failed when injected 2 hours after the onset of sepsis (Supplemental Fig. 7), confirming the necessity for an early intervention to reduce immediate myeloid cell infiltration (Buscher, Wang et al. 2016).

New text: Methods part, Page 11, Paragraph 4: Cecal ligation and puncture (CLP). Surgical induction of polymicrobial sepsis was performed according to available standard protocols (Rittirsch, Huber-Lang et al. 2009). The peritoneal cavity of 8- or 40-week old male C57Bl/6 mice was opened during ketamine/xylazine anesthesia, and the cecum was exteriorized and ligated about 2mm distal of the ileo-cecal valve using a non-absorbable 6-0 suture. As indicated, 8-week-old female C57Bl/6 mice were tested in addition. The distal end of the cecum was then perforated using a 23G or a 30G needle as indicated, and a small drop of feces was extruded through the puncture. The cecum was relocated into the peritoneal cavity, and the peritoneum was closed by a suture.

New text, Supplement, Page 8: Supplemental Figure 7: Only immediate, but not delayed, treatment with anti-M7 protects from peritoneal leukocyte accumulation in CLP-induced abdominal sepsis. Cecal-ligation and puncture (CLP) sepsis was induced in 8-week-old, male C57Bl/6J mice. 50µg of the indicated F_{ab}-antibody preparations (IgG or anti-M7) were injected i.p. 1 hour before surgery (0 hours) or with a delay of 2 hours. 20 hours later, the number of peritoneal exudate cells (PECs), granulocytes (Ly6-G⁺ Mac-1⁺ F4/80⁻), inflammatory (Ly6-C⁺) and patrolling (Ly6-C⁻) monocytes, F4/80^{low} and F4/80^{high} macrophages (Mac-1⁺ Ly-6G⁻), myeloid derived dendritic cells (CD11c⁺), and lymphocytes was quantified and expressed as % of IgG control in the same treatment group. Error bars indicate mean ± SEM. Significance was assessed by an unpaired, two-sided Student's T-test. ** P < 0.01, *** P < 0.001. N≥5 mice per group. n.s.: not significant.

Comment 4: Myeloid-derived suppressor cells: The authors work involves the analysis of granulocytes, macrophages, inflammatory monocytes and non-inflammatory monocytes. Myeloid-derived suppressor cells have recently been demonstrated to play a very important role in sepsis. It is unclear if the authors staining and isolation techniques do not include this immature myeloid cell population, which have been shown to significantly increase after murine and human sepsis. The authors should further evaluate the leukocytes they have analyzed to see if MDSCs make up some of the population. This would include seeing if the leukocytes they analyzed after CLP suppress T-cell function. This is particularly important as it's still unclear why exactly decreasing myeloid cell

accumulation to the site of infection specifically improves outcomes (although the authors speculate in the conclusions).

Authors: We thank the Reviewer for raising this issue. To accommodate the Reviewer's comment, we now quantified MDSCs. These cells are a part of a very heterogeneous cell population that is identified by the expression of Gr-1 (Ly6-C, Ly6-G) and CD11b (Mac-1). These cells have been reported to suppress T cell function and are elevated in sepsis (Brudecki, Ferguson et al. 2012, Dai, El Gazzar et al. 2015). Two major sub-populations of MDSCs have further been sub-classified: Mo-MDSC (immature monocytes, Ly6-G⁺Ly6-C⁺ Mac-1⁺) and PNM-MDSC (immature neutrophils, Ly6-G⁺Ly6-C^{low} Mac-1⁺). We performed an published gating strategy (Youn and Gabrilovich 2010, Bronte, Brandau et al. 2016), now included in Revised Figure 4 and Supplemental Figure 3. Following this strategy, we find that only Mo-MDSC are detected to relevant proportions in the peritoneal cavity, but not PNM-MDSCs. Further down-gating identified a proportion of the Mo-MDSCs as F4/80⁺ macrophages and CD11c⁺ dendritic cells. These have to be excluded by definition. The remnant cells are located in the gate named "Ly-6C⁺". The exact gating strategy and back-gating of these populations can be found in Supplemental Figure 3. This population is decreasing, suggesting, that if this population had immune-suppressive functions, the protective effect in the anti-M7 treatment group could at least be partially explained. Indeed, we also followed the Reviewer's comment (see below) to screen for T cell activation. We found an up-regulation of T cell activation markers (CD62L, CD44) after treatment with anti-M7, indicating that the T cell specific response is improved (New Supplemental Fig. 11). Also, we detected increased levels of IgM antibodies in the plasma of anti-M7 treated mice and decreased levels of the immune-suppressive cytokine IL-10 (New Supplemental Fig. 8). Please also see below in the next comments for a detailed description of the T cell phenotype. We hope the Reviewer will deem our new data as sufficient to address his point.

New text: Results part, Page 6, Paragraph 1: Aberrant activation of leukocytes and excessive cytokine secretion increases disease severity and mortality in sepsis (Sarangi, Hyun et al. 2012, Lerman, Lim et al. 2014, Chen, Ding et al. 2016). On the other hand, integrins are required for host defense, demonstrated by the requirement for Mac-1 to clear bacteria (Natanson, Hoffman et al. 1994, Rosenkranz, Coxon et al. 1998, Liu, Han et al. 2014, Jawhara, Pluskota et al. 2017). Clinically, blood concentrations of soluble CD40L (Lorente, Martin et al. 2011) and sCD18 (Kragstrup, Juul-Madsen et al. 2017) are associated with sepsis, suggesting that both, Mac-1 and CD40L, participate in the acute inflammatory response in sepsis. To outweigh its conflicting effects on host defense and pathologic inflammation, we hypothesized that a ligand-specific blockade would be superior to an unspecific inhibition of Mac-1 in polymicrobial sepsis. We first evaluated the effects of anti-M7 in a sterile systemic inflammatory response syndrome (SIRS) induced by an i.p. injection of LPS, a known TLR agonist. We applied a novel FACS gating strategy that made it possible to identify several Mac-1⁺ cell subsets, including granulocytes, F4/80^{low} (monocyte-derived) and F4/80^{high} (resident) macrophages, dendritic cells, myeloid-derived suppressor cells (MDSCs), and others (Fig. 4A, B, Supplemental Fig. 3). 20 hours after injection of a medium dose of LPS (20µg) we observed an increase in total cell numbers in the peritoneal cavity (not shown) with a relative increase of granulocytes (Fig. 4C). Anti-M7 efficiently blocked the infiltration of Mac-1⁺ myeloid cells, in particular of granulocytes and F4/80^{low} macrophages (Fig. 4D), validating its neutralizing effect on inflammatory cell recruitment. This effect was not caused by changes in the frequency of apoptosis in particular myeloid cell subsets, while an unspecific anti-Mac-1 inhibition increased the percentage of apoptotic macrophages (Supplemental Fig. 4). Also, we did not detect changes in the generation of reactive-oxygen-species as a surrogate for myeloid cell effector function (Supplemental Fig. 5) in anti-M7 treated mice.

New text: Results part, Page 6, Paragraph 3: Bacterial uptake and antigen-presentation is crucial to induce an antigen-specific immune response. In accordance, we detected higher numbers of activated T cells with an effector and memory phenotype, decreased systemic levels of the immune-suppressive cytokine IL-10 (Supplemental Fig. 8, 11), a reduced number of immune-dampening MDSCs in the peritoneal cavity (within the Ly6-C⁻ population, Fig. 4D), and increased IgM antibody levels (Supplemental Fig. 8), indicating an improvement in host defense.

New text: Discussion part, Page 8, Paragraph 1: During sepsis, some cytokines released by myeloid cells or lymphocytes can further fuel myeloid cell recruitment by driving monocyte egress from the bone marrow or the spleen (Swirski, Nahrendorf et al. 2009, Weber, Chousterman et al. 2015) or by secreting chemokines, such as MCP-1. Selective Mac-1 blockade by anti-M7 may avoid this positive feedback loop by inhibiting TNF α , MCP-1, or IL-10. In line, anti-M7 treatment attenuated inflammatory monocytosis post CLP. The decrease of IL-10 may also stem from a decrease of immune-suppressive MDSCs.

New text, Supplement, Page 4: Supplemental Figure 3: Identification of Myeloid-derived Suppressor Cells (MDSCs) in the peritoneal cavity by flow cytometry. Peritoneal cells were collected from 8-week-old, male C57Bl/6J mice by a peritoneal lavage 20 hours after an i.p. injection of 20 μ g LPS (0111:B4, Invivogen ultra-pure) and 50 μ g of the indicated antibodies (IgG or anti-M7). The two previously described MDSC-subpopulations, Monocyte-precursor (Mo-MDSC) and PMN-precursor (PMN-MDSC), were gated within viable, CD45⁺, Mac-1⁺ leukocytes based on Ly6-G and Ly6-C expression (A, left graph). Relative expression of Mac-1 (within Mac-1⁺ leukocytes with a color code from Mac-1⁺ min. to Mac-1⁺ max. is shown in (A, right graph). Color-coded backgating of both MDSC-subsets within the gates applied to define and quantify myeloid cell populations in the peritoneal cavity (Main Fig. 4). Representative plots are shown in (A, B).

Comment 5: In addition, it's still unclear if lowering cytokine attenuation is pathologic or beneficial – lack of TNF-alpha, IL-1 or IFN-gamma can be good or bad depending on the timing and location/organ of the cytokine/chemokine post infection.

Authors: This is a very important point highlighted by the Reviewer. The role of many cytokines in sepsis is indeed controversial. The best possible explanation is that a basal degree of cytokines is required to fully enable immune functions. However, over-shooting inflammation can be considered as harmful. During sepsis, an auto-amplifying secretion of cytokines is considered as one pathologic and causal hallmark, although clinical trials have failed to proof the efficacy of anti-cytokine therapy so far (Chousterman, Swirski et al. 2017). Some examples of such strategies in experimental sepsis include: Blocking IL-10, an immuno-suppressive cytokine, improves survival during CLP (Song, Chung et al. 1999). Genetic inhibition of IL-3, a driver of myelopoiesis, reduced mortality (Weber, Chousterman et al. 2015). Likewise, mice deficient for the receptor of TNF survived significantly longer than WT mice (Leon, White et al. 1998). In our study, we show that anti-M7 attenuates leukocyte recruitment without completely blocking it. As a result, cytokine concentrations (in part produced by invading leukocytes) are lowered, but not neutralized. In addition to its beneficial effect on phagocytosis and complement binding as well as on T cell activation (potentially by attenuated MDSC accumulation), this attenuated inflammatory response protects from sepsis-related mortality. Even though we did not focus on the precise role of cytokines, we show that such mild anti-inflammatory treatment is beneficial. To accommodate the Reviewers concerns we included our new results on cytokines (New Supplemental Fig. 8) in the revised version of the manuscript and included a more careful discussion of cytokines in sepsis.

New text: Results part, Page 6, Paragraph 1: Aberrant activation of leukocytes and excessive cytokine secretion increases disease severity and mortality in sepsis (Sarangi, Hyun et al. 2012, Lerman, Lim et al. 2014, Chen, Ding et al. 2016). On the other hand, integrins are required for host defense, demonstrated by the requirement for Mac-1 to clear bacteria (Natanson, Hoffman et al. 1994, Rosenkranz, Coxon et al. 1998, Liu, Han et al. 2014, Jawhara, Pluskota et al. 2017). Clinically, blood concentrations of soluble CD40L (Lorente, Martin et al. 2011) and sCD18 (Kragstrup, Juul-Madsen et al. 2017) are associated with sepsis, suggesting that both, Mac-1 and CD40L, participate in the acute inflammatory response in sepsis.

New text: Results part, Page 6, Paragraph 3: Bacterial uptake and antigen-presentation is crucial to induce an antigen-specific immune response. In accordance, we detected higher numbers of activated T cells with an effector and memory phenotype, decreased systemic levels of the immune-suppressive cytokine IL-10 (Supplemental Fig. 8, 11), a reduced number of immune-dampening MDSCs in the peritoneal cavity (within the Ly6C⁺ population, Fig. 4D), and increased IgM antibody levels (Supplemental Fig. 8), indicating an improvement in host defense. In contrast, histology of organs vulnerable for septic injury, such as the kidney, was not affected by an anti-M7 treatment 20 hours after CLP, although we detected an expected decrease in granulocyte accumulation (Supplemental Fig. 12, 13).

New text: Discussion part, Page 8, Paragraph 1: The role of pro-inflammatory cytokine secretion in sepsis remains controversial. Depending on the specific context, dose, and location, these can either improve or aggravate host defense (Chousterman, Swirski et al. 2017). However, some therapeutic strategies to block aberrant cytokines have been demonstrated efficacy in experimental sepsis: Blocking IL-10, an immuno-suppressive cytokine, improves survival during CLP (Song, Chung et al. 1999). Genetic inhibition of IL-3, a driver of myelopoiesis, reduced mortality (Weber, Chousterman et al. 2015). Likewise, mice deficient for the receptor of TNF survived significantly longer than WT mice (Leon, White et al. 1998). In this study, we observed that a treatment with anti-M7 decreased while treatment with a conventional anti-Mac-1 antibody clone (M1/70 in the mouse and 2LPM19c in humans) strongly increased systemic and local pro-inflammatory cytokine secretion. These findings confirm previous reports on isolated macrophages and likely result from outside-in signaling by non-selective antibody binding (Ding, Wright et al. 1987), an action avoided by anti-M7. The decrease in cytokines observed for anti-M7, in contrast, is likely caused by a reduction of cell infiltration and subsequent cytokine secretion by these cells³⁶ without an induction of outside-in-signaling. During sepsis, some cytokines released by myeloid cells or lymphocytes can further fuel myeloid cell recruitment by driving monocyte egress from the bone marrow or the spleen (Swirski, Nahrendorf et al. 2009, Weber, Chousterman et al. 2015) or by secreting chemokines, such as MCP-1. Selective Mac-1 blockade by anti-M7 may avoid this positive feedback loop by inhibiting TNF α , MCP-1, or IL-10. In line, anti-M7 treatment attenuated inflammatory monocytosis post CLP. The decrease of IL-10 may also stem from a decrease of immune-suppressive MDSCs.

New text: Methods part, Page 11, Paragraph 5: **Analysis of murine plasma and peritoneal lavage samples.** Plasma levels of Serum Amyloid-A (SAA), the mouse analogue for C-reactive Protein (CRP), were measured by ELISA, according to the manufacturer's protocols (USCN Life Science). Cytokines in the plasma were determined by a cytometric bead array (CBA, BD Biosciences), according to the manufacturer's

protocol. LPS titers in plasma and peritoneal lavage samples were quantified by a Limulus assay (Lonza), IgM titers by a ELISA (R&D Systems, USA).

New text, Supplement, Page 9: **Supplemental Figure 8: The CD40L/Mac-1 interaction improves innate immunity during polymicrobial sepsis.** Cecal-ligation and puncture (CLP) sepsis was induced in male, 8-week-old C57Bl/6J mice. 50µg of the indicated F_{ab}-antibody preparations were injected i.p. 1 hours before CLP surgery. 20 hours after CLP surgery, peripheral leukocytes were quantified in blood samples by flow cytometry: Inflammatory (Ly6-C⁺) and patrolling (Ly6-C⁻) CD115⁺ Mac-1⁺ monocytes (A), Ly6-G⁺ CD115⁻ Mac-1⁺ neutrophils (B), and CD19⁺ B cells (C). Cell numbers were expressed as % of all CD45⁺ leukocytes. (D) Levels of circulating IgM antibodies. (E) Plasma levels of the acute-phase protein Serum-Amyloid A (SAA) and of (F) IL-10 were quantified by ELISA in plasma samples. Error bars indicate mean ± SEM. Significance was assessed by an unpaired, two-sided Student's T-test. N≥10 mice per group.

Comment 6: Sacrifice times and CLP lethality: It's unclear in the manuscript and figures when the exact mouse sacrifice times were.

Authors: We apologize for omitting this important information in the old version of the manuscript. We sacrificed animals at a standardized time point of 20 hours post CLP surgery or LPS injection. We chose this time point as intermediate time in between macrophage accumulation (starting at 24-36 hours) and the peak of granulocyte infiltration (starting a few hours after LPS or CLP, peaking, and then decreasing, because neutrophils die very fast after invasion). We now included this relevant information in the revised methods section of the manuscript.

New text: Methods part, Page 11, Paragraph 4: **Cecal ligation and puncture (CLP).** Surgical induction of polymicrobial sepsis was performed according to available standard protocols (Rittirsch, Huber-Lang et al. 2009). The peritoneal cavity of 8- or 40-week old male C57Bl/6 mice was opened during ketamine/xylazine anesthesia, and the cecum was exteriorized and ligated about 2mm distal of the ileo-cecal valve using a non-absorbable 6-0 suture. As indicated, 8-week-old female C57Bl/6 mice were tested in addition. The distal end of the cecum was then perforated using a 23G or a 30G needle as indicated, and a small drop of feces was extruded through the puncture. The cecum was relocated into the peritoneal cavity, and the peritoneum was closed by a suture. Buprenorphine was applied as s.c. injection. 1 hour before or 2 hours after surgery F_{ab}-fragment preparations of the indicated antibodies (50µg of IgG, anti-M7, or anti-mouse Mac-1 (clone M1/70)) were injected i.p. After 20 hours, mice were sacrificed and a peritoneal lavage was performed. Organs were prepared for cell isolation or histology. A plasma sample was stored at -20°C for further analysis. Alternatively, a set of mice was followed up in survival analysis and received additional antibody injections at 48 and 96 hours after surgery. Cell infiltration in the peritoneal cavity was calculated as described above.

Comment 7: In addition, although significant, the improvement in mortality in the mice after CLP over the long-term is unclear in what appears to be a fairly lethal CLP model (all mice in the control group were dead prior to 7 days). Since longer term outcomes are of significant interest now in sepsis, the authors should follow the mice out for 10-14 days for survival. In addition, is the antibody as effective in a less lethal model of CLP, such as an LD40-50?

Authors:

Point 1: Longer follow-up: We fully agree with the Reviewer that a longer follow up is informative. In the data presented in our manuscript, within the observation of 7 days period 93% of IgG-treated animals (control, only one mouse remaining) and all of the anti-Mac-1 treated mice died.

50 % of the anti-M7 treated mice remained with no death within the last 50 of 150 hour observation period. Because we already observed an improvement at this time point (anti-M7 vs. IgG) we have not been granted a second validation survival experiment by our local animal committee that would include repeating the experiment to follow up the anti-M7 control group for an additional 3-7 days as requested by the Reviewer. We apologize for not being able to deliver this additional time point, but are also convinced that the main conclusion of this and other experiments in Fig. 4 will not change. Given the reproducibility of these results (in fact this phenotype is supported by previous evidence (Rosenkranz, Coxon et al. 1998, Liu, Han et al. 2014)), we would expect to see either sustained survival of anti-M7 treated mice over the requested, additional 3-7 days or delayed death of these mice. In both cases, maintained survival or delayed death, one would suggest that anti-M7 is more beneficial than anti-Mac-1 blockade.

Point 2: To address the Reviewer's concern we performed an additional, less lethal CLP model with a needle size of 30 G (instead of 23G). The size of the needle used to puncture the cecum, and the proportion of the ligated cecum define the severity of bacterial contamination and the clinical outcome/mortality. The lethality for our 30G-CLP models was ~93 % for the control within 7 days, which is in accord with previous reports for the 'high' ligation (>75 % of the cecum ligated) (Rittirsch, Huber-Lang et al. 2009). For a 23G model we would expect a lethality of ~30-40 % based on literature research (Ebong, Call et al. 1999, Drechsler, Weixelbaumer et al. 2011). Therefore, we performed a 30G CLP model with a sacrifice after 20 hours. These results are now shown in New Supplemental Fig. 9. We found that under this more moderate sepsis condition, anti-M7 treatment did not significantly change the proportion of accumulated leukocytes, suggesting that under a low-grade or moderate acute inflammation, anti-M7 is not effective. These results are in accord with our *in vitro* finding that full activation of the integrin (i.e. full activation of the Mac-1 bearing cell and integrin activation by inside-outside signaling) is necessary for the antibody to assess the binding epitope (New Supplemental Fig. 1). In addition, we performed a model of sterile sepsis-like inflammation (SIRS) by LPS injection. These results validate our main findings from the CLP-experiments with a decrease of myeloid cell populations, including MDSCs, in the peritoneal cavity. These results are now included in the revised Main Figure 4.

New text: Results part, Page 6, Paragraph 1: To outweigh its conflicting effects on host defense and pathologic inflammation, we hypothesized that a ligand-specific blockade would be superior to an unspecific inhibition of Mac-1 in polymicrobial sepsis. We first evaluated the effects of anti-M7 in a sterile systemic inflammatory response syndrome (SIRS) induced by an i.p. injection of LPS, a known TLR agonist. We applied a novel FACS gating strategy that made it possible to identify several Mac-1⁺ cell subsets, including granulocytes, F4/80^{low} (monocyte-derived) and F4/80^{high} (resident) macrophages, dendritic cells, myeloid-derived suppressor cells (MDSCs), and others (Fig. 4A, B, Supplemental Fig. 3). 20 hours after injection of a medium dose of LPS (20µg) we observed an increase in total cell numbers in the peritoneal cavity (not shown) with a relative increase of granulocytes (Fig. 4C). Anti-M7 efficiently blocked the infiltration of Mac-1⁺ myeloid cells, in particular of granulocytes and F4/80^{low} macrophages (Fig. 4D), validating its neutralizing effect on inflammatory cell recruitment. This effect was not caused by changes in the frequency of apoptosis in particular myeloid cell subsets, while an unspecific anti-Mac-1 inhibition increased the percentage of apoptotic macrophages (Supplemental Fig. 4). Also, we did not detect changes in the generation of reactive-oxygen-species as a surrogate for myeloid cell effector function (Supplemental Fig. 5) in anti-M7 treated mice.

New text: Results part, Page 6, Paragraph 2: **Systemically**, anti-M7 treatment attenuated the CLP-induced increase in inflammatory monocytosis and of the acute-phase protein SAA (Supplemental Fig. 8), indicating that anti-M7 specifically reduced the excessive inflammatory response during sepsis. Accordingly, we did not find an inhibitory effect of anti-M7 on peritoneal leukocyte recruitment in a less severe CLP sepsis model, where peritoneal cell numbers did not relevantly increase over baseline (Supplemental Fig. 9). Surprisingly, anti-M7 improved bacterial clearance in the plasma, while anti-Mac-1 worsened bacterial load in both, plasma and in the peritoneal cavity (Fig. 4F). This tempted us to directly test bacterial uptake, which was increased in anti-M7 treated mice, likely by favoring complement iC3b binding (Fig. 4G, Supplemental Fig. 10).

New text: Discussion part, Page 7, Paragraph 3: That anti-M7 specifically blocked the high-affinity interaction of Mac-1 with CD40L and therefore was selective to high-grade (pathologic), but not low-grade (beneficial), inflammation, might represent an additional mechanism by which a CD40L-specific blockade of Mac-1 could limit sepsis-related mortality. Here, we show by several *in vitro* and *in vivo* approaches that in addition to the established ligands ICAM-1 (Diamond, Staunton et al. 1990) and RAGE (Chavakis, Bierhaus et al. 2003), CD40L represents a powerful adhesive ligand that is expressed on inflamed endothelial cells (Wolf, Hohmann et al. 2011, Michel, Zirlik et al. 2017). Anti-M7 highly specifically blocked the interaction of Mac-1 with CD40L, but not with ICAM-1 or RAGE. Overall, anti-M7 was less effective compared to anti-Mac-1 or anti-ICAM-1 treatment. In addition, anti-M7 lost its ability to block leukocyte recruitment in the setting of a milder CLP procedure. These results indicate that the Mac-1/CD40L selectively contributes to aberrant inflammation.

New text: Methods part, Page 11, Paragraph 3: **LPS-induced inflammation.** A systemic inflammatory response syndrome (SIRS) was induced in 8-week-old, male C57Bl/6 mice by an i.p. injection of 20 μ g LPS (0111:B4, Invivogen, USA). Simultaneously, F_{ab}-fragment preparations of the indicated antibodies (50 μ g of IgG, anti-M7, or anti-mouse Mac-1 (clone M1/70)) were injected i.p. After 20 hours, mice were sacrificed and a peritoneal lavage was performed. Cell infiltration in the peritoneal cavity was calculated as follows: total count of cells (within the size range of 5 to 15 μ m, assessed by a Beckman Coulter Z2 cell counter) x fraction of cell sub-population of all leukocytes in flow cytometry and subsequent normalization as % of cell numbers in the IgG-treated control in the same experimental cohort.

New text: Methods part, Page 11, Paragraph 4: **Cecal ligation and puncture (CLP).** Surgical induction of polymicrobial sepsis was performed according to available standard protocols (Rittirsch, Huber-Lang et al. 2009). The peritoneal cavity of 8- or 40-week old male C57Bl/6 mice was opened during ketamine/xylazine anesthesia, and the cecum was exteriorized and ligated about 2mm distal of the ileo-cecal valve using a non-absorbable 6-0 suture. As indicated, 8-week-old female C57Bl/6 mice were tested in addition. The distal end of the cecum was then perforated using a 23G or a 30G needle as indicated, and a small drop of feces was extruded through the puncture. The cecum was relocated into the peritoneal cavity, and the peritoneum was closed by a suture.

New text, Supplement, Page 10: **Supplemental Figure 9: Anti-M7 does not modulate low-grade inflammatory cell accumulation in moderate CLP-30G sepsis.** Cecal ligation and puncture (CLP) sepsis was induced in 8-week-old male C57Bl/6J mice. For the puncture, a standard 23G needle or a 30G needle with a smaller outlet was used. 50 μ g of F_{ab}-preparations of the indicated antibodies (IgG or anti-M7) were injected i.p. one

hour before surgery. 20 hours after CLP surgery, the number of peritoneal exudate cells (PECs), granulocytes (Ly6-G⁺ Mac-1⁺ F4/80⁻), inflammatory (Ly6-C⁺) and patrolling (Ly6-C⁻) monocytes, F4/80^{low} and F4/80^{high} macrophages (Mac-1⁺ Ly-6G⁻), and myeloid derived dendritic cells (CD11c⁺) in the peritoneal cavity was quantified. Error bars indicate mean \pm SEM. Significance was assessed by an unpaired, two-sided Student's T-test. ** P < 0.01, *** P < 0.001, **** P < 0.0001. N \geq 5 mice per group.

Comment 8: Organ failure: Since decreased organ failure is proposed by the authors as a potential mechanism for their improved outcomes, the authors should evaluate lung, kidney and liver histology after CLP with and without the antibody and not just leukocyte infiltration.

Authors: As we performed a proof-of-concept study, we chose a time point for all standardized animal studies 20 hours after CLP surgery to quantify leukocyte recruitment in the peritoneal cavity. We have shown in the previous version of the manuscript that leukocyte accumulation was decreased in kidney tissue as this potentially precedes organ damage and failure (these results have now been moved to Supplemental Fig. 13). The Reviewer made the important suggestion to investigate organ histology to quantify organ damage. To adequately address this comment we performed H&E as well as PAS staining of kidneys, lungs, and liver. Representative pictures are now shown in Supplemental Fig. 12. In conclusion, we did not observe relevant organ damage at the (early) time point of 20 hours and could not detect relevant changes in organ histology between both groups, IgG and anti-M7. In our CLP model with a 23G needle, mortality was minimal in IgG-treated controls at the time point of 20 hours post surgery. We assume that at this time point leukocyte recruitment and bacteremia were improved by anti-M7 without obvious changes in organ histology. These results point towards a primary role of altered leukocyte accumulation in the peritoneal cavity. To accommodate the Reviewers well justified point we have now discussed the possibility of organ failure as potential cause of improved mortality in a more appropriate statement.

New text: Results part, Page 6, Paragraph 3: In contrast, histology of organs vulnerable for septic injury, such as the kidney, was not affected by an anti-M7 treatment 20 hours after CLP, although we detected an expected decrease in granulocyte accumulation (Supplemental Fig. 12, 13).

New text, Supplement, Page 13: **Supplemental Figure 12: anti-M7 does not impact on sepsis-related organ damage 20 hours after initiation of abdominal sepsis.** Cecal-ligation and puncture (CLP) sepsis was induced in male, 8-week-old C57Bl/6J mice. 50 μ g of F_{ab}-preparations of the indicated antibodies were injected i.p. one hour before surgery. 20 hours after CLP surgery, organs were excised, fixed in 4% PFA, sliced, and stained with HE or PAS. Representative tissue sections from 10 mice per group are shown.

New text, Supplement, Page 14: **Supplemental Figure 13: Anti-M7 reduces the accumulation of granulocytes in the kidney during abdominal sepsis.** Cecal-ligation and puncture (CLP) sepsis was induced in male, 8-week-old C57Bl/6J mice. 50 μ g of F_{ab}-preparation of the indicated antibodies (IgG, anti-M7, or the anti-mouse anti-Mac-1 clone M1/70) were injected i.p. one hour before CLP surgery. 20 hours after CLP surgery, kidneys were excised, fixed in 4% PFA, sliced, and stained with DAPI, and an antibody against the granulocyte epitope Ly6-G. In fluorescent microscopy, granulocytes were identified as DAPI⁺Ly6-G⁺ cells (A). Error bars indicate mean \pm SEM (B). Significance was assessed by an unpaired, two-sided Student's T-test between the indicated groups. N \geq 10 mice per group.

Comment 9: Innate Immune Function: The authors should conduct further experimentation to see how the antibody effects myeloid cell phagocytosis, generation of reactive oxygen species and T-cell activation/antigen presentation.

Authors: We thank the Reviewer for pointing us towards these highly interesting points. We like to comment in a point-by-point manner:

- Phagocytosis: Mac-1 contributes to bacterial clearance. This was demonstrated in Mac-1 KO mice, which showed defective phagocytosis and increased bacterial titers (Rosenkranz, Coxon et al. 1998, Liu, Han et al. 2014). In accordance, we have shown that anti-Mac-1 antibody treatment enhanced bacterial titers, while anti-M7 at least partially protected. We have therefore re-visited 2 functional hallmarks, complement binding and direct phagocytosis. First, we observed that anti-M7 increased complement binding. This was a highly un-anticipated finding. It is known that binding sites for iC3b are different from the CD40L binding epitope. However, it is not known whether CD40L does not (additionally) employ iC3b binding sites apart from its (unique) M7 binding site. While blocking anti-M7 may be sufficient to block CD40L, it could be possible that iC3b binding sites are now open again for complement binding. Another potential explanation is that the M7 sequence was described as binding site with a regulatory function that may result in enhanced binding of iC3B (Zhang and Plow 1996). This regulatory function may also cause the slightly elevated interaction with additional binding partners (see Main Fig. 1). Second, we directly tested uptake/phagocytosis of FITC-labeled bacteria by mouse peritoneal macrophages (Revised Fig. 4G,H). As expected, anti-Mac-1 treated macrophages showed less bacterial uptake, while anti-M7 treated macrophages demonstrated a higher uptake. In conclusion, these results indicate that anti-M7 may positively regulate bacterial clearance. These results are in accord with the (previously unexplained) findings from the first version of the manuscript that anti-M7 improved bacteremia. We thank the Reviewer for raising this possibility.
- ROS: We followed the Reviewers suggestion to quantify the generation of reactive-oxygen species after treatment of mouse macrophages with LPS and anti-M7 or anti-Mac-1. We did not find any differences in the ROS capacity of both treatment compared to an IgG control, although ROS production was higher in anti-Mac-1 vs. anti-M7. These results indicate that ROS generation was not affected in a relevant manner. These results are now shown in Supplemental Fig. 5 of the revised manuscript.
- Immunity: As pointed out correctly by the Reviewer, adaptive immune function is important in host defense against bacteria. T cells will guide this response by generating antigen-specific CD4⁺ T-helper cells that orchestrate the inflammatory milieu and activate B cells to produce specific antigen-neutralizing IgG antibodies (T-cell dependent B cell immunity). A hallmark of T cell activation after antigen-presentation is the generation of T-effector-memory cells (T_{EM}, CD44⁺CD62L⁻) and T-central-memory cells (T_{CM}, CD44⁺CD62L⁺). We have tested both activation states in CD4⁺ and CD8⁺ T cells and found that in anti-M7 treated mice, fractions of T_{EM} and T_{CM} were significantly higher in some of the tested locations. In addition, we observed that anti-M7 treatment elevated protective IgM titers. These results indicate an improvement in the adaptive immune response, which could theoretically be caused by (1) enhanced phagocytosis of bacteria and increased antigen-presentation, or (2) removal of suppressive MDSCs. These new results are now shown in Supplemental Fig. 3, 8, and 11.

New text: Results part, Page 6, Paragraph 1: **Aberrant activation of leukocytes and excessive cytokine secretion increases disease severity and mortality in sepsis**(Sarangi, Hyun et al. 2012, Lerman, Lim et al. 2014, Chen, Ding et al. 2016). On the other hand, integrins are required for host defense, demonstrated by the requirement for Mac-1 to clear bacteria(Natanson, Hoffman et al. 1994, Rosenkranz, Coxon et al. 1998, Liu, Han et al. 2014,

Jawhara, Pluskota et al. 2017). Clinically, blood concentrations of soluble CD40L(Lorente, Martin et al. 2011) and sCD18(Kragstrup, Juul-Madsen et al. 2017) are associated with sepsis, suggesting that both, Mac-1 and CD40L, participate in the acute inflammatory response in sepsis. To outweigh its conflicting effects on host defense and pathologic inflammation, we hypothesized that a ligand-specific blockade would be superior to an unspecific inhibition of Mac-1 in polymicrobial sepsis. We first evaluated the effects of anti-M7 in a sterile systemic inflammatory response syndrome (SIRS) induced by an i.p. injection of LPS, a known TLR agonist. We applied a novel FACS gating strategy that made it possible to identify several Mac-1⁺ cell subsets, including granulocytes, F4/80^{low} (monocyte-derived) and F4/80^{high} (resident) macrophages, dendritic cells, myeloid-derived suppressor cells (MDSCs), and others (Fig. 4A, B, Supplemental Fig. 3). 20 hours after injection of a medium dose of LPS (20µg) we observed an increase in total cell numbers in the peritoneal cavity (not shown) with a relative increase of granulocytes (Fig. 4C). Anti-M7 efficiently blocked the infiltration of Mac-1⁺ myeloid cells, in particular of granulocytes and F4/80^{low} macrophages (Fig. 4D), validating its neutralizing effect on inflammatory cell recruitment. This effect was not caused by changes in the frequency of apoptosis in particular myeloid cell subsets, while an unspecific anti-Mac-1 inhibition increased the percentage of apoptotic macrophages (Supplemental Fig. 4). Also, we did not detect changes in the generation of reactive-oxygen-species as a surrogate for myeloid cell effector function (Supplemental Fig. 5) in anti-M7 treated mice.

We next performed a model of polymicrobial sepsis by a surgical cecal-ligation and puncture (CLP), that causes an increase in granulocytes and macrophages in the peritoneal cavity (Fig. 4C). A preventative injection of anti-M7 dampened this increase in myeloid cell recruitment to the peritoneal cavity in 8-week-old, male C57Bl/6 mice, while an unspecific anti-Mac-1 treatment was only partially effective (Fig. 4E). The effect of anti-M7 was consistent in female mice and aged mice (40 weeks, Supplemental Fig. 6). Anti-M7 failed when injected 2 hours after the onset of sepsis (Supplemental Fig. 7), confirming the necessity for an early intervention to reduce immediate myeloid cell infiltration(Buscher, Wang et al. 2016). Systemically, anti-M7 treatment attenuated the CLP-induced increase in inflammatory monocytes and of the acute-phase protein SAA (Supplemental Fig. 8), indicating that anti-M7 specifically reduced the excessive inflammatory response during sepsis. Accordingly, we did not find an inhibitory effect of anti-M7 on peritoneal leukocyte recruitment in a less severe CLP sepsis model, where peritoneal cell numbers did not relevantly increase over baseline (Supplemental Fig. 9). Surprisingly, anti-M7 improved bacterial clearance in the plasma, while anti-Mac-1 worsened bacterial load in both, plasma and in the peritoneal cavity (Fig. 4F). This tempted us to directly test bacterial uptake, which was increased in anti-M7 treated mice, likely by favoring complement iC3b binding (Fig. 4G, Supplemental Fig. 10).

Bacterial uptake and antigen-presentation is crucial to induce an antigen-specific immune response. In accordance, we detected higher numbers of activated T cells with an effector and memory phenotype, decreased systemic levels of the immune-suppressive cytokine IL-10 (Supplemental Fig. 8, 11), a reduced number of immune-dampening MDSCs in the peritoneal cavity (within the Ly6-C⁻ population, Fig. 4D), and increased IgM antibody levels (Supplemental Fig. 8), indicating an improvement in host defense.

New text: Discussion part, Page 7, Paragraph 3: The β 2-integrins Mac-1 (CD11b/CD18), LFA-1 (CD11a/CD18), along with members of the β 1- and β 3-integrin family support leukocyte recruitment to inflamed tissue^{11, 12, 13, 56}. Although several studies have demonstrated that reducing myeloid cell accumulation by inhibition of the integrins α 3 β 1 and α V β 3 decreases aberrant leukocyte infiltration in sepsis and protects from sepsis-related mortality(Sarangi, Hyun et al. 2012, Lerman, Lim et al. 2014, Chen, Ding et al. 2016), neutralization of Mac-1 results in a strong inhibition of leukocyte mobilization, but also causes an accelerated bacterial sepsis with a higher mortality^{22, 23, 24, 52}. This phenomenon is best explained by the observation

that unlike $\alpha\beta 1$ and $\alpha V\beta 3$, Mac-1 is necessary to bind the complement factor iC3b, to opsonize bacteria, and to ultimately clear bacterial particles (Liu, Han et al. 2014)^{22, 23, 24, 52}. Unexpectedly, we found that binding of anti-M7 to its epitope M7 within the Mac-1 I-domain, increased iC3b binding, promoted phagocytosis, and improved adaptive T cell immunity. Currently, it remains unclear an engagement of the epitope M7 enhances iC3b binding. The binding epitopes for iC3b within the I-domain have been located to the residues P¹⁴⁷-R¹⁵², P²⁰¹-K²¹⁷, and K²⁴⁵-R²⁶¹ within the α_M I-domain (Ueda, Rieu et al. 1994, Zhang and Plow 1999, Li and Zhang 2003) and are therefore distinct from the epitope targeted by anti-M7 (E¹⁶²-L¹⁷⁰). P¹⁴⁷-R¹⁵² (according peptide M1), however, is a flanking sequence of M7 (Fig. 1A). In a previous study, the contribution of M1 to CD40L binding could not be clearly excluded (Wolf, Hohmann et al. 2011). It therefore remains speculative whether CD40L has a partial, low-affinity interaction with M1 that is lost after inhibition of M7, and thus re-accessible to iC3b binding. That levels of sCD40L are particularly increased in inflammation, including septic patients, has been shown before (Lorente, Martin et al. 2011). As proposed before, the M7 epitope may also be required to (sterically) regulate iC3b binding due to a regulatory activity of this domain (Zhang and Plow 1996).

New text: Methods part, Page 9, Paragraph 4: Apoptosis was quantified by binding of Annexin-V-Alexa488 and incubation in a Ca²⁺ rich Annexin-V-binding buffer (eBioscience). Generation of reactive-oxygen species (ROS) was quantified by incubation of cells with 1,2,3 Dihydrorhodamine (123 DHR, Invitrogen) for 30 minutes in the cell culture incubator at 37°C. Uptake of Alexa Fluor 488-labeled E. coli (Molecular Probes, Cat. E-13231) was evaluated by co-incubation of peritoneal macrophages with bacteria in the presence of freshly prepared opsonizing reagent (Molecular Probes, Cat E-2870) for 30 minutes and subsequent quantification of Alexa Fluor 488⁺ macrophages.

New text, Supplement, Page 6: Supplemental Figure 5: Anti-Mac-1 antibodies do not alter the production of reactive oxygen species (ROS) in macrophages. Peritoneal cells were collected from untreated, male 8-week-old C57Bl/6J mice by a peritoneal lavage. 400,000 cells were transferred into a 96-well plate in 50 μ l complete RPMI cell culture media containing 2 μ g/ml LPS (0111:B4, Invivogen ultra-pure) per well. The indicated antibodies were added to the wells at a final concentration of 20 μ g/ml (anti-mouse clone M1/70 as anti-Mac-1). After 4 hours, 123-Dihydro-Rhodamine (123 DHR) was added to the cells and incubated for 30 minutes at 37°C. Cells were washed and stained for flow cytometry. Macrophages were identified as F4/80⁺CD11b⁺Gr-1⁻CD45⁺ cells. The mean fluorescence (MFI) for 123 DHR was quantified. Error bars indicate mean \pm SEM. Significance was assessed by an unpaired, two-sided Student's T-test. * P < 0.05. N=5 donor mice per group. n.s.: not significant.

New text, Supplement, Page 9: Supplemental Figure 8: The CD40L/Mac-1 interaction improves innate immunity during polymicrobial sepsis. Cecal-ligation and puncture (CLP) sepsis was induced in male, 8-week-old C57Bl/6J mice. 50 μ g of the indicated Fab-antibody preparations were injected i.p. 1 hour before CLP surgery. 20 hours after CLP surgery, peripheral leukocytes were quantified in blood samples by flow cytometry: Inflammatory (Ly6-C⁺) and patrolling (Ly6-C⁻) CD115⁺ Mac-1⁺ monocytes (A), Ly6-G⁺ CD115⁻ Mac-1⁺ neutrophils (B), and CD19⁺ B cells (C). Cell numbers were expressed as % of all CD45⁺ leukocytes. (D) Levels of circulating IgM antibodies. (E) Plasma levels of the acute-phase protein Serum-Amyloid A (SAA) and of (F) IL-10 were quantified by ELISA in plasma samples. Error bars indicate mean \pm SEM. Significance was assessed by an unpaired, two-sided Student's T-test. N \geq 10 mice per group.

New text, Supplement, Page 11: **Supplemental Figure 10: Binding of the complement factor iC3b to peritoneal macrophages is enhanced after an *in vitro* treatment with anti-M7.** Peritoneal cells were collected from untreated, male 8-week-old C57Bl/6J mice by a peritoneal lavage. 400,000 leukocytes were transferred into a 96-well plate in 50 μ l complete RPMI cell culture media containing 1:125 cell stimulation cocktail. Fresh rat serum (containing iC3b) in a final concentration of 10% was added. The indicated antibodies (anti-mouse clone M1/70 as anti-Mac-1) were added to the wells at a final concentration of 20 μ g/ml. After 4 hours incubation at 37°C, cells were stained for flow cytometry with an antibody cocktail containing anti-iC3b. Macrophages were identified as F4/80⁺CD11b⁺Gr-1⁻CD45⁺ cells. A representative FACS plot of anti-iC3b gated on peritoneal macrophages is shown.

New text, Supplement, Page 12: **Supplemental Figure 11: Anti-M7 induces a state of enhanced T cell activation.** Peritoneal cells (perit.), splenocytes (spleen), or blood leukocytes were collected from male, 8-week-old C57Bl/6J mice 20 hours after an i.p. injection of 20 μ g LPS (0111:B4) and 50 μ g of the indicated antibodies (IgG or anti-M7). Cells were stained for flow cytometry and T cells were identified as CD45⁺L/D⁻TCR- β ⁺CD4/8⁺ single cells in the lymphocyte gate (**A**). The markers L-Selectin (CD62L) and CD44 were used to determine the activation state of T cells: T-effector memory cells (T_{EM}, CD44⁺CD62L⁻), T-central memory cells (T_{CM}, CD44⁺CD62L⁺), naïve T cells (T_{naïve}, CD44⁻CD62L⁻). In addition, CD25 was used as an activation marker (**B**). Percentage of T_{EM}, T_{CM}, T_{naïve}, and CD25⁺ T cells of all CD4⁺ or CD8⁺ T cells from the different sources. Error bars indicate mean \pm SEM. Significance was assessed by an unpaired, two-sided Student's T-test between IgG and anti-M7 treated groups in the same location. N \geq 5 donor mice per group.

Comment 10: The term "fuels" in the title is somewhat of a colloquialism and should be modified to another term.

Authors: We agree with the Reviewer and have re-worded the title: "A ligand-specific blockade of the integrin Mac-1 (CD11b/CD18) selectively targets pathologic inflammation, while maintaining protective host-defense".

Comment 11: What did the authors mean when they stated the mice were subjected to "wound healing" in the Methods section. I believe this is the equivalent of what is known as a "Sham" operation, and if this is true the wording should be changed for the audience.

Authors: We apologize for this mistake. We have conducted a set of wound healing experiments, which were removed from the initial manuscript, while we have not taken out the according part from the methods section. This was corrected. All mice tested in CLP underwent the same surgery, as controls we injected unspecific, control IgG antibodies.

Comment 12: The authors need to include as a supplemental figure how the flow cytometry analysis was conducted, including how the FSC and SSC were used to determine the leukocyte subsets as well as the subsequent white blood cell analysis.

Authors: We followed the reviewer's suggestion and have now included the comprehensive description of our gating strategy with all 15 parameters tested in the revised Figure 4 for all classical leukocyte subsets, and for MDSCs in the New Supplemental Fig. 3.

New text: Results part, Page 6, Paragraph 1: To outweigh its conflicting effects on host defense and pathologic inflammation, we hypothesized that a ligand-specific blockade would be superior to an unspecific inhibition of Mac-1 in polymicrobial sepsis. We first evaluated the effects of anti-M7 in a sterile systemic inflammatory response syndrome (SIRS) induced by an i.p. injection of LPS, a known TLR agonist. We applied a novel FACS gating strategy that made it possible to identify several Mac-1⁺ cell subsets,

including granulocytes, F4/80^{low} (monocyte-derived) and F4/80^{high} (resident) macrophages, dendritic cells, myeloid-derived suppressor cells (MDSCs), and others (Fig. 4A, B, Supplemental Fig. 3). 20 hours after injection of a medium dose of LPS (20µg) we observed an increase in total cell numbers in the peritoneal cavity (not shown) with a relative increase of granulocytes (Fig. 4C). Anti-M7 efficiently blocked the infiltration of Mac-1⁺ myeloid cells, in particular of granulocytes and F4/80^{low} macrophages (Fig. 4D), validating its neutralizing effect on inflammatory cell recruitment. This effect was not caused by changes in the frequency of apoptosis in particular myeloid cell subsets, while an unspecific anti-Mac-1 inhibition increased the percentage of apoptotic macrophages (Supplemental Fig. 4). Also, we did not detect changes in the generation of reactive-oxygen-species as a surrogate for myeloid cell effector function (Supplemental Fig. 5) in anti-M7 treated mice.

We next performed a model of polymicrobial sepsis by a surgical cecal-ligation and puncture (CLP), that causes an increase in granulocytes and macrophages in the peritoneal cavity (Fig. 4C). A preventative injection of anti-M7 dampened this increase in myeloid cell recruitment to the peritoneal cavity in 8-week-old, male C57Bl/6 mice, while an unspecific anti-Mac-1 treatment was only partially effective (Fig. 4E). The effect of anti-M7 was consistent in female mice and aged mice (40 weeks, Supplemental Fig. 6). Anti-M7 failed when injected 2 hours after the onset of sepsis (Supplemental Fig. 7), confirming the necessity for an early intervention to reduce immediate myeloid cell infiltration (Buscher, Wang et al. 2016).

New text: Methods part, Page 9, Paragraph 4: **Flow cytometry.** Peritoneal exudate cells (PECs), mouse peripheral blood samples, mouse splenocytes, or human blood samples from healthy volunteers were obtained as described. Red blood cells were lysed by a RBC-lysis buffer (eBioscience). Cells were washed multiple times in PBS and Fc- Receptors were blocked by an anti-mouse CD16/CD32 antibody cocktail (Tonbo) for 10 minutes on ice. Cells were then labeled with the indicated extracellular antibodies, and as indicated fixed in 4%PFA, permeabilized, and stained with intracellular antibodies before analysis on a flow cytometer (FACS Calibur or a FACS LSRII, BD Biosciences). Distinct leukocyte populations were identified by the gating strategies shown in Fig. 4 and Supplemental Fig. 3, 11.

Comment 13: Although there is a timing aspect to what cells are present at an infectious source to induce source control, I'm surprised that most of the peritoneal exudate cells (PECs) were F4/80 positive and macrophages. Usually in abdominal sepsis and pneumonia murine models a good portion of these cells are still neutrophils at the time points the authors analyzed the PECs.

Authors: We appreciate this helpful comment. Following the Reviewer's point, we now present additional leukocyte sub-populations in the revised version of our manuscript. These data are presented in revised Figure 4. To clarify the Reviewers question we now present the relative proportions of myeloid cell populations in untreated, LPS-treated, and after CLP (all IgG-treated) in Fig. 4 C. The percentage of granulocytes among all Mac-1⁺ lineage⁻ myeloid cells in the peritoneal cavity increases by 51-fold). We hope these data help to address the Reviewer's concern.

New text: Results part, Page 6, Paragraph 1: To outweigh its conflicting effects on host defense and pathologic inflammation, we hypothesized that a ligand-specific blockade would be superior to an unspecific inhibition of Mac-1 in polymicrobial sepsis. We first evaluated the effects of anti-M7 in a sterile systemic inflammatory response syndrome (SIRS) induced by an i.p. injection of LPS, a known TLR agonist. We applied a novel FACS gating strategy that made it possible to identify several Mac-1⁺ cell subsets,

including granulocytes, F4/80^{low} (monocyte-derived) and F4/80^{high} (resident) macrophages, dendritic cells, myeloid-derived suppressor cells (MDSCs), and others (Fig. 4A, B, Supplemental Fig. 3). 20 hours after injection of a medium dose of LPS (20µg) we observed an increase in total cell numbers in the peritoneal cavity (not shown) with a relative increase of granulocytes (Fig. 4C).

Comment 14: The authors need to report what the LD50 is in their model of CLP in their methods, or what the LD is over a set period of time (as CLP models can vary depending on how and who is performing them).

Authors: This is an important point raised by the Reviewer: The lethality for our 30G-CLP models was ~93 % for the control within 7 days, which is in accord with previous reports for the 'high' ligation (>75 % of the cecum ligated) (Rittirsch, Huber-Lang et al. 2009). For a 23G model we would expect a lethality of ~30-40 % based on literature research (Ebong, Call et al. 1999, Drechsler, Weixelbaumer et al. 2011). The expected needle size to reach a lethality of 50% would be >23G<30G. To make our procedure comparable to published CLP mouse models, we now present lethality over the observation period (~7 days, 170 hours), which was 60% for anti-M7, 93% for IgG, and 100% for anti-Mac-1.

New text: Results part, Page 6, Paragraph 3: Finally, we assessed whether our ligand-specific approach ultimately affects survival during sepsis. After CLP, mice received F_{ab}-preparations of IgG, anti-Mac-1, or anti-M7 at 1 hour before, 48, and 96 hours after surgery. Survival rates were calculated by Kaplan-Maier analysis and compared by log-rank testing. Anti-M7 treated showed a lethality of 60% at the end of the study compared with 93% in IgG treated mice (p=0.027, Fig. 4J). All animals treated with anti-Mac-1 died. These findings indicate that – in contrast to an unspecific blockade – a ligand specific anti-integrin therapy can reduce excessive inflammation without negatively affecting host defense.

References used in this response letter:

- Al-Banna, N. A., D. Pavlovic, M. Grundling, J. Zhou, M. Kelly, S. Whynot, O. Hung, B. Johnston, T. B. Issekutz, H. Kern, V. Cerny and C. Lehmann (2013). "Impact of antibiotics on the microcirculation in local and systemic inflammation." Clin Hemorheol Microcirc **53**(1-2): 155-169.
- Auffray, C., D. Fogg, M. Garfa, G. Elain, O. Join-Lambert, S. Kayal, S. Sarnacki, A. Cumano, G. Lauvau and F. Geissmann (2007). "Monitoring of blood vessels and tissues by a population of monocytes with patrolling behavior." Science **317**(5838): 666-670.
- Bronte, V., S. Brandau, S. H. Chen, M. P. Colombo, A. B. Frey, T. F. Greten, S. Mandruzzato, P. J. Murray, A. Ochoa, S. Ostrand-Rosenberg, P. C. Rodriguez, A. Sica, V. Umansky, R. H. Vonderheide and D. I. Gabrilovich (2016). "Recommendations for myeloid-derived suppressor cell nomenclature and characterization standards." Nat Commun **7**: 12150.
- Brudecki, L., D. A. Ferguson, C. E. McCall and M. El Gazzar (2012). "Myeloid-derived suppressor cells evolve during sepsis and can enhance or attenuate the systemic inflammatory response." Infect Immun **80**(6): 2026-2034.
- Brunkhorst, F. M., C. Engel, F. Bloos, A. Meier-Hellmann, M. Ragaller, N. Weiler, O. Moerer, M. Gruendling, M. Oppert, S. Grond, D. Olthoff, U. Jaschinski, S. John, R. Rossaint, T. Welte, M. Schaefer, P. Kern, E. Kuhnt, M. Kiehntopf, C. Hartog, C. Natanson, M. Loeffler, K. Reinhart and S. German Competence Network (2008). "Intensive insulin therapy and pentastarch resuscitation in severe sepsis." N Engl J Med **358**(2): 125-139.
- Buscher, K., H. Wang, X. Zhang, P. Striewski, B. Wirth, G. Saggi, S. Lutke-Enking, T. N. Mayadas, K. Ley, L. Sorokin and J. Song (2016). "Protection from septic peritonitis by rapid neutrophil recruitment through omental high endothelial venules." Nat Commun **7**: 10828.
- Chavakis, T., A. Bierhaus, N. Al-Fakhri, D. Schneider, S. Witte, T. Linn, M. Nagashima, J. Morser, B. Arnold, K. T. Preissner and P. P. Nawroth (2003). "The pattern recognition receptor (RAGE) is a counterreceptor for leukocyte integrins: a novel pathway for inflammatory cell recruitment." J Exp Med **198**(10): 1507-1515.
- Chen, Z., X. Ding, S. Jin, B. Pitt, L. Zhang, T. Billiar and Q. Li (2016). "WISP1-alpha/beta3 integrin signaling positively regulates TLR-triggered inflammation response in sepsis induced lung injury." Sci Rep **6**: 28841.
- Chousterman, B. G., F. K. Swirski and G. F. Weber (2017). "Cytokine storm and sepsis disease pathogenesis." Semin Immunopathol **39**(5): 517-528.
- Coxon, A., P. Rieu, F. J. Barkalow, S. Askari, A. H. Sharpe, U. H. von Andrian, M. A. Arnaout and T. N. Mayadas (1996). "A novel role for the beta 2 integrin CD11b/CD18 in neutrophil apoptosis: a homeostatic mechanism in inflammation." Immunity **5**(6): 653-666.
- Dai, J., M. El Gazzar, G. Y. Li, J. P. Moorman and Z. Q. Yao (2015). "Myeloid-derived suppressor cells: paradoxical roles in infection and immunity." J Innate Immun **7**(2): 116-126.
- Diamond, M. S., D. E. Staunton, A. R. de Fougerolles, S. A. Stacker, J. Garcia-Aguilar, M. L. Hibbs and T. A. Springer (1990). "ICAM-1 (CD54): a counter-receptor for Mac-1 (CD11b/CD18)." J Cell Biol **111**(6 Pt 2): 3129-3139.

Ding, A., S. D. Wright and C. Nathan (1987). "Activation of mouse peritoneal macrophages by monoclonal antibodies to Mac-1 (complement receptor type 3)." J Exp Med **165**(3): 733-749.

Ding, Z. M., J. E. Babensee, S. I. Simon, H. Lu, J. L. Perrard, D. C. Bullard, X. Y. Dai, S. K. Bromley, M. L. Dustin, M. L. Entman, C. W. Smith and C. M. Ballantyne (1999). "Relative contribution of LFA-1 and Mac-1 to neutrophil adhesion and migration." J Immunol **163**(9): 5029-5038.

Drechsler, S., K. M. Weixelbaumer, H. Redl, M. van Griensven, S. Bahrami and M. F. Osuchowski (2011). "Experimentally approaching the ICU: monitoring outcome-based responses in the two-hit mouse model of posttraumatic sepsis." J Biomed Biotechnol **2011**: 357926.

Dunne, J. L., C. M. Ballantyne, A. L. Beaudet and K. Ley (2002). "Control of leukocyte rolling velocity in TNF-alpha-induced inflammation by LFA-1 and Mac-1." Blood **99**(1): 336-341.

Ebong, S., D. Call, J. Nemzek, G. Bolgos, D. Newcomb and D. Remick (1999). "Immunopathologic alterations in murine models of sepsis of increasing severity." Infect Immun **67**(12): 6603-6610.

Ehlers, R., V. Ustinov, Z. Chen, X. Zhang, R. Rao, F. W. Luscinskas, J. Lopez, E. Plow and D. I. Simon (2003). "Targeting platelet-leukocyte interactions: identification of the integrin Mac-1 binding site for the platelet counter receptor glycoprotein Ibalpha." J Exp Med **198**(7): 1077-1088.

Fan, Z. and K. Ley (2015). "Leukocyte arrest: Biomechanics and molecular mechanisms of beta2 integrin activation." Biorheology **52**(5-6): 353-377.

Fan, Z., S. McArdle, A. Marki, Z. Mikulski, E. Gutierrez, B. Engelhardt, U. Deutsch, M. Ginsberg, A. Groisman and K. Ley (2016). "Neutrophil recruitment limited by high-affinity bent beta2 integrin binding ligand in cis." Nat Commun **7**: 12658.

Fossati-Jimack, L., G. S. Ling, A. Cortini, M. Szajna, T. H. Malik, J. U. McDonald, M. C. Pickering, H. T. Cook, P. R. Taylor and M. Botto (2013). "Phagocytosis is the main CR3-mediated function affected by the lupus-associated variant of CD11b in human myeloid cells." PLoS One **8**(2): e57082.

Hotchkiss, R. S. and I. E. Karl (2003). "The pathophysiology and treatment of sepsis." N Engl J Med **348**(2): 138-150.

Jawhara, S., E. Pluskota, W. Cao, E. F. Plow and D. A. Soloviev (2017). "Distinct Effects of Integrins alphaXbeta2 and alphaMbeta2 on Leukocyte Subpopulations during Inflammation and Antimicrobial Responses." Infect Immun **85**(1).

Kellum, J. A., L. Kong, M. P. Fink, L. A. Weissfeld, D. M. Yealy, M. R. Pinsky, J. Fine, A. Krichevsky, R. L. Delude, D. C. Angus and I. M. S. I. Gen (2007). "Understanding the inflammatory cytokine response in pneumonia and sepsis: results of the Genetic and Inflammatory Markers of Sepsis (GenIMS) Study." Arch Intern Med **167**(15): 1655-1663.

Kolaczowska, E. and P. Kuberski (2013). "Neutrophil recruitment and function in health and inflammation." Nat Rev Immunol **13**(3): 159-175.

- Kragstrup, T. W., K. Juul-Madsen, S. H. Christiansen, X. Zhang, J. Krog, T. Vorup-Jensen and A. G. Kjaergaard (2017). "Altered levels of soluble CD18 may associate immune mechanisms with outcome in sepsis." Clin Exp Immunol **190**(2): 258-267.
- Leon, L. R., A. A. White and M. J. Kluger (1998). "Role of IL-6 and TNF in thermoregulation and survival during sepsis in mice." Am J Physiol **275**(1 Pt 2): R269-277.
- Lerman, Y. V., K. Lim, Y. M. Hyun, K. L. Falkner, H. Yang, A. P. Pietropaoli, A. Sonnenberg, P. P. Sarangi and M. Kim (2014). "Sepsis lethality via exacerbated tissue infiltration and TLR-induced cytokine production by neutrophils is integrin alpha3beta1-dependent." Blood **124**(24): 3515-3523.
- Ley, K., C. Laudanna, M. I. Cybulsky and S. Nourshargh (2007). "Getting to the site of inflammation: the leukocyte adhesion cascade updated." Nat Rev Immunol **7**(9): 678-689.
- Li, Y. and L. Zhang (2003). "The fourth blade within the beta-propeller is involved specifically in C3bi recognition by integrin alpha M beta 2." J Biol Chem **278**(36): 34395-34402.
- Liu, J. R., X. Han, S. G. Soriano and K. Yuki (2014). "The role of macrophage 1 antigen in polymicrobial sepsis." Shock **42**(6): 532-539.
- Liu, J. R., X. Han, S. G. Soriano and K. Yuki (2014). "The Role of Macrophage-1 Antigen in Polymicrobial Sepsis." Shock.
- Lorente, L., M. M. Martin, N. Varo, J. M. Borreguero-Leon, J. Sole-Violan, J. Blanquer, L. Labarta, C. Diaz, A. Jimenez, E. Pastor, F. Belmonte, J. Orbe, J. A. Rodriguez, E. Gomez-Melini, J. M. Ferrer-Aguero, J. Ferreres, M. C. Lliminana and J. A. Paramo (2011). "Association between serum soluble CD40 ligand levels and mortality in patients with severe sepsis." Crit Care **15**(2): R97.
- Lu, H., C. W. Smith, J. Perrard, D. Bullard, L. Tang, S. B. Shappell, M. L. Entman, A. L. Beaudet and C. M. Ballantyne (1997). "LFA-1 is sufficient in mediating neutrophil emigration in Mac-1-deficient mice." J Clin Invest **99**(6): 1340-1350.
- Michel, N. A., A. Zirlik and D. Wolf (2017). "CD40L and Its Receptors in Atherothrombosis-An Update." Front Cardiovasc Med **4**: 40.
- Myburgh, J. A., S. Finfer, R. Bellomo, L. Billot, A. Cass, D. Gattas, P. Glass, J. Lipman, B. Liu, C. McArthur, S. McGuinness, D. Rajbhandari, C. B. Taylor, S. A. Webb, C. Investigators, Australian and G. New Zealand Intensive Care Society Clinical Trials (2012). "Hydroxyethyl starch or saline for fluid resuscitation in intensive care." N Engl J Med **367**(20): 1901-1911.
- Natanson, C., W. D. Hoffman, A. F. Suffredini, P. Q. Eichacker and R. L. Danner (1994). "Selected treatment strategies for septic shock based on proposed mechanisms of pathogenesis." Ann Intern Med **120**(9): 771-783.
- Quintar, A., S. McArdle, D. Wolf, A. Marki, E. Ehinger, M. Vassallo, J. Miller, Z. Mikulski, K. Ley and K. Buscher (2017). "Endothelial Protective Monocyte Patrolling in Large Arteries Intensified by Western Diet and Atherosclerosis." Circ Res **120**(11): 1789-1799.
- Rittirsch, D., M. S. Huber-Lang, M. A. Flierl and P. A. Ward (2009). "Immunodesign of experimental sepsis by cecal ligation and puncture." Nat Protoc **4**(1): 31-36.

Rosenkranz, A. R., A. Coxon, M. Maurer, M. F. Gurish, K. F. Austen, D. S. Friend, S. J. Galli and T. N. Mayadas (1998). "Impaired mast cell development and innate immunity in Mac-1 (CD11b/CD18, CR3)-deficient mice." J Immunol **161**(12): 6463-6467.

Rosetti, F. and T. N. Mayadas (2016). "The many faces of Mac-1 in autoimmune disease." Immunol Rev **269**(1): 175-193.

Sarangi, P. P., Y. M. Hyun, Y. V. Lerman, A. P. Pietropaoli and M. Kim (2012). "Role of beta1 integrin in tissue homing of neutrophils during sepsis." Shock **38**(3): 281-287.

Schuler, P., D. Assefa, J. Ylänne, N. Basler, M. Olschewski, I. Ahrens, T. Nordt, C. Bode and K. Peter (2003). "Adhesion of monocytes to medical steel as used for vascular stents is mediated by the integrin receptor Mac-1 (CD11b/CD18; alphaM beta2) and can be inhibited by semiconductor coating." Cell Commun Adhes **10**(1): 17-26.

Simon, D. I. (2011). "Opening the field of integrin biology to "biased agonism"." Circ Res **109**(11): 1199-1201.

Sisco, M., J. D. Chao, I. Kim, J. E. Mogford, T. N. Mayadas and T. A. Mustoe (2007). "Delayed wound healing in Mac-1-deficient mice is associated with normal monocyte recruitment." Wound Repair Regen **15**(4): 566-571.

Song, G. Y., C. S. Chung, I. H. Chaudry and A. Ayala (1999). "What is the role of interleukin 10 in polymicrobial sepsis: anti-inflammatory agent or immunosuppressant?" Surgery **126**(2): 378-383.

Swirski, F. K., M. Nahrendorf, M. Etzrodt, M. Wildgruber, V. Cortez-Retamozo, P. Panizzi, J. L. Figueiredo, R. H. Kohler, A. Chudnovskiy, P. Waterman, E. Aikawa, T. R. Mempel, P. Libby, R. Weissleder and M. J. Pittet (2009). "Identification of splenic reservoir monocytes and their deployment to inflammatory sites." Science **325**(5940): 612-616.

Ueda, T., P. Rieu, J. Brayer and M. A. Arnaout (1994). "Identification of the complement iC3b binding site in the beta 2 integrin CR3 (CD11b/CD18)." Proc Natl Acad Sci U S A **91**(22): 10680-10684.

Wang, Y., H. Gao, C. Shi, P. W. Erhardt, A. Pavlovsky, A. S. D, K. Bledzka, V. Ustinov, L. Zhu, J. Qin, A. D. Munday, J. Lopez, E. Plow and D. I. Simon (2017). "Leukocyte integrin Mac-1 regulates thrombosis via interaction with platelet GPIIb/IIIa." Nat Commun **8**: 15559.

Weber, G. F., B. G. Chousterman, S. He, A. M. Fenn, M. Nairz, A. Anzai, T. Brenner, F. Uhle, Y. Iwamoto, C. S. Robbins, L. Noiret, S. L. Maier, T. Zonnchen, N. N. Rahbari, S. Scholch, A. Klotzsche-von Ameln, T. Chavakis, J. Weitz, S. Hofer, M. A. Weigand, M. Nahrendorf, R. Weissleder and F. K. Swirski (2015). "Interleukin-3 amplifies acute inflammation and is a potential therapeutic target in sepsis." Science **347**(6227): 1260-1265.

Wolf, D., N. Bukosza, D. Engel, M. Poggi, F. Jehle, N. Anto Michel, Y. C. Chen, C. Colberg, N. Hoppe, B. Dufner, L. Boon, H. Blankenbach, I. Hilgendorf, C. von Zur Muhlen, J. Reinohl, B. Sommer, T. Marchini, M. A. Febbraio, C. Weber, C. Bode, K. Peter, E. Lutgens and A. Zirlik (2017). "Inflammation, but not recruitment, of adipose tissue macrophages requires signalling through Mac-1 (CD11b/CD18) in diet-induced obesity (DIO)." Thromb Haemost **117**(2): 325-338.

Wolf, D., J. D. Hohmann, A. Wiedemann, K. Bledzka, H. Blankenbach, T. Marchini, K. Gutte, K. Zeschky, N. Bassler, N. Hoppe, A. O. Rodriguez, N. Herr, I. Hilgendorf, P. Stachon, F. Willecke, D. Duerschmied, C. von zur Muhlen, D. A. Soloviev, L. Zhang, C. Bode, E. F. Plow, P. Libby, K. Peter and A. Zirlik (2011). "Binding of CD40L to Mac-1's I-domain involves the EQLKKSRTL motif and mediates leukocyte recruitment and atherosclerosis--but does not affect immunity and thrombosis in mice." Circ Res **109**(11): 1269-1279.

Youn, J. I. and D. I. Gabrilovich (2010). "The biology of myeloid-derived suppressor cells: the blessing and the curse of morphological and functional heterogeneity." Eur J Immunol **40**(11): 2969-2975.

Zhang, L. and E. F. Plow (1996). "A discrete site modulates activation of I domains. Application to integrin alphaMbeta2." J Biol Chem **271**(47): 29953-29957.

Zhang, L. and E. F. Plow (1999). "Amino acid sequences within the alpha subunit of integrin alpha M beta 2 (Mac-1) critical for specific recognition of C3bi." Biochemistry **38**(25): 8064-8071.

Zirlik, A., C. Maier, N. Gerdes, L. MacFarlane, J. Soosairajah, U. Bavendiek, I. Ahrens, S. Ernst, N. Bassler, A. Missiou, Z. Patko, M. Aikawa, U. Schonbeck, C. Bode, P. Libby and K. Peter (2007). "CD40 ligand mediates inflammation independently of CD40 by interaction with Mac-1." Circulation **115**(12): 1571-1580.

Reviewer #1 (Remarks to the Author):

The authors have sufficiently addressed my concerns and added many valuable experiments. I have one last question: In Figure 2E (new) anti Mac-1 treatment leads to a strong reduction in rolling. How do you explain this? Does your anti Mac-1 treatment affect systemic leukocyte counts?

Please comment.

Reviewer #2 (Remarks to the Author):

This resubmission was very responsive to my critique, as well as the other reviewers. The manuscript now provides impactful mechanistic insight to Mac-1's role in sepsis and how a novel anti CD40L site confers protection.

Reviewer #3 (Remarks to the Author):

significant revisions and additional experiments done, data reported even when negative results. critiques addressed.

Response-to-Reviewers

We thank the Reviewers for their thorough, fair, and positive evaluation of our manuscript “A ligand-specific blockade of the integrin Mac-1 selectively targets pathologic inflammation while maintaining protective host-defense”. We addressed each of the Reviewers’ comments that are quoted in bold (marked as Reviewer) and are directly followed by our response (marked as Authors). **New text is marked with red.**

Referee 1: The authors have sufficiently addressed my concerns and added many valuable experiments. I have one last question: In Figure 2E (new) anti Mac-1 treatment leads to a strong reduction in rolling. How do you explain this? Does your anti Mac-1 treatment affect systemic leukocyte counts? Please comment.

Authors: We thank the Reviewer for deeming our new experiments valuable. The Reviewer’s observation that anti-Mac-1 (clone M1/70) reduces leukocyte rolling is correct and it is indeed a surprising finding. Our best explanation is that Mac-1 may actually mediate leukocyte rolling depending on the specific context (location, type of vessel, stimulation). Indeed, it has been shown in intravital microscopy of TNF- α stimulated cremasteric venules that Mac-1, but not LFA-1, is responsible for (slow) leukocyte rolling (Dunne et al. 2003; Dunne et al. 2002). In line, in our experiments, LFA-1 blockade had no inhibitory effect on leukocyte rolling. The slight increase in rolling leukocytes observed after blockade of LFA-1, ICAM-1, and RAGE indicates that activated leukocytes (which lost the ability to firmly adhere) keep rolling. In conclusion, our results suggest that leukocyte rolling in TNF- α primed mesenteric venules may also depend on Mac-1. We agree that more experimental work is certainly needed to clarify the exact context of this unexpected effect. We have now added this potential explanation and the mentioned citations to the Discussion section of our revised manuscript.

In addition, to exclude that anti-Mac-1 reduces the number of circulating leukocytes (by depleting effects or inducing apoptotic pathways), we have quantified leukocyte numbers 4 and 24 hours after injection of anti-mouse Mac-1 (M1/70) compared with an appropriate IgG control in male C57Bl6/J mice. We did not observe relevant differences between both groups. We present the results of this experiment exclusively to the Reviewer in this response letter (see graph on the right side).

To experimentally address the Reviewer’s second question whether our anti-M7 treatment affects leukocyte numbers, we performed an additional experiment to test for potential long-term depleting effects in the sepsis experiments. We have therefore quantified leukocyte counts before, 3 and 6 days after i.p. injection. We did not observe a significant change of leukocyte counts at the tested time points between anti-M7 and the IgG control. Thus, anti-M7 does not seem to modulate the pool of available circulating leukocytes. These new data are now shown as new Supplementary Figure 3. We hope these new data can help to clarifying the

Reviewer's question.

New data (shown in Supplementary Figure 3):

New text (Results Section, Page 4, Paragraph 2): To test the antibody's *in vivo* applicability, we prepared F_{ab}-fragments of anti-M7, anti-Mac-1 (M1/70), a control-IgG, and antibodies directed against ICAM-1, LFA-1 (CD11/CD18), and RAGE to avoid unspecific F_c-receptor interactions. **Anti-M7 did not change peripheral leukocyte counts after i.p. injection (Supplementary Figure 3).** In intravital microscopy (Figure 2C) of inflamed mesenteric venules, the number of adhering (Figure 2D) but not of rolling leukocytes (Figure 2E) fell after anti-M7 injection.

New text (Discussion Section, Page 7, Paragraph 4): Anti-M7 specifically blocked the interaction of Mac-1 with CD40L, but not with ICAM-1 or RAGE. Overall, anti-M7 was less effective compared to anti-Mac-1 or anti-ICAM-1 treatment. **In contrast to an unspecific anti-Mac-1 treatment, which blocked adhesion and rolling of leukocytes, anti-M7 only blocked cellular adhesion.** These findings are in accord with previous observations (Dunne et al. 2003; Dunne et al. 2002; Wolf et al. 2011) and point out that the interaction of Mac-1 with one or more of its alternative ligands, but not with CD40L, may mediate leukocyte rolling in TNF- α stimulated mesenteric venules. Notably, neither anti-Mac-1 (data not shown), nor anti-M7 caused a depletion of leukocytes, ruling out a reduction of the circulating leukocyte pool as potential explanation for this effect. In addition, anti-M7 lost its ability to block leukocyte recruitment in the setting of a milder CLP procedure. These results indicate that the Mac-1/CD40L interaction selectively contributes to aberrant inflammation **by blocking firm adhesion and transmigration of leukocytes in the setting of acute inflammation.**

New text (Supplementary Figures): **Supplementary Figure 3: Anti-M7 does not reduce peripheral leukocyte counts after i.p. injection.** Anti-M7 was injected i.p. in male, C57Bl/6J mice and peripheral leukocyte counts were quantified before (0 days) and 3 and 6 days after injection. Leukocyte counts were normalized to counts before injections in each individual animal (expressed as %). Error bars indicate mean \pm SEM. Statistical significance was tested by an unpaired, two-sided Student's T-test

between IgG and anti-M7 injected mice at each time point. N=5 animals were included per group.

Referee 1: This resubmission was very responsive to my critique, as well as the other reviewers. The manuscript now provides impactful mechanistic insight to Mac-1's role in sepsis and how a novel anti CD40L site confers protection.

Authors: We thank the Reviewer for this positive feed-back.

Referee 1: Significant revisions and additional experiments done, data reported even when negative results. critiques addressed.

Authors: We thank the Reviewer for assessing our changes to the revised manuscript as significant.

CITATIONS USED IN THIS RESPONSE LETTER:

- Dunne, J. L., C. M. Ballantyne, A. L. Beaudet, and K. Ley. 2002. 'Control of leukocyte rolling velocity in TNF-alpha-induced inflammation by LFA-1 and Mac-1', *Blood*, 99: 336-41.
- Dunne, J. L., R. G. Collins, A. L. Beaudet, C. M. Ballantyne, and K. Ley. 2003. 'Mac-1, but not LFA-1, uses intercellular adhesion molecule-1 to mediate slow leukocyte rolling in TNF-alpha-induced inflammation', *J Immunol*, 171: 6105-11.
- Wolf, D., J. D. Hohmann, A. Wiedemann, K. Bledzka, H. Blankenbach, T. Marchini, K. Gutte, K. Zeschky, N. Bassler, N. Hoppe, A. O. Rodriguez, N. Herr, I. Hilgendorf, P. Stachon, F. Willecke, D. Duerschmied, C. von zur Muhlen, D. A. Soloviev, L. Zhang, C. Bode, E. F. Plow, P. Libby, K. Peter, and A. Zirlik. 2011. 'Binding of CD40L to Mac-1's I-domain involves the EQLKKSRTL motif and mediates leukocyte recruitment and atherosclerosis--but does not affect immunity and thrombosis in mice', *Circ Res*, 109: 1269-79.